# BVOC and speciated monoterpene concentrations and fluxes at a Scandinavian boreal forest

Ross C. Petersen[1], Thomas Holst[1], Cheng Wu[2,3,a], Radovan Krejci[2,3], Jeremy K.Chan[4], Claudia Mohr[2,3,b,c], and Janne Rinne[1, 5]

[1]Department of Physical Geography and Ecosystem Science, Lund University, Lund, Sweden
[2]Department of Environmental Science, Stockholm University, Stockholm, Sweden
[3]Bolin Centre for Climate Research, Stockholm University, 11418, Stockholm, Sweden
[4]Center for Volatile Interactions (VOLT), Department of Biology, University of Copenhagen, Universitetsparken 15, DK-2100 Copenhagen Ø, Denmark
[5]Bioeconomy and Environment, Natural Resources Institute Finland (Luke), Helsinki, Finland
[a]now at: Department of Chemistry and Molecular Biology, University of Gothenburg, 41296, Gothenburg, Sweden
[b]now at: Department of Environmental Systems Science, ETH Zürich, 8006 Zürich, Switzerland
[c]now at: PSI Center for Energy and Environmental Sciences, Paul Scherrer Institute, 5232 Villigen, Switzerland

*Correspondence to*: Janne Rinne (janne.rinne@luke.fi)

**Abstract.** Boreal forests emit terpenoid biogenic volatile organic compounds (BVOCs) that significantly impact atmospheric chemistry. Our understanding of the variation of BVOC species emitted from boreal ecosystems is based on relatively few datasets, especially at the ecosystem-level. We conducted measurements to obtain BVOC flux observations above the boreal forest at the ICOS (Integrated Carbon Observation System) station Norunda in central Sweden. The goal was to study concentrations and fluxes of terpenoids, including isoprene, speciated monoterpenes (MT), and sesquiterpenes (SQT), during a Scandinavian summer. Measurements (10 Hz sampling) from a Vocus proton-transfer-reaction time-of-flight mass spectrometer (Vocus PTR-ToF-MS) were used to quantify a wide range of BVOC fluxes, including total MT (386 ($\pm$ 5) ng m$^{-2}$ s$^{-1}$; $\beta = 0.1$ ℃$^{-1}$), using the eddy-covariance (EC) method. Surface-layer-gradient (SLG) flux measurements were performed on selected daytime sampling periods, using thermal-desorption adsorbent tube sampling, to establish speciated MT fluxes. The impact of chemical degradation on measured terpenoid fluxes relative to surface exchange rates (F/E) was also investigated using stochastic Lagrangian transport modeling in forest-canopy. While the impact on isoprene was within EC-flux uncertainty ($F_{ISO}/E_{ISO}<5\%$), the effect on SQT and nighttime MT was significant, with average F/E-ratios for nighttime $F_{MT}/E_{MT}$ = ca. 0.9 (0.87 - 0.93), nighttime $F_{SQT}/E_{SQT}$ = 0.35 (0.31 - 0.41) and daytime $F_{SQT}/E_{SQT}$ = 0.41 (0.37 - 0.47). The main compounds contributing to MT flux were $\alpha$-pinene and $\Delta^3$-carene. Summer shifts in speciated MT emissions for $\Delta^3$-carene were detected, featuring a decrease in its relative fraction among observed MT compounds from June to August sampling periods, indicating that closer attention to seasonality of individual MT species in BVOC emission and climate models is warranted.

## 1 Introduction

Biogenic volatile organic compounds (BVOCs) play a central role in tropospheric chemistry, influencing both regional air quality and global climate (Fall, 1999). Many BVOCs participate in new particle formation (NPF) and growth (Bianchi et al., 2016; Boucher et al., 2013; Kirkby et al., 2016; Mohr et al., 2019; Riipinen et al., 2012; Tunved et al., 2006; Went, 1960). As BVOCs react with ozone, OH, and $NO_3$ (in the case of nighttime chemistry), the subsequent reaction products often have lower volatility that in suitable conditions can condense into new secondary organic aerosol (SOA) particles or contribute to the growth of existing aerosol particles (Bonn et al., 2009; Hallquist et al., 2009; Hodzic et al., 2016; Kulmala et al., 2004). BVOCs also affect the production and lifetime of tropospheric ozone through their photooxidation in the presence of NOx, as well as through their interactions with OH and other radicals (Atkinson and Arey, 2003). As reactive BVOCs compete with methane for reacting with ambient OH, they may also have an influence on the atmospheric lifetime of this greenhouse gas (e.g., Kaplan et al., 2006). Oxygenated VOCs, such as acetone – one of the most abundant in the atmosphere (e.g., Singh et al., 1994), can also modify OH concentrations in the upper troposphere (Fehsenfeld et al., 1992; Mckeen et al., 1997; Monks, 2005) and/or contribute to the formation of peroxyacyl nitrates (PAN) that can act as a reservoir for NOx (Read et al., 2012; Roberts et al., 2002).

Globally, biogenic VOC (BVOC) emissions are several times greater than anthropogenic emissions, accounting for up to ca. 90% of total VOC emissions worldwide (Guenther et al., 1995; Müller, 1992). In densely populated European regions, anthropogenic VOC emissions are estimated to exceed biogenic emissions, but in most cases, particularly those areas which are sparsely populated, biogenic emissions are dominant (Lindfors and Laurila, 2000; Simpson et al., 1999). These include sparsely populated countries in the European boreal zone (Simpson et al., 1999). The boreal vegetation zone, one of the Earth's largest terrestrial biomes and forming a near-continuous band around the Northern Hemisphere, is one of the major sources of BVOCs to the atmosphere at the global scale (Guenther et al., 1995; Sindelarova et al., 2014). Across all plant functional types (e.g., Guenther et al., 2012), boreal MT emissions make up as much as ca. 26.3 % of the global summertime MT emission inventory (June: 28.3 %, July: 29.7 %, and August: 20.7 %), and up to ca. 37.1 % of it (June: 39.2 %, July: 41 %, and August: 30.5 %) for the Northern Hemisphere (Sindelarova et al., 2022).

Terpenoid compounds are an important fraction of BVOCs emitted globally (Guenther et al., 1995) and also from boreal forests (Rinne et al., 2009), and include isoprene ($C_5H_8$), the monoterpenes ($C_{10}H_{16}$), the sesquiterpenes ($C_{15}H_{24}$), and so on to diterpenes and larger compounds. Typically, for European boreal forests such as those largely composed of Scots pine and Norway Spruce (Rinne et al., 2009), isoprene emissions are relatively low while monoterpene emissions predominate during typical ambient conditions (Hakola et al., 2006; Hakola et al., 2017), with isoprene emission about 10% - 15% of monoterpene emissions by mass (Rinne et al., 2009). When under significant stresses such as insect herbivory or drought, boreal forests are also known to be significant sesquiterpene emitters (Rinne et al., 2009; Niinemets, 2010). Isoprene is mainly involved in

influencing production and lifespan of tropospheric ozone (Atkinson and Arey, 2003) but is relatively ineffective at enhancing tropospheric SOA yields compared to MT. In the lower troposphere, isoprene can even inhibit SOA new particle formation (NPF) and early-stage particle growth, such as when present as an isoprene + MT mixture relative to a pure MT mixture (e.g., Heinritzi et al., 2020; Kiendler-Scharr et al., 2009; Mcfiggans et al., 2019). However, monoterpenes as such emitted by the boreal forest biome are mainly involved in SOA particle formation and growth (Spracklen et al., 2008).

Globally, α-Pinene, β-pinene and limonene typically have the highest net atmospheric abundance for monoterpenes (MT). Many monoterpene isomers, however, are present and vary widely in structure (Geron et al., 2000; Guenther et al., 2012). In boreal forests, the emission of MT from predominant tree species, particularly Scots pine and Norway spruce, can vary as distinct chemotypes (referred to as α-pinene and $\Delta^3$-carene types) (Hiltunen and Laakso, 1995; Holzke et al., 2006; Janson,

1993; Komenda and Koppmann, 2002; Manninen et al., 2002; Tarvainen et al., 2005). The chemical speciation of MT emissions can also vary significantly within the same tree species population (e.g., Persson et al., 2016). This has direct impacts on the resulting atmospheric chemistry, as MT reactivity and SOA formation potential varies by isomeric structure (Friedman and Farmer, 2018; Griffin et al., 1999; Lee et al., 2006a; Zhao et al., 2015). For example, Lee et al. (2006b) reported SOA yields from ozonolysis of monoterpenes ranging from 17% for β-pinene to 54% for $\Delta^3$-carene (41% for α-pinene) and in

photolysis experiments (Lee et al., 2006a) found that $\Delta^3$-carene had an SOA mass yield that is ca. 16 – 30 % and 15 – 22 % greater than that of β-pinene and α-pinene, respectively. Accurately characterizing the speciation of MT fluxes from boreal forests is therefore of significant importance to parameterizing chemical species-specific emission factors in current and future climate and air quality models.

Observations of BVOC concentration above forest canopy are important for interpreting atmospheric chemistry processes, particularly in the lower troposphere and surface boundary layer directly above the forest, which are dependent on the concentration of precursor BVOC compounds (Atkinson, 2000; Pryor et al., 2014; Tunved et al., 2006). Observed concentrations are impacted by both BVOC emission rate and micrometeorological conditions in the surface boundary layer (e.g., Karl et al., 2004; Petersen et al., 2023), as well as the chemical lifetime of relatively short-lived BVOC compounds

(Atkinson and Arey, 2003). Very brief chemical lifetimes, such as for sesquiterpenes ($\tau_{SQT} \sim$ seconds), can impact their observed above-canopy fluxes as well (e.g., Rinne et al., 2012). Evaluating the ecosystem-atmosphere exchange of reactive BVOCs in the absence of chemical degradation, and hence isolating the roles of surface emission, deposition, and physical transport from its effects, represents an important goal for separating the relative influences on BVOC ecosystem-scale surface exchange and physical transport processes from atmospheric chemistry. Meanwhile, BVOC flux observations are important

for understanding BVOC exchange between forest ecosystems and the atmosphere. Quantifying fluxes is also important for accurately parameterizing the functional dependencies of BVOC emissions on environmental parameters, such as temperature and solar radiation, as well as non-constitutive influences on ecosystem-scale emissions such as drought and disturbance stress, for regional and global atmospheric chemistry models (e.g. Rinne et al., 2007; Taipale et al., 2011).

There are multiple methods used to measure BVOC concentrations and fluxes (Rantala et al., 2014; Rinne et al., 2021). Proton-transfer-reaction mass spectrometry (PTR-MS) has been widely used to study BVOCs in the atmosphere (e.g., Yuan et al., 2017; Lindinger et al., 1998). In this technique, proton-transfer reactions with $H3O^+$ ions are used to ionize atmospheric VOCs for detection of the product ions by mass spectrometry. In recent years, it has become possible to perform eddy-covariance (EC) measurements of BVOC fluxes using fast-response ($\tau \lesssim 1$ s) PTR Time-of-Flight MS (PTR-ToF-MS) over a wide range of BVOC molar masses (Müller et al., 2010). The EC flux approach, based on the covariance between fast (10 – 20 Hz) observations of the fluctuations in chemical concentration and vertical wind speed (see section 2.4), has the advantage of being the most direct and accurate approach for measuring ecosystem-level BVOC fluxes, and thus is an important component for biosphere BVOC emission research. Fluxes measured by micrometeorological methods can be used to study the effects of environmental parameters on BVOC emissions. They can also be applied in up-scaling studies, where canopy-scale measurements provide an important intermediate step between leaf-level and regional-scale for use in model verification (Guenther, 2012; Peñuelas and Staudt, 2010; Rinne et al., 2009). A high sampling rate is essential to resolve fluctuations from small, short-lived eddies (0.1–5 Hz) that drive turbulent transport, as lower rates can lead to significant attenuation of measured fluxes due to unresolved turbulence. The high sampling rate capability of PTR-ToF-MS (>10 Hz) makes it well-suited for measuring BVOC fluxes using the EC method. Additionally, EC-based methods utilizing PTR-ToF-MS can be implemented for various mobile platforms, including aircraft, for spatially resolved landscape-scale flux assessments over wide areas (Pfannerstill et al., 2023). When combined with the high sensitivity and accuracy of modern instrumentation (e.g., Krechmer et al., 2018), PTR-ToF-MS stands as one of the most effective tools currently available for measuring ecosystem-scale BVOC fluxes. A limitation of PTR-MS analysis is, however, that it identifies compounds collectively by their mass-to-charge ratio, and cannot differentiate between compounds with the same molar mass and molecular composition (i.e. isomeric compounds). For example, the monoterpenes ($C_{10}H_{16}$), which can be emitted in a boreal forest as any one of many structurally unique compounds, all have a protonated nominal mass-to-charge (m/z) ratio of 137.130.

While BVOC flux studies have made great strides due the development of PTR instrumentation, improvements to the study of speciated MT fluxes are still lacking. In addition to total fluxes (such as total monoterpene flux), speciated flux information can be used to further improve model verification for compound-specific emission potentials used in emission models (e.g., MEGAN)(Guenther et al., 2012). Methods used to measure speciated BVOC fluxes include variants of the gradient method (see section 2.6), which assumes that a compound's ecosystem-atmosphere turbulent flux is proportional to its vertical concentration gradient above the forest canopy (where the proportionality constant is the turbulent exchange coefficient) (Fuentes et al., 1996; Goldstein et al., 1995; Guenther et al., 1996; Rinne et al., 2000a; Schween et al., 1997). By using Automated Thermal Desorption (ATD) gas chromatograph-mass spectrometery (GC-MS) for adsorbent tube sampling of the vertical BVOC gradient, in conjunction with PTR-ToF-MS measurements of the eddy-covariance-derived BVOC fluxes, the

information from these two approaches to BVOC flux observations can provide additional insight into the ecosystem-atmosphere exchange of BVOCs between boreal forests and the troposphere.

In this work, concentrations and fluxes of BVOCs, particularly the monoterpenes, at a Swedish site in the European boreal zone are presented. Six weeks of Vocus PTR-ToF-MS measurements were performed from July 21 to August 26, 2020. Thermal desorption (TD) tube samples of the BVOC concentration were collected during the daytime (typically between 9:00am and 5:00pm) at 37 m and 60 m a.g.l. on the Norunda flux tower over 3-day periods, during June 8-10 (prior to Vocus deployment), July 22-24, and August 16-18.

## 2 Methods

### 2.1 Measurement site

This study was conducted at the Norunda research station, located at 60°05′N, 17°29′E, ca. 30 km north of Uppsala, in central Sweden. The station is part of the Integrated Carbon Observation System (ICOS) research infrastructure ([www.icos-cp.eu](www.icos-cp.eu); (Heiskanen et al., 2022)), a network for monitoring greenhouse gases and short-lived climate forcers (more recently, part of Aerosol, Clouds, and Trace Gases Research Infrastructure (ACTRIS) in Sweden ([www.actris.eu](www.actris.eu)) as well). The station was surrounded by a mixed-conifer forest of Scots pine (*Pinus sylvestris*) and Norway spruce (*Picea abies*). This forest was between 80 and 120 years old (Lagergren et al., 2005) and the forest canopy height was ca. 28 m (Wang et al., 2017). The forest (subsequently clearcut in 2022 for lumber) has been managed for economic purposes for approximately the last 200 years. The research station has been in operation since 1994 as a $CO_2$ and trace gas flux station and is equipped with a 102m tower (Lindroth et al., 1998; Lundin et al., 1999). A station map and its location in Sweden are presented in Figure 1.

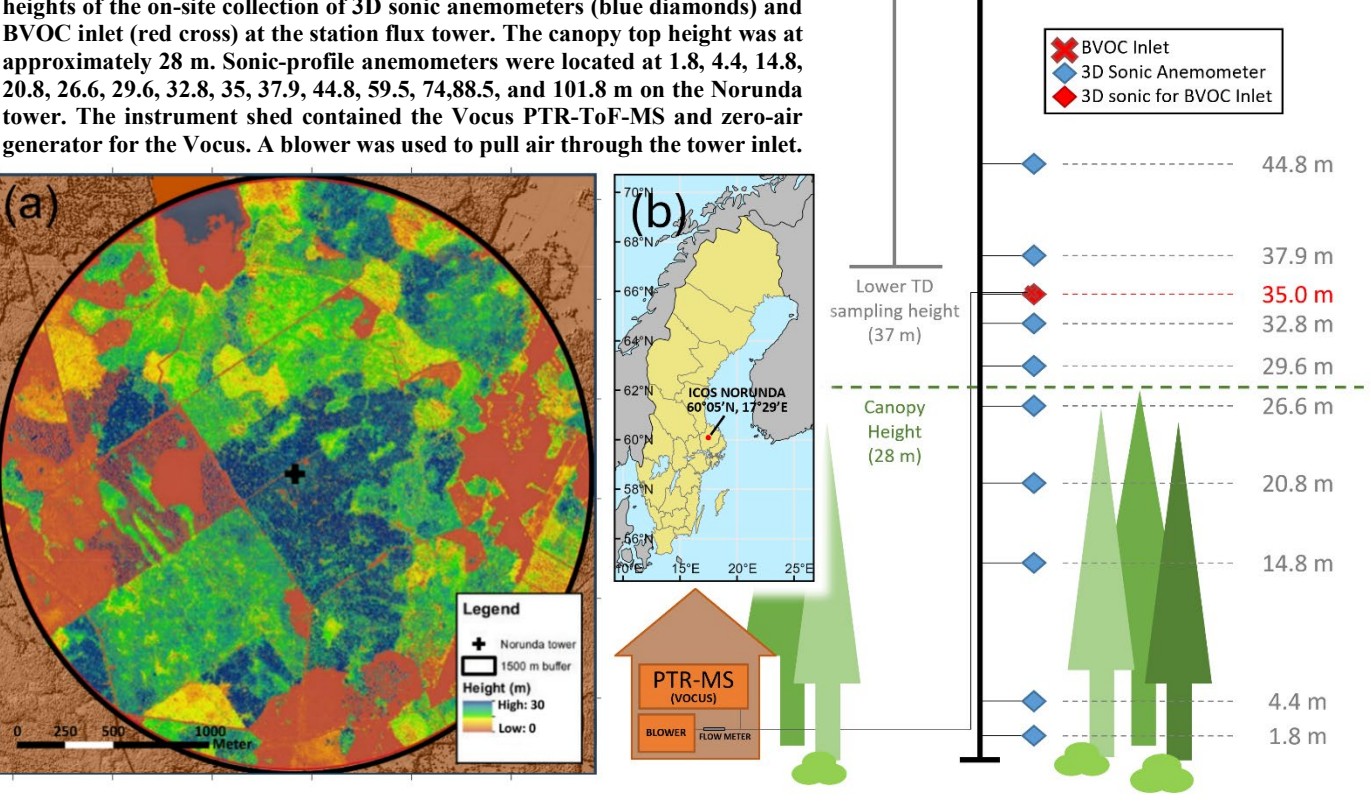

**Figure 1: Forest map, station location, and BVOC inlet setup for ICOS Norunda.** (a) A map of tree heights surrounding the station flux tower (out to 1500 m radially from tower base) for the Norunda forest. (b) Location and coordinates of ICOS station Norunda in Sweden. (c) BVOC inlet, infrastructure, and instrumentation setup for Vocus PTR-ToF-MS measurements on the Norunda tower (BVOC inlet at 35 m). Shown are the heights of the on-site collection of 3D sonic anemometers (blue diamonds) and BVOC inlet (red cross) at the station flux tower. The canopy top height was at approximately 28 m. Sonic-profile anemometers were located at 1.8, 4.4, 14.8, 20.8, 26.6, 29.6, 32.8, 35, 37.9, 44.8, 59.5, 74,88.5, and 101.8 m on the Norunda tower. The instrument shed contained the Vocus PTR-ToF-MS and zero-air generator for the Vocus. A blower was used to pull air through the tower inlet.

Together, Norway spruce (53.4 %) and Scots pine (39.8 %) make up 93.2 % of mean tree number per hectare. There was also a small number of deciduous trees, consisting primarily of Black Alder (*Alnus glutinosa (L.) Gaertn*; 2.5 %) and Downy Birch (*Betula pubescens Ehrh.*; 3.9 %). The dominant ground vegetation at the station was Bilberry (*Vaccinuim myrtillus*) and Lingonberry (*Vaccinium vitis-idaea*), in addition to several species of dwarf-shrubs, ferns, and grasses. The bottom layer vegetation predominantly consisted of a thick layer of feather moss (*Pleurozium Schreberi* and *Hylocomium Splendens*). During the 25 years prior to 2020, the mean monthly temperature varied between -5°C and 25°C, and the mean annual precipitation was approximately 540 mm. The growing season, with daily air temperatures above 5 °C, ranges typically from May to September. During the same calendar period as the collected 2020 Norunda campaign measurements (8 June – August 28), the local 25-year climatological average daily temperature was $16.4 \pm 2.7$ °C, with an average nighttime minimum of $10.7 \pm 3$ °C and daytime maximum of $21.2 \pm 3.6$ °C. New needle growth typically begins in April. Foliation of existing deciduous trees and plants usually occurs in May and senescence usually in October (± 15 days). From 2009 to 2014, the leaf area index

(LAI) of the Norunda forest in proximity of the tower was determined to be approximately 3.6 (±0.4) using a LAI 2000 (Li Cor Inc., Lincoln Nebraska, USA).

## 2.2 Instrumentation & sampling setup

The canopy temperature of the Norunda forest was measured using a precision SI-111 infrared radiometer (Campbell Scientific Inc., Logan UT, USA) mounted at 55 m on the station flux tower. The wind velocity components were measured using a three-dimensional sonic anemometer (USA-1, Metek GmbH, Germany). BVOC volume mixing ratios were measured using a Vocus PTR-ToF-MS (Vocus-2R, TOFWERK, Thun, Switzerland) (Krechmer et al., 2018). The VOCUS has several advantages over previous PTR-ToF-MS designs, such as in detecting low-volatility BVOC compounds (Krechmer et al., 2018). The Metek sonic anemometer and BVOC sampling inlet were co-located at 35 m on the Norunda flux tower. Ambient air at 35 m was transported from the BVOC inlet head through a heated and insulated perfluoroalkoxy (PFA) Teflon tube mounted on the station flux tower (Fig. 1) to the instrument shed housing the Vocus PTR-ToF-MS using a side-channel blower. The tower Teflon tubing was ca. 49m in length with an outer diameter (OD) of 3/8 in (inner diameter 1/2 in). The flow rate through the BVOC inlet tubing to the Vocus was 20 L/min to minimize BVOC losses from prolonged residence time within sample tubing. Sample air residence time in tower Teflon tubing before reaching instrumentation was ca. 4.7 ± 1 s.

Inside the instrumentation shed, 6 L/min of the flow coming from the main inlet tubing were directed to the Vocus-inlet, and 100 ml/min were sampled into the Vocus, and the remainder was directed to the inlet exhaust. The Vocus ion source was set at 2 mbar. From field measurements, the mass-resolving power (m/$\Delta$m) of the Vocus, where $\Delta$m is the full-width at half-maximum (fwhm) of a spectrum peak, was found to be ca. 9900 Th/Th fwhm for the MT parent ion $C_{10}H_{16}H^+$ (nominal m/z = 137). During the campaign, data from the Vocus were recorded at a frequency of 10 Hz. Reference measurements to determine the instrumental background of the Vocus PTR-MS were periodically performed (for 1 min each hr) using zero air from a heated catalytic converter (Zero Air Generator, Parker Balston, Haverhill, MA, USA). The mass flow controller for the Vocus zero-air was set to 500 mL/min. The Vocus calibration was performed using a gravimetrically prepared calibration standard (Ionimed Analytik, Innsbruck, Austria). The calibration standard gas was diluted before sampling by the Vocus using a Gas Calibration Unit (GCU-b, Ionicon, Innsbruck, Austria).

## 2.3 Vocus data processing

For compounds present in the calibration gas standard bottle, the corresponding calibration factor was applied directly to the processed Vocus trace data. For the remaining compounds, calibration factors were calculated using the analysis approach implemented by the *PTR data toolkit* as described by Jensen et al. (2023)(See table A2 of calibration coefficients and calculated terpenoid sensitivities). The fractionation rate estimate for sesquiterpenes (0.20±0.1) is based on *PTR Library* reference of proton-transfer reactions in trace gas sampling (Pagonis et al., 2019).

The processing of the raw Vocus data was performed using Julia-based analysis scripts from the software package *TOF-Tracer2* (Breitenlechner et al., 2017; Fischer et al., 2021; Stolzenburg et al., 2018), modified for use with the Vocus PTR-ToF-MS dataset. For each campaign day, all Vocus spectra data acquired within 24 h were mass-scale calibrated every 6 minutes and averaged for peak shape analysis. The program *PeakFit* (Fischer et al., 2021) was used for peak fitting and identification.

*PeakFit* was used to create a mass peak list of more than 2000 identified compounds for our Vocus dataset. Based on this mass peak list and using the modified *TOF-Tracer2* scripts, the 10Hz time traces were obtained by integrating the spectra for intervals around each mass, then applying deconvolution to reduce cross-talk caused by signal contributions from neighboring masses and isotopes in each spectrum (Müller et al., 2010). The script runtime on an 10-core processor system with 1Tb harddrive and 96 GB of RAM is about 2.5 hours for 24 hours of 10 Hz data. The recorded amount of raw Vocus data collected

each day in the native HDF5-file format was about 20GB.

**2.4 Eddy-Covariance fluxes**

The most direct method for measuring a chemical flux above a forest canopy is the eddy-covariance (EC) method. It requires simultaneous, fast (10 – 20 Hz) measurements of the compound concentration ($c$) and vertical wind velocity ($w$). The covariance between the time-dependent fluctuations of these variables gives the flux ($F_c$), with

$$F_c = \overline{w'c'} = \frac{1}{t_2 - t_1} \int_{t_1}^{t_2} w'(t)c'(t)\, dt, \tag{1}$$

Where the overbar denotes time-averaging from time $t_1$ to time $t_2$, $c' = c - \bar{c}$, and $w' = w - \overline{w}$. Further information regarding the EC approach can be found in the literature (e.g., Aubinet et al., 2012a). As a significant proportion of a turbulent flux in the atmospheric surface layer is carried by relatively small eddies, for the EC-flux method to accurately quantify the vertical exchange, the BVOC concentration and vertical wind speed must be measured by fast-response instrumentation.

Typically, many analyzers for greenhouse gas fluxes have response times of around 0.1 s. While the PTR analyzer's characteristic response time $\tau$ (i.e., time for $100 \times 1/e\,\% \approx 63.2\,\%$ of instrument signal to fully transition between two levels of concentration) is around 1 s, due to transit time of the BVOC product ions in the Vocus drift tube reaction chamber (Krechmer et al., 2018), the effect of this is relatively minimal for evaluating BVOC EC flux (high-frequency attenuation of total flux signal is about 1%), and can be accounted for by a transfer function correction factor (Striednig et al., 2020). This is

sufficient for fluxes measured above tall vegetation, such as forest canopies (Rantala et al., 2014).

The EC flux calculations were performed using the MATLAB-code *innFLUX* (Striednig et al., 2020). The analysis routines implemented in this code include a wind sector-dependent tilt correction, lag time determination, and calculation of several

quality tests. In particular, a tilt-correction was performed on the Metek sonic data to align the instrument's coordinate system with the mean wind streamlines (e.g., Wilczak et al., 2001). The time delay between the Metek sonic and Vocus BVOC signals

was determined by maximizing the correlation coefficient of the Vocus signals with the vertical wind component. For the final processing of the data points for the Vocus EC flux timeseries, 30-minute ensemble averages were selected.

## 2.5 Adsorption sampling and ATD-GC-MS analysis

For a gradient-flux approach targeting the speciated monoterpene fluxes, ambient air BVOC samples were collected over three-day periods, approximately once a month, from June to August, from two heights. These air samples were simultaneously collected at 37 and 60 m on the Norunda flux tower for 30 min periods, once an hour, starting at 9:30 am and concluded with the collection of the last sample-pair at 5:00 pm from the tower.

Air samples were collected using stainless steel thermal desorption (TD) tubes (10 cm in length and 1/4 inch in diameter) and filled with Tenax GR and Carbograph 5TD (Markes International Inc., USA). Ambient air was pumped through these TD tubes at 200 mL/min for 30 min (6 L total per sample) using flow-controlled sampling pumps (Pocket Pump Pro, SKC Ltd., Dorset UK). Two $MnO_2$-coated copper mesh filters (50 mm grid, type ETO341FC003, Ansyco, Karlsruhe, Germany) were placed inside a Teflon inlet head affixed in front of each sampling tube, to remove ozone from the ambient air. In previous

studies it was found that these $MnO_2$-coated mesh filters destroy about 80% of the ozone but leave α-pinene, β-pinene, limonene, $\Delta^3$-carene and other terpenoids intact (e.g., Bäck et al., 2012; Calogirou et al., 1996). TD samples were refrigerated and then analyzed within 30 days of collection. Calibration tubes were also prepared and analyzed to calibrate the subsequent GC-MS analysis (e.g., Bai et al., 2016).

TD tubes were thermally desorbed and cryo-focused using a Perkin Elmer TurboMatrix 650 automatic thermal desorber (ATD). This ATD was operated in splitless mode. Samples were then injected into a gas chromatograph-mass spectrometery system (GC-MS, Shimadzu QP2010 Plus, Shimadzu Corportation, Japan). For the gas chromatography portion of the ATD-GC-MS analysis, the BVOC gas samples were separated using a BPX5 capillary column (50m, I.D. 0.32mm, film thickness 1.0 μm, Trajan Scientific, Australia). The carrier gas was helium. The GC oven temperature program was as follows: start

temperature, 40C; hold time, 1 min; ramp 1, 6 C per min to 210 C; ramp 2, 10 C per min to 250 C; final hold time, 5 min. The GC-MS was run simultaneously in both SIM and SCAN modes (e.g., Duhl et al., 2013). This choice was to enable detection of both common and unanticipated compounds. For sample calibration, a pure standard solution containing isoprene, α-pinene, β-pinene, p-cymene, eucalyptol, limonene, 3-carene, linalool, α-humulene, β-caryophyllene, longifolene and myrcene (Merck KGaA, Darmstadt, Germany) was prepared in methanol. This standard solution was injected onto conditioned adsorbent

sampling tubes in a helium stream, with cartridges receiving different concentrations of the standard solution. These prepared TD tubes were then analyzed with the samples in order to provide a calibration curve (e.g., Noe et al., 2012).

## 2.6 Gradient-method fluxes

In the Surface Layer Gradient (SLG) method we obtain the turbulent flux by using the vertical gradient of the measured volume mixing ratios (i.e., concentrations $\bar{c}$) and a turbulent exchange coefficient $K$ in the manner analogous to molecular diffusion, with $F_C = -K \frac{\delta \bar{c}}{\delta z}$. The turbulent exchange coefficient $K$ can be obtained in several approaches, such as using results from another scalar quantity through the modified Bowen ratio approach or by application of Monin-Obukhov similarity theory (e.g., Rannik, 1998; Rantala et al., 2014). In the case of BVOC gradient sampling using TD tubes at two heights, we use the Monin-Obukhov similarity approach as in e.g., Fuentes et al. (1996) and Rinne et al. (2000b),

$$F_C = \frac{-ku_*[\bar{c}(z_1) - \bar{c}(z_2)]}{ln\left(\frac{z_2-d}{z_1-d}\right) + \psi_h\left(\frac{z_1-d}{L}\right) - \psi_h\left(\frac{z_2-d}{L}\right)}, \tag{2}$$

where $u_*$ is the friction velocity, d is the displacement height, $L$ is the Obukhov length, $\psi_h$ is the Monin-Obukhov stability function for heat, $k$ is the von Kármán constant, and $\bar{c}(z_1)$ and $\bar{c}(z_2)$ are the BVOC concentrations at heights $z_1$ and $z_2$, respectively. The values of the integrated stability functions $\psi_h$ were calculated using the Businger-Dyer equations (Dyer, 1974), with

$$\begin{aligned} \psi_h &= 2\,ln\left(\frac{1+Y}{2}\right), (for\ \zeta < 0), \\ \psi_h &= -\beta_h\zeta, (for\ \zeta > 0) \end{aligned} \tag{3}$$

where $\zeta = (z-d)/L$, and coeffcients $Y = (1 - 12\zeta)^{1/2}$ and $\beta_h = 7.8$ were selected based on those used in previous investigations (Businger et al., 1971; Dyer, 1974; Rannik, 1998; Rantala et al., 2014; Rinne et al., 2000a).

In the atmosphere directly above rough surfaces, such as forest canopy, flux-gradient laws tend to break down (Cellier and Brunet, 1992; Fazu and Schwerdtfeger, 1989; Garratt, 1980; Högström et al., 1989; Mölder et al., 1999; Simpson et al., 1998). The layer in which these laws are not directly applicable is called the roughness sublayer (RSL). In the RSL, the eddy diffusivities are increased, and consequently, vertical gradients are decreased compared to their nondimensionalized gradient form, which for a scalar concentration C is expressed by $\Phi_C(\zeta) = \frac{k(z-d)}{c_*}\frac{\partial \bar{c}}{\partial z}$, where $c_*$ is the scalar flux concentration (e.g., Rannik 1998). The decrease in gradients due to the influence of the RSL layer can be quantified and corrected for by using the nondimensional factor γ, expressed as

$$\gamma = \frac{\Phi_S}{\Phi}, \tag{4}$$

where $\Phi_S$ is the dimensionless gradient of a scalar according to the Monin–Obukhov similarity theory (i.e., Eq. 4) and $\Phi$ is the dimensionless gradient according to the measurements. The observed $\gamma$-coefficients above a forest canopy at the heights our TD samples were collected typically vary from unity to 3 depending on the measurement height and type of the forest (e.g., Simpson et al., 1998).

From $\gamma$, it is possible to calculate a mean enhancement factor $\Gamma$ by integrating the $\gamma$-coefficient from the measurement height $z_1$ to $z_2$ (e.g., Rinne et al., 2000b), such that in general we have

$$\Gamma(z_1, z_2) = \frac{1}{z_2 - z_1}\left[\int_{z_1}^{z_2} \gamma(z)dz\right]. \tag{5}$$

Details of the $\gamma$-coefficient profile analysis for the RSL above the Norunda canopy, as well as determination of displacement height $d$ (24 m), are included in the Appendix. Based on this analysis, we have found a $\Gamma$-coefficient value of $1.36 \pm 0.09$ for TD gradient flux calculation.

**2.7 Gradient-method error analysis**

For the two-point SLG gradient method, as alluded to by the form of $F_C = -K\frac{\delta\bar{c}}{\delta z}$ and Eq. 2, the sources of uncertainty can be divided into those from the gradient (i.e., measured concentration difference $\bar{c}(z_1) - \bar{c}(z_2)$) and those from to the turbulent exchange coefficient $K$. A detailed evaluation of SLG-related uncertainties for BVOC flux measurements is presented in Rinne et al. (2000).

The adsorption sampling and analysis of the BVOCs represents the largest single source of uncertainty in the flux calculation. This is due to the relatively small difference in concentrations between sampling heights as compared to the uncertainty of the concentration measurements themselves. The two error sources which can be evaluated for the chemical gradient measurements are the sampling uncertainty and the analysis uncertainty from the ATD-GC-MS (e.g., Kajos et al., 2015). Measurement results from the ATD-GC-MS include values for peak area mean and standard deviation, as well as signal-to-noise ratio, which were used in the uncertainty analysis. Sampling uncertainty during field measurements includes the sampling pump flow rate (typically $\pm 5\%$ of set flow rate), whereas sources of uncertainty in the analysis include the preconditioned tube background, as well as the ATD-GC-MS instrumental uncertainty and standard calibration uncertainty. From the combined (in quadrature) uncertainties of TD sampling and laboratory analysis, a total estimated uncertainty of $\pm 15\%$ is assumed for each monoterpene compound. The total uncertainty of the measured concentration difference, $\bar{c}(z_1) - \bar{c}(z_2)$, is then determined by summing the uncertainties of $\bar{c}(z_1)$ and $\bar{c}(z_2)$ in quadrature.

In addition to these concentration-related uncertainties, the random and systematic uncertainties associated with the turbulent exchange were also considered. Following the approach of Rinne et al. (2000), the principal uncertainties of the turbulent exchange coefficient can be further divided into those originating from the Norunda tower flux measurements used in

calculating $K$ and those arising from the parametrization of $K$. For the former, uncertainties in $K$ are dominated by measurement noise and sampling error in the EC-derived friction velocity and buoyancy flux, contributing an estimated random error of $\pm20\%$. For the latter, systematic uncertainties in $K$ primarily arise from the use of universal flux–gradient relationships. While consensus exists on their functional form, a range of values for the empirical constants used to parameterize these relationships has been reported in the literature (e.g., Businger et al., 1971; Dyer, 1974; Wieringa, 1980; Högström, 1988; Oncley et al., 1996). In practice, alongside the von Kármán constant, the constants used in parameterizing the Businger–Dyer relationships are not determined independently from each other and hence, in principle, should be treated as a single parameter set. Variability among reported parameter sets produces up to 25% systematic uncertainty in calculated estimates of K. Evaluated directly from Norunda station data (e.g., Rantala et al. 2014), the zero-plane displacement height d was treated as an independent parameter and contributed an estimated systematic error of $\pm10\%$. The final uncertainty of the SLG flux is then assessed by applying the standard propagation of error method for summing up these four key uncertainties for each SLG flux estimate.

## 2.8 Correction for chemical degradation

One of the common assumptions of any surface layer flux measurement technique is the constancy of the vertical flux between the surface and the measurement level. In the case of reactive trace gases, such as VOCs, this assumption is not strictly valid, and the invalidity can cause systematic errors if interpreting the measured fluxes as surface exchange rates. This systematic error can be quite substantial for compounds with high Damköhler number ($Da$), i.e. the ratio of time-scale of the turbulent mixing to that of the chemical reactions, such as certain sesquiterpenes (Rinne et al., 2012). The ratio of flux to the surface exchange (R=F/E) depends on the chemical lifetime ($\tau_c$) of the compound in question and it's turbulent mixing time-scale ($\tau_t$). The chemical lifetime of BVOCs typically depends on $O_3$, OH and $NO_3$ mixing ratios and reactivities of the VOC in question against these reactants. The turbulent mixing time-scale depends on friction velocity ($u_*$) and measurement height (z). Thus, we can estimate the surface exchange rate (SER) of a compound if we can estimate the reactant levels and friction velocity by

$$E = \frac{F}{R(Da(\tau_t/\tau_c),z/h)} \tag{6}$$

and using the look-up tables by Rinne et al. (2012) to estimate the R. Furthermore, the R varies considerably depending on the emission height, i.e. between canopy and soil surface emission. For example, for isoprene, $Da$ ranged from 0.0003 to 0.014. For MT (based on the reactivities of α-pinene), $Da$ ranged from 0.005 to 0.037, and for SQT (based on the reactivities β−caryophyllene) between 0.3 to 0.9.

The effect of chemical degradation on isoprene, MT, and SQT surface exchange rates was explored using the modeling work of Rinne et al. (2012), which made use of a stochastic Lagrangian transport chemistry model, and by parameterizing the diurnal cycle of the reaction rates for ozone, OH and $NO_3$, to estimate the surface exchange rates for the 2020 Norunda campaign's terpenoid EC flux dataset. The OH, $O_3$, and $NO_3$ reaction rate constants of isoprene, MT, and SQT for the chemical degradation analysis are from those reported by Atkinson (1997) and Shu and Atkinson (1995). In section 3.3.1, the reaction rate

coefficients of α-pinene and β-caryophyllene were implemented to assess surface exchange rates from the measured fluxes of total MT and total SQT, respectively, obtained using Vocus PTR-ToF-MS. Both compounds are common and frequently dominant examples of their terpenoid classes in the emissions of Norunda and similar boreal forests (e.g., Hakola et al., 2006; Hellén et al., 2018; Rinne et al., 2009; Rinne et al., 2012; Wang et al., 2018)." it is important to note, however, that the dominance of a particular compound in total emissions, such as β-caryophyllene among SQTs, might not always be the case, particularly for non-constitutive (stressed) emissions, such as from insect herbivory (e.g., Wang et al., 2017). Care must be taken when inferring total surface exchange rates that underlying assumptions regarding the relative mixture of emitted compounds are correct. A full description of the surface exchange rate calculations, as well as the influence of relative speciation on total exchange rate estimate uncertainties, can be found in the appendix.

## 2.9 Emission algorithm fitting

To estimate the mixed contribution of *de novo* biosynthesis and storage pool emission to terpene ecosystem-scale emissions, a hybrid emission algorithm (Taipale et al., 2011; Ghirardo et al., 2010) was fitted to emission measurements. This hybrid algorithm was developed under the hypothesis that the two origins of emission can combined as $E = E_{synth} + E_{pool}$, where $E_{synth}$ represents emission originating directly from biosynthesis and $E_{pool}$ represents emission from specialized storage structures, such as resin ducts . This hybrid algorithm formulation takes the form

$$E = E_o[f_{denovo}C_T C_L + (1 - f_{denovo})\gamma], \tag{7}$$

where $E_o = E_{o,denovo} + E_{o,pool}$ is the total emission potential, $f_{denovo} = E_{o,denovo}/E_o$ is the ratio of the *de novo* emission potential to the total emission potential, $C_T$ and $C_L$ are the synthesis activity factors for temperature and light, respectively (Guenther 1997), and $\gamma$ is the temperature activity factor ($\exp[\beta(T - 30°C)]$) for the traditional pool emission algorithm, where β is an empirical constant (°C$^{-1}$) and T is the canopy temperature (°C).

To determine $E_o$, β, and $f_{denovo}$, the fitting of Eq. 7 to the campaign data was performed using nonlinear regression. As isoprene is widely understood to have no storage source for emission (Guenther 1993, 1995, 1997), fitting for isoprene emission was investigated by setting $f_{denovo}$ to 1 (i.e., pure *de novo* synthesis emission). Meanwhile, pool emission for MT and SQT was investigated by setting $f_{denovo}$ to 0. In the case of investigating the fraction of MT and SQT emissions deriving as a mix of *de novo* synthesis and pool emission, $f_{denovo}$ was allowed to vary as a fitting parameter.

# 3 Results

## 3.1 Meteorological & other conditions during campaign

Temperatures during the campaign varied between 13 and 22 °C. The typical prevailing wind direction was between South and Northwest. Light precipitation occurred on manual sampling day June 9. The meteorological conditions during the campaign EC-measurements are displayed in Fig. 2. Typical peak daytime photosynthetic photon flux density (PPFD) varied from 700 to 1500 µmol m$^{-2}$ s$^{-1}$. Conditions during days when TD sampling was performed (June 8-10, July 22-24, and August 16-18) had consistent temperature (mean $17.4 \pm 3.7$ °C) and PPFD (mean $901 \pm 319$ µmol m$^{-2}$ s$^{-1}$) conditions. Ozone monitoring was available throughout the campaign from the nearby Norunda-stenen station, via a Model E400 Teledyne ozone analyzer, located 1.4 km east of the Norunda tower. One interruption to Vocus sampling occurred on August 12-13 due to an electrical failure in the instrument shelter. A summary of the campaign timeseries is presented in Fig. 2, in which the campaign observations of the station water vapor, vapor pressure deficit (VPD), and dewpoint temperature are also included.

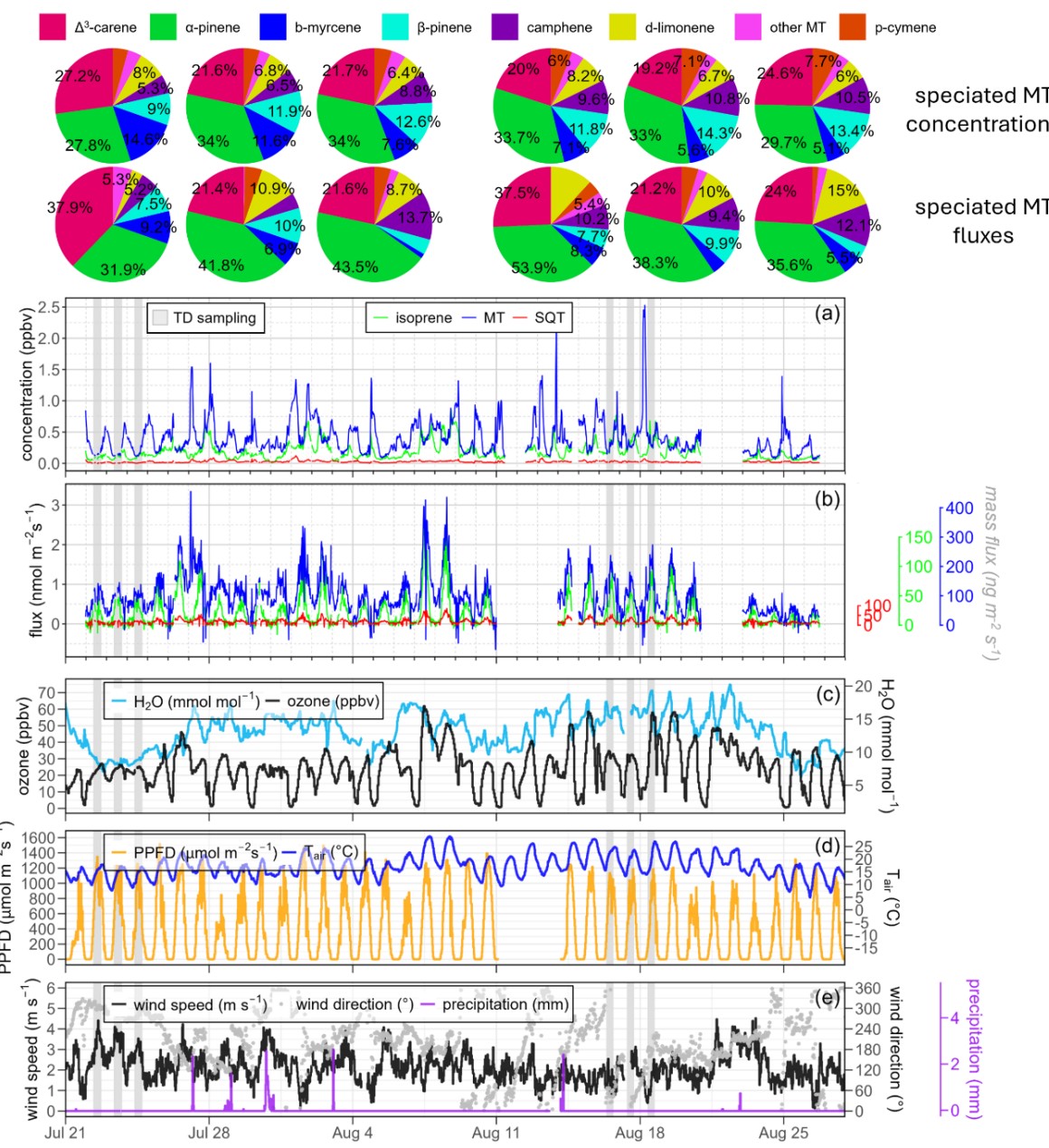

**Figure 2: The 30-min BVOC concentrations (ppbv) and fluxes (nmol m⁻² s⁻¹) sampled at the 35 m BVOC inlet at ICOS Norunda, as well as related meteorological measurements. Shaded areas depict manual TD BVOC sampling periods at 37 and 60 m on the Norunda flux tower. The pie charts indicate the relative speciation of MT compound (top row) concentrations at 37 m and (bottom row) fluxes (via SLG flux method) as determined from the TD BVOC samples (percentages shown are >5%). (a) MT, isoprene, and SQT concentrations (ppbv) and (b) eddy-covariance derived fluxes (nmol m⁻² s⁻¹) from the Vocus PTR-ToF-MS. The equivalent mass-flux (ng m⁻² s⁻¹) for isoprene, MT, and SQT is depicted along the righthand y-axis. (c) Ozone concentration (ppbv)**

and water vapor (mmol mol[-1]). (d) PPFD (µmol m[-2] s[-1]; at height 55 m) and air temperature (°C; at 37 m). (e) wind speed (m s[-1]), wind direction (°), and precipitation (mm). Set of displayed measurements span from 21 July 2020 to 27 August 2020.

## 3.2 Gradient sampling conditions during campaign

### 3.2.1 Sampling footprint comparison

To accurately interpret BVOC fluxes derived from concentration gradient measurements, it is important to assess the gradient-flux method footprint for the two TD sampling levels (37 m and 60 m). A flux footprint analysis at their geometric-mean height

(47.1 m), as suggested by Horst (1999) for two-height gradient-profile flux estimates, was conducted for each daily period corresponding to TD tube BVOC gradient sampling on the Norunda flux tower. Each footprint was calculated using the flux footprint model developed by Kljun et al. (2015). The Flux Footprint Prediction (FFP) is a two-dimensional parameterization for the flux footprint based on a scaling approach to its crosswind distribution (e.g., Kljun et al., 2004; Kljun et al., 2015). It was found that the footprints, particularly for ca. 85[th] percentile and below footprint contours (depicted in Fig. 3), in general

compared well with each other in terms of the forest area and composition covered. Since the geometric-mean height for the SLG estimates is above the Vocus inlet height (35 m), the total extent of the estimated SLG footprints tended to be slightly larger than the EC flux footprint.

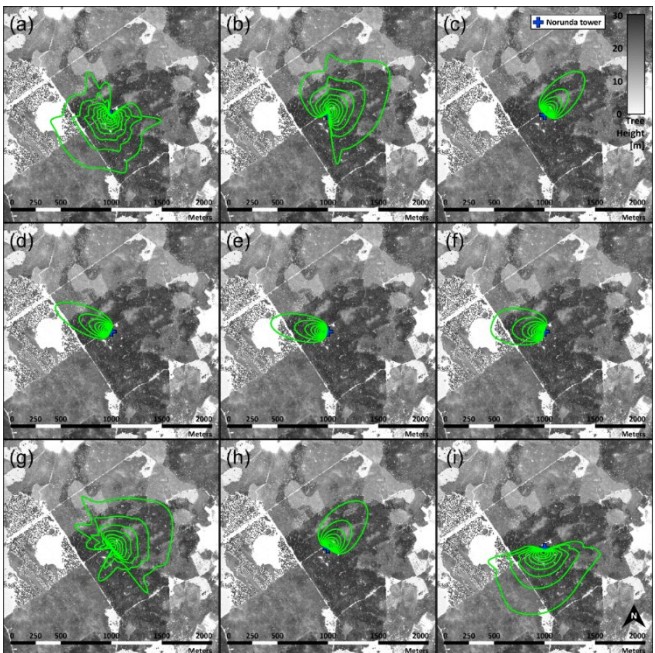

**Figure 3: Daytime (from 9:00 CEST to 17:00 CEST) average footprint estimates for SLG-derived fluxes using the two TD BVOC**
**sampling heights (37 and 60 m) on the Norunda flux tower. Footprint contour lines (green) are shown in 10% increments from 10% to 90%. Displayed footprints assessed at geometric-mean height (47.1 m) of TD sampling levels, following Horst (1999) for footprint estimation of SLG-method surface fluxes under unstable atmospheric stratification above-canopy (see Fig. 5f). The panels show these footprints for (a-c) June 8, 9, and 10, (d-f) July 22, 23, and 24, and (g-f) August 16, 17, and 18, respectively. Footprint con**

During the campaign TD measurements, approximately 90 % of the flux measured by the Vocus tower inlet at the 35 m level originated within 350 m of the tower itself. For comparison, at 47.1 m, approximately 90 % of the observations originated from within 420 m of the tower.

### 3.2.2 Vertical micrometeorological conditions

An important prerequisite for the gradient flux method is that vertical fluxes remain constant within the observation layer. To validate this constant-flux assumption, we compared the friction velocities, heat fluxes, and roughness lengths measured at both the 37 m and 60 m measurement levels.

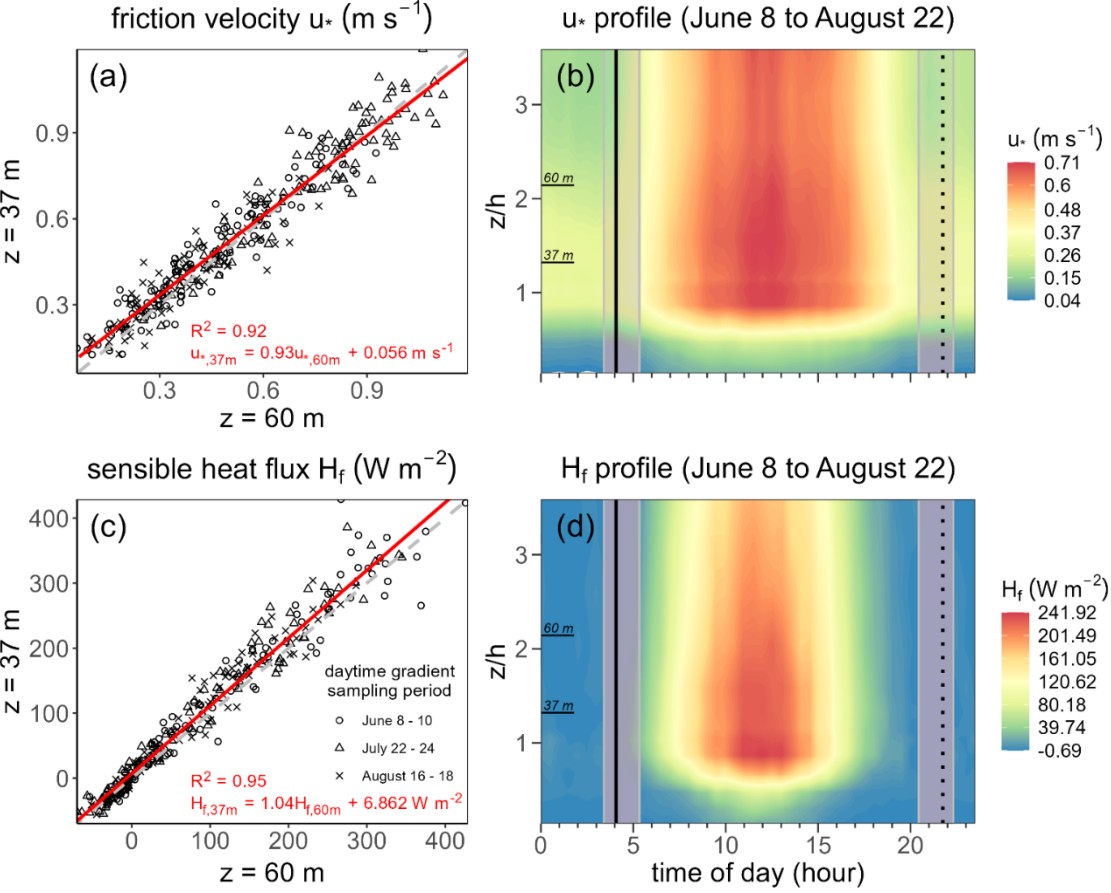

**Figure 4: Comparison between friction velocity $u_*$ (m s$^{-1}$) and sensible heat flux $H_f$ (W m$^{-2}$) measured at two heights above the forest canopy. (a; c) A best-fit comparison between 37 m and 60 m for (a) friction velocity and (c) sensible heat flux during the 3-day**
**sampling periods in June, July and August. The dashed line indicates a 1:1 relation, while the solid red line indicates the best fit linear regression. (b; d) Diurnal mean contour profiles for the 2020 Norunda campaign of (b) friction velocity $u_*$ and (d) $H_f$ with respect to the normalized height z/h (canopy height h=28 m) in and above the Norunda forest. The relative heights of the 37 and 60**

m sampling levels on the normalized height scale are indicated on the y-axis. In (b) and (d) panels, mean daily sunrise (solid vertical line) and sunset (dotted vertical line) are indicated. Shaded region indicates the range of sunrise and sunset times during the June 9th to August 22th campaign period.

Measured sensible heat and momentum fluxes were similar at both measurement heights (see Fig. 4), with the mean linear fit of friction velocity and sensible heat flux between the 37 m and 60 m heights being $u_{*,37\,m} = 0.93\,u_{*,37\,m} + 0.056 \text{ m s}^{-1}$ and $H_{f,37m} = 1.04\,H_{f,60m} + 6.9 \text{ W m}^{-2}$, respectively. Table A1 (see Appendix) provides a summary of these friction velocity and sensible heat flux ratios under different stability conditions and wind directions. Atmospheric instability above canopy (i.e., Obukhov length $L^{-1} < -1000$; see Fig. 5f) prevailed during the times of daytime TD sampling. Stable atmospheric conditions were generally observed at night, while near-neutral conditions were typically observed during the transition in stability following sunrise and preceding sunset.  It was found that, while near-neutral and stable conditions could lead to relatively large deviations in these ratios, for unstable atmospheric conditions the ratios were close to 1.

## 3.3 Concentrations and fluxes of terpenes and other VOCs

During the campaign, the mean daytime isoprene concentration measured by the Vocus PTR-Tof-MS was 250 pptv. As shown in Figure 2, the maximum concentration values occurred during daylight hours, at a time when temperature and PPFD were high (>20°C and >1000 μmol m$^{-2}$ s$^{-1}$) as well, with concentrations falling to daily lows towards the evening. Little to no isoprene flux was observed at night. Peak total MT concentrations (1 – 1.4 ppbv) were typically significantly higher than isoprene concentrations. Unlike isoprene, peak total MT concentration typically occurred at night during stable atmospheric conditions in the canopy similarly to the observations by Petersen et al. (2023).

Such diurnal cycles in concentration were regularly observed throughout the campaign. Fig. 5 shows the diurnal variation in isoprene, total MT, and total SQT from the Vocus PTR-ToF-MS measurements collected (July 21 to August 27) during the 2020 Norunda field campaign.

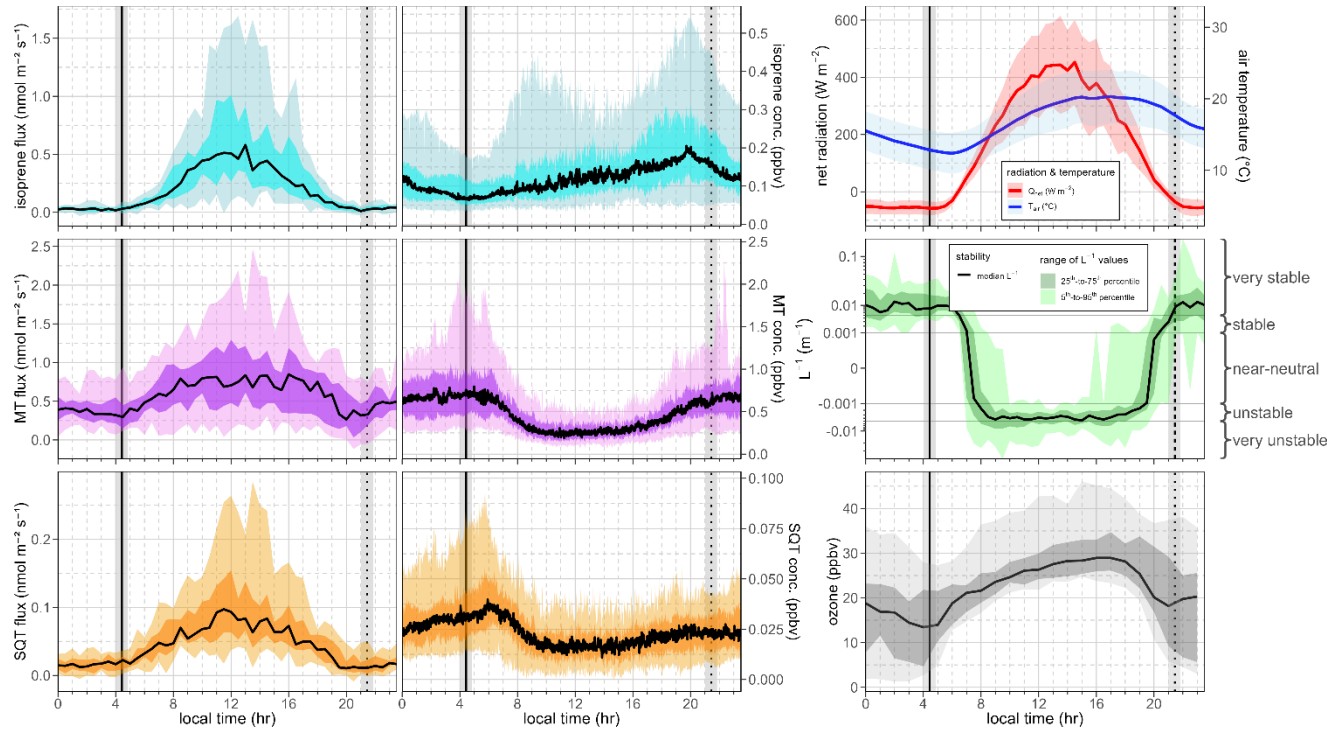

**Figure 5: Diurnal range of fluxes and concentrations for (a,b) isoprene, (d,e) total MT, and (g, h) total SQT, as measured from the tower BVOC inlet using Vocus PTR-ToF-MS during the Norunda 2020 BVOC field campaign. Diurnal fluxes shown represent 30-min EC ensemble averages and diurnal concentrations represent 1-min averaged timeseries data. Shaded regions in panels indicate the 5th-to-95th and the 25th-to-75th quantile range, as well as the mean (solid line). (c) Net radiation (red) and canopy temperature (blue). Shaded regions for net radiation and temperature indicate one standard deviation. (f) Plot of the inverse of the Obukhov length ($L^{-1}$), indicating atmospheric stability above the canopy (measured at 36 m). (i) local diurnal ozone concentration (ppbv). In all the panels, mean sunrise and sunset (solid and dotted vertical lines, respectively) for the period of Vocus deployment (July 21 to August 27) are indicated. Vertical bars (grey) indicate the range of sunrise and sunset times during this period.**

Isoprene had its highest concentrations above forest canopy during the day, with peaks typically in the morning between 7:00-11:00 CEST and more strongly between 16:00 CEST and sunset, whereas MT and SQT concentrations typically peaked at night. Isoprene and terpene fluxes, meanwhile, generally all peaked around noon. This diurnal concentration behavior by isoprene and total SQT, as was observed for total MT, was due to interplay of emission dynamics and the changes in atmospheric stability in the surface boundary layer. For isoprene, this was due to the light-dependent nature of emissions (i.e., *de novo* synthesis emissions (e.g., Guenther et al., 1995; Guenther et al., 1993)), with emissions effectively shut down with the cessation of photosynthesis activity at night. On the other hand, the MT and SQT emissions of evergreen boreal tree species like Scots pine and Norway Spruce still have considerable night-time emission from resin ducts and other tissue structures that

are temperature-only-dependent (i.e., storage emission, rather than wholly *de novo* emission (e.g., Ghirardo et al., 2010; Guenther et al., 1995; Guenther et al., 1993; Tingey et al., 1980)). It should also be noted that many SQTs have a short (~0.1 – 10 s) chemical lifetime relative to MT, and that this is reflected in the observed concentrations measured by the Vocus, as a substantial fraction is expected to react with atmospheric radicals or other compounds before reaching the measurement height (e.g., Atkinson and Arey, 2003; Rinne et al., 2012). This diurnal behavior highlights the interplay between constitutive (non-stress induced) emissions, driven by environmental conditions (principally, photosynthetic light and temperature), and the atmospheric stability within the forest canopy. Despite lower emission rates of terpenes at night, as observed from their EC fluxes (see Fig. 5d&g), the stable nighttime atmosphere, as indicated by the Obukhov length (see Fig. 5f), causes a buildup in their concentrations at night. It was observed that terpene concentrations are typically greatest at or just following sunrise. Similar diurnal behavior was observed in other VOC compounds measured by the Vocus, such as acetone, acetaldehyde, and toluene, among others, and is also consistent with previous BVOC PTR-MS field campaigns at ICOS Norunda (Petersen et al., 2023). This diurnal terpene behavior is also consistent with observations at other boreal sites (e.g., Borsdorf et al., 2023; Hakola et al., 2012; Hellén et al., 2018).

### 3.3.1 Surface exchange rates

To quantify the actual surface exchange rate of reactive VOCs, the effect of chemical degradation on the surface exchange rate (SER) of isoprene, MT and SQT was estimated. For ecosystem-atmosphere exchange, whereas above-canopy fluxes quantify the cumulative effect of within-canopy processes, including chemical degradation, the surface exchange rate characterizes the net emission and deposition occurring at the ecosystem's soil and vegetation surfaces in the absence of this chemical sink. The impact of chemical degradation on the ratio of measured flux to surface exchange rate, R=F/E, is shown in Fig. 6. Following the modeling work of Rinne et al. (2012) for a similar boreal forest, the F/E ratios for MT are based on the reaction rate constants of $\alpha$-pinene and for SQT on the reaction rate constants of $\beta$-caryophyllene. The impact of the chemical degradation rate of isoprene and MT, relative to the observed fluxes, was found to be minimal. The influence of chemical degradation on isoprene (F/E = 0.981 ± 0.013 at night and 0.957 ± 0.013 during day) was within the EC flux measurement uncertainty. The main influence of chemical degradation for total MT was found to be at night and, in terms of absolute emission increases, during peak flux periods around noon. For total MT, a daytime average of F/E = 0.957 ± 0.013 due to chemical degradation was calculated for daytime (within the uncertainty of EC flux measurements), while the nighttime loss was estimated at an average F/E = 0.90 ± 0.03.

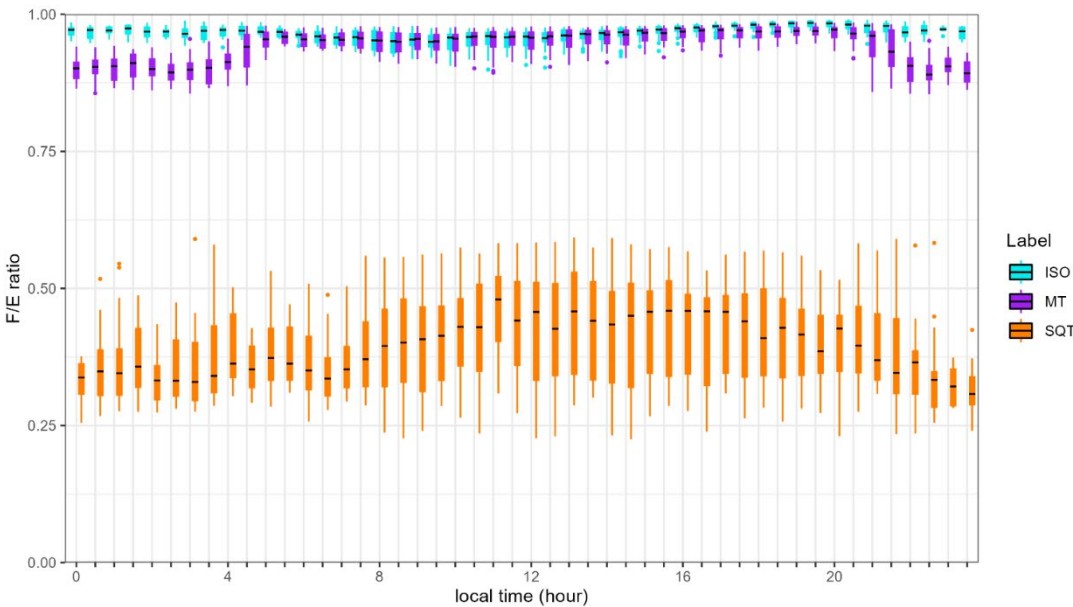

**Figure 6: Boxplot of mean diurnal F/E ratios for chemical degradation estimates of isoprene (cyan), MT (purple), and SQT (orange) for the 2020 Norunda BVOC campaign. Black dash of whisker plot indicates the median value.**

In contrast, the measured SQT flux was significantly impacted by chemical degradation (see Fig 7). For the full campaign, the estimated SQT nighttime F/E ratio is typically ca. 0.35 (varying from 0.31 to 0.41), while the daytime ratio is ca. 0.41 (varying from 0.37 to 0.47). The mean diurnal measured fluxes and inferred surface exchange rates of isoprene, MT and SQT, for their respective OH, $O_3$, and $NO_3$ reaction rate constants (table A3 in the Appendix), are shown in Figure 7. As a consequence, for the diurnal average over the full campaign, peak SQT nighttime emission rates are typically ca. 240 % to 310 % (mean ca. 290

510    %) times greater, and SQT daytime emissions ca. 240 % to 290 % (mean ca. 260 %) times greater than would otherwise be inferred solely from EC flux measurements if the impact of chemical degradation on SQT exchange and subsequent SQT flux observations were neglected.

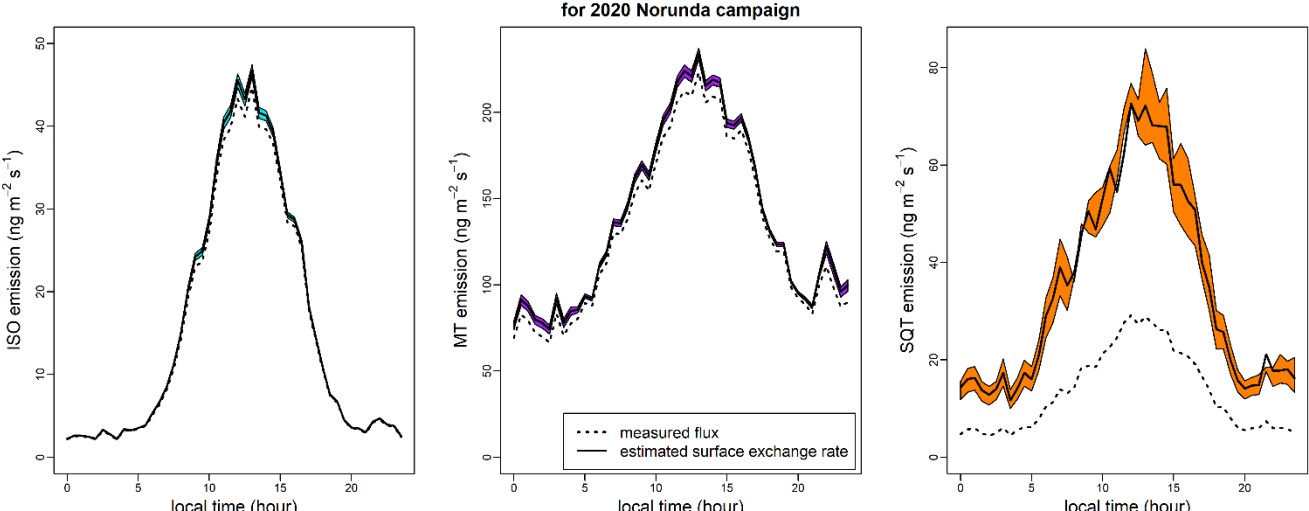

**Figure 7: Mean diurnal surface exchange rates for isoprene, MT, and SQT during the 2020 Norunda campaign. Shaded region indicates the range of uncertainty for the modeled ozone, OH, and NO₃ reaction rate coefficients. Dashed black line indicates the corresponding measured flux.**

The impact of chemical degradation on the F/E ratio between the TD sampling heights was also investigated. This was done to gauge an underpinning assumption of the SLG method: that there is no significant chemical sink between the sampling heights used to determine the above-canopy gradient. This was performed by evaluating, instead of $R = F/E$, the value of $R_2/R_1$ = $F_2/F_1$, where $R_1$ and $R_2$ are the flux-to-emission ratios for the sampling heights $z_1 = 37$ m and $z_2 = 60$ m, respectively. This provides a measure for the percent of flux lost between the lower and upper heights for the gradient measurement. For MT, typically just 1.7 ($\pm0.5$) % flux loss between sampling heights, with $R_{2,MT}/R_{1,MT} = 98.3$ (varying from 97.8 to 98.8). This is less than the F/E ratio previously found for the total MT EC flux measurements. In contrast, for SQT, there was typically between 33 and 43 % loss in SQT flux between these heights, with $R_{2,SQT}/R_{1,SQT} = 61.8$ (varying from 57.1 to 66.6). As expected, this demonstrates significant SQT chemical loss between the two intervening heights, and that the SLG method would be inapplicable to SQT fluxes without significant modification to account for chemical degradation.

**3.3.2 Comparing measured terpenoid emissions with emission algorithms**

The presented emission algorithm regression results (specifically, for the terpenes) are presented in Fig. 8. As the estimated isoprene surface exchange rates were within the range of uncertainty of the measured fluxes, only the measured isoprene fluxes were used for the regression analysis. For isoprene, we used the fixed $\beta = 0.09$ ℃$^{-1}$ of Guenther (1997), and the emission algorithm was set to pure *de novo* synthesis emission by setting $f_{denovo}$ to 1. The fitted isoprene $E_o$ was found to be 85.0 ($\pm$ 1.5) ng m$^{-2}$ s$^{-1}$.

For MT emissions based on the temperature-dependent pool emission algorithm (Guenther et al. 1993; Guenther et al. 2012), by setting setting $f_{denovo}$ to 0, the fitted MT standardized emission $E_o$ was found to be 386 ($\pm$ 5) ng m$^{-2}$ s$^{-1}$ for $\beta$ = 0.1 $^{\circ}$C$^{-1}$ (Guenther et al., 2012). Allowing $\beta$ to vary as a regression coefficient for the pool algorithm as well yields 370 ($\pm$9) ng m$^{-2}$ s$^{-1}$ and $\beta$ = 0.094 ($\pm$ 0.003) $^{\circ}$C$^{-1}$. Applying the full hybrid MT emission algorithm, combining de novo synthesis and pool storage emission (e.g., Taipale et al., 2011), with the previously fitted $\beta$ = 0.094 $^{\circ}$C$^{-1}$, yields $E_o$ = 374 ($\pm$ 7) ng m$^{-2}$ s$^{-1}$, and the fraction of MT emissions originating from de novo synthesis as $f_{denovo}$ = 26 ($\pm$ 4) %. When using the MT surface exchange rates instead of the observed EC fluxes at 37 m, fitting the hybrid algorithm then yields $E_o$ = 393 ($\pm$ 8) ng m$^{-2}$ s$^{-1}$ and $f_{denovo}$ = 24 ($\pm$ 4) %.

As the SQT emission rate is significantly underrepresented by the measured flux due to chemical degradation, we have fitted the hybrid emission algorithm to the estimated SQT surface exchange rates for the campaign. For fitting from SQT surface exchange rates, based on the temperature-dependent pool emission algorithm of Guenther (Guenther et al. 1993; Guenther et al. 2012), the fitted standardized SQT emission $E_o$ was found to be 171 ($\pm$ 3) ng m$^{-2}$ s$^{-1}$ for $\beta$ = 0.17 $^{\circ}$C$^{-1}$ (Guenther et al., 2012). Allowing $\beta$ to vary as a regression coefficient for the pool algorithm as well yields 160 ($\pm$ 5) ng m$^{-2}$ s$^{-1}$ and $\beta$ = 0.156 ($\pm$ 0.004) $^{\circ}$C$^{-1}$. Applying the hybrid emission algorithm, combining de novo synthesis and pool storage emission (e.g., Taipale et al., 2011), with the previously fitted $\beta$ = 0.156 $^{\circ}$C$^{-1}$, yields $E_o$ = 156 ($\pm$ 3) ng m$^{-2}$ s$^{-1}$, and the fraction of SQT emissions originating from *de novo* synthesis as $f_{denovo}$ = 45 ($\pm$ 8) %. Fitting the hybrid algorithm using the estimated surface exchange rates for SQT yielded better fits than simply applying the measured SQT flux data without the corresponding correction for chemical degradation. From the surface exchange rates, the hybrid algorithm estimates for $f_{denovo}$ for both MT and SQT were found to be quite similar (38 ($\pm$ 8) % and 31 ($\pm$ 7) %, respectively).

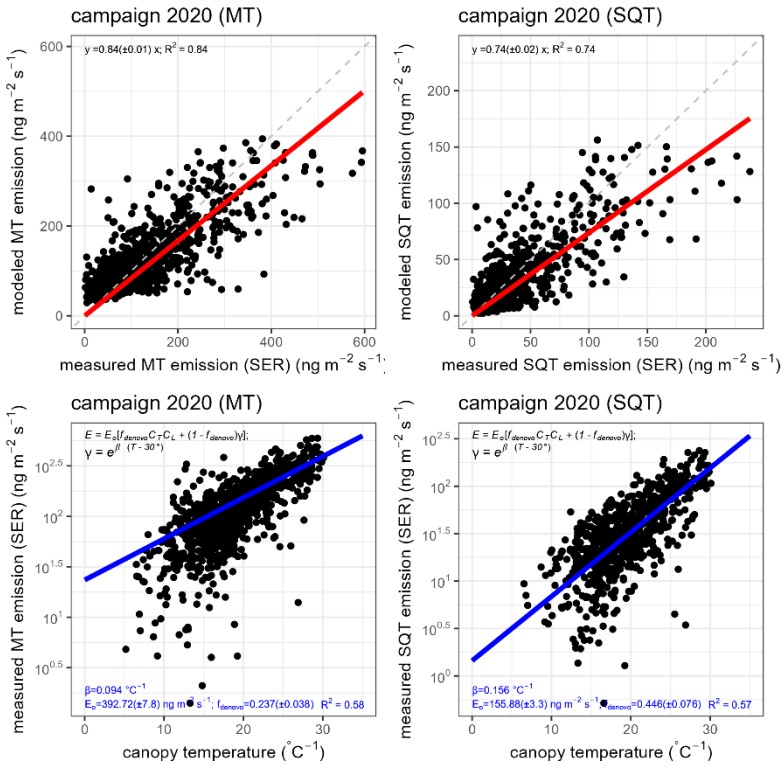

**Figure 8: Hybrid emission algorithm comparison to terpene measurements. (a,b) Comparison of measured vs modeled emission (using the fitted coefficients) for (a) MT and (b) SQT surface exchange rates. The red line indicates the linear regression best-fit line for the data points, and dashed grey line indicates the one-to-one line. (c,d) Plots of the canopy temperature vs (c) MT emission and (d) SQT emission. Blue line shows the best-fit regression line.**

### 3.3.3 Other VOCs

Acetaldehyde (m/z+ = 45.034) exhibited a mean daily concentration of 0.7 ppbv. Concentrations of toluene (m/z+ = 93.07) were generally low during daytime (~12 pptv) and increased during nighttime (~30 pptv), This behavior by toluene is consistent with the build-up of anthropogenic background emissions during night in the shallow nocturnal boundary layer (Karl et al., 2004). Similar behavior was found for the mass peak at m/z+ = 95.049 (i.e., phenol), which had a concentration minimum during daytime (~9 pptv) and maximum during nighttime (~40 pptv). Acetic acid (m/z+ = 61.028) was typically lowest after sunrise (~10 pptv), gradually increasing throughout the day and peaking before sunset (~33 pptv), then declining overnight. The exception to this trend occurred when high nighttime canopy concentrations coincided with similar peaks in acetone and acetaldehyde. The dirurnal signal for m/z+ = 41.039 and m/z+ = 103.112, representing the PTR-protonated hexanol fragment and the hexanol parent ion, respectively, followed a similar pattern to acetone. The minimum in hexanol concentration (~50

pptv) typically occurred in the morning following sunrise and peaked after sunset (~130 pptv). Methyl vinyl ketone and methacrolein (MVK+MACR, m/z+ = 71.049), two important intermediate products from the photochemical oxidation of isoprene, averaged 7 pptv daily.

.

### 3.4.1 Speciated MT concentrations and fluxes

During the campaign, α-pinene, $\Delta^3$-carene, β-pinene, camphene, myrcene, and limonene were detected in the ATD-GC–MS analysis. The most abundant monoterpene species throughout the campaign was α-pinene, followed by $\Delta^3$-carene (see Fig. 2 & 9). Limonene concentrations contributed about 5-10% of total monoterpene concentrations. An overview of the monthly sampling-period mean concentrations and SLG method-derived fluxes of the speciated MT compounds, observed via thermal desorption sampling at 35 m above the forest canopy, during the 3-day monthly sampling periods in June, July, and August is shown in Fig. 9. A daily mean evaluation can be found in the Appendix. During all sampling periods, α-pinene was the most prevalent MT compound present, fairly constant and typically representing between ca. 28 to 34 % of the total MT concentration. The second most prevalent MT compound was $\Delta^3$-carene. From June to August, fraction of $\Delta^3$-carene among the MT compounds decreases by ca. 8 %, from monthly sampling-period averages of ca. 30 % in June, to ca. 24% in July and ca. 22 % in August. During the June measurements, the typical concentration abundance at 37 m above the forest canopy among the MT species were 32 (± 4) % α-pinene, 30 (± 4) % $\Delta^3$-carene, 7.6(±0.8) % myrcene, 5(±0.6) % limonene, and 10(±1.1) % β-pinene, 7.2(±0.6) % camphene, and 5.6(±0.5) % cymene. In July, the typical abundances were 31 (±3) % α-pinene, 24 (±6) % $\Delta^3$-carene, 12(±2) % myrcene, 7(±2) % limonene, and 10.8(±0.9) % β-pinene, 4.5(±0.7) % camphene, and 4.5(±0.7) % cymene. In August, the typical abundances were 32(±3) % α-pinene, 22(±2) % $\Delta^3$-carene, 5.9(±0.4) % myrcene, 6.9(±0.6) % limonene, 13(±2) % β-pinene, 10.3(±0.8) % camphene, and 7(±0.5) % cymene.

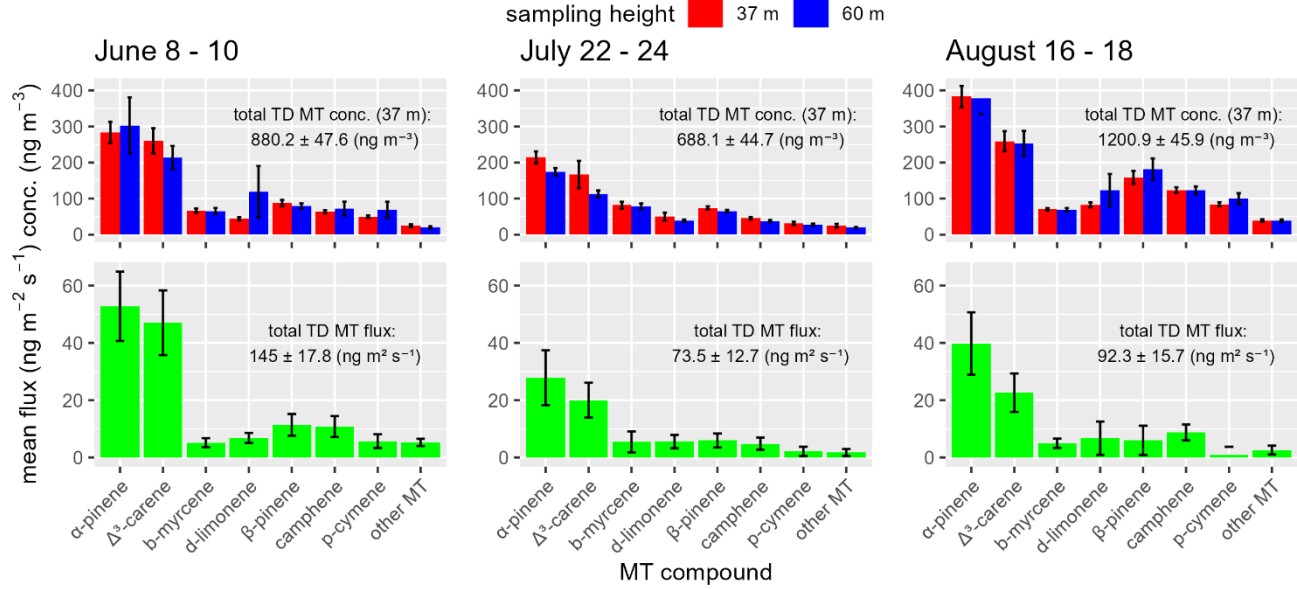

**Figure 9: Monthly sampling-period mean concentrations of MT compound species at 37 (red) and 60 m (blue) on the station flux tower at the ICOS Norunda boreal forest, as well as sampling period mean speciated MT fluxes (green), during June 8 – 10 (a), July 22 - 24 (b) and August 16 - 18 (c). Vertical error bars indicate the standard mean error. The daily mean concentrations and fluxes are shown in Fig. A4 in the Appendix.**

### 3.4.2 Vocus & ATD-GC-MS concentration comparison

We compared the sum of speciated MTs measured by GC–MS with the total MT concentration measured by Vocus PTR-ToF-MS. The comparison is displayed in Fig. 10. Based on comparison with 2022 precut TD measurements, it is expected that the sum of MT compounds identified by TD sample analysis represent 70 - 80% of total atmospheric MT concentration. During July 22 - 24, the ATD-GC–MS obtained a concentration of 610($\pm$30) ng m$^{-3}$ while the Vocus measured 851($\pm$27) ng m$^{-3}$ for MTs. During August 16 - 18, the ATD-GC–MS obtained a daytime concentration of 1201($\pm$73) ng m$^{-3}$ while the Vocus measured 1790($\pm$73) ng m$^{-3}$ for MTs.

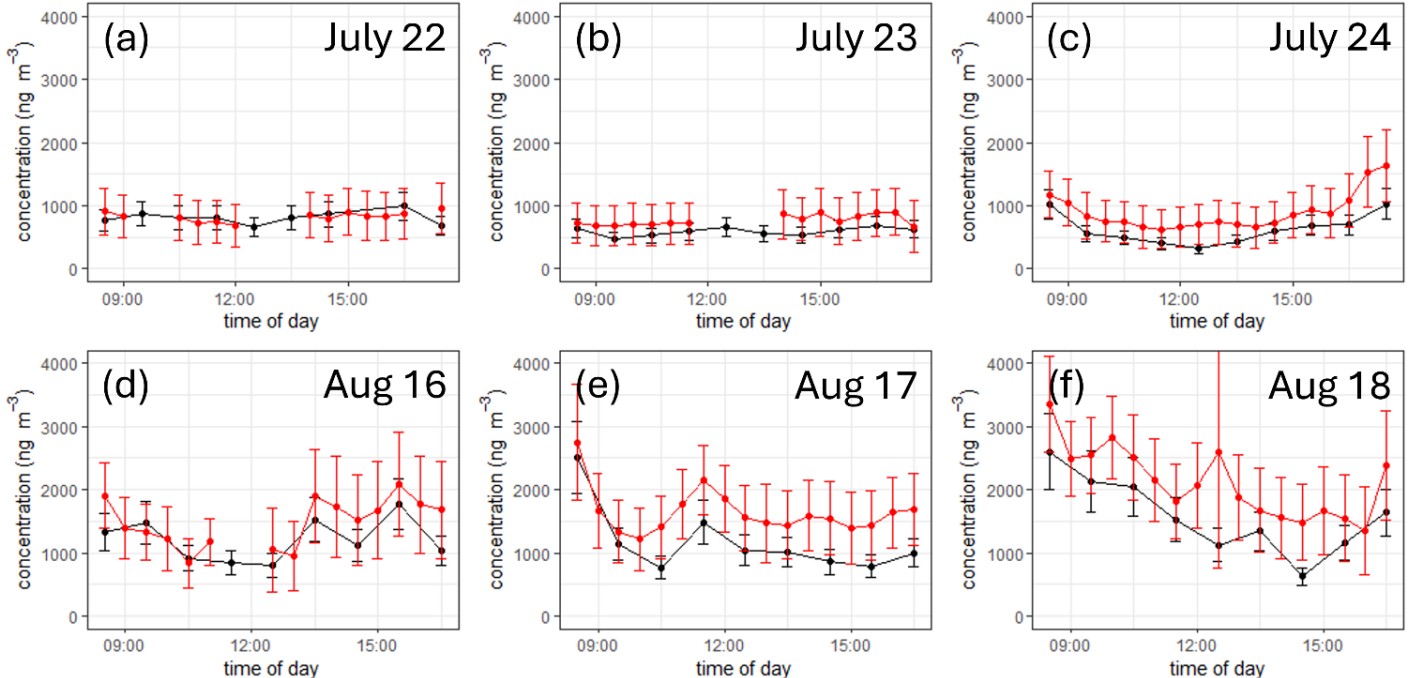

**Figure 10: Comparison of MT concentration measurements from the Vocus PTR-ToF-MS (red) and the sum of speciated MT concentrations (black) from the TD sample GC-MS analysis. Summed TD MT displayed were measured at the 37 m TD sampling height. Error bars indicate (red) standard deviation of 30min Vocus MT concentration and (black) uncertainty of summed TD MT concentrations based on the error analysis for TD sampling and analysis presented in section 2.7. Shown are half-hourly timeseries of the MT concentration measurements from both Vocus and manual TD sampling for the (a-c) 2020 July 22 - 24 and (d - f) 2020 August 16 - 18 TD sample periods.**

## 3.5 Vocus EC flux results

During daytime sampling, the values of the Vocus monoterpene flux typically ranged from 100 to 150 ng m$^{-2}$ s$^{-1}$ (see Fig. 2 & 5). Nighttime flux measurements should be treated with care, as stable atmospheric conditions during the evening lead to large uncertainties (e.g., Aubinet et al., 2012b). A combined estimate of the mean flux from the three-day sampling periods in June, July (start of Vocus campaign) and August (end of combined campaign) was performed for the surface layer gradient (SLG) fluxes. The mean flux for each of these three-day sampling periods was calculated to reduce the large uncertainty in the SLG flux estimates, by reducing the influence of random errors in the calculation. For the period of June 8-10 gradient samples, the estimated mean summed MT flux was 108($\pm$14) ng m$^{-2}$ s$^{-1}$. For the period of July 22-24 TD gradient samples, the estimated mean of the summed MT flux was 70($\pm$20) ng m$^{-2}$ s$^{-1}$. For the period of August 16-18 TD gradient samples, the estimated mean summed MT flux was 90($\pm$20) ng m$^{-2}$ s$^{-1}$. For comparison, during the 2020 campaign period when the Vocus was also operating, for the daytime sampling periods when TD sampling was performed the mean total MT flux from Vocus eddy-covariance measurements for July 22-24 was 105($\pm$3) ng m$^{-2}$ s$^{-1}$ and during August 16-18 was 155($\pm$7) ng m$^{-2}$ s$^{-1}$. For these same periods, the estimated isoprene flux was 20($\pm$9) ng m$^{-2}$ s$^{-1}$ and 23($\pm$10) ng m$^{-2}$ s$^{-1}$, respectively.

To determine whether there were any significant changes in the relative mixture of speciated MT compounds emitted over the course of the 2020 campaign, a two-way ANOVA statistical analysis was performed using the gradient-derived speciated MT fluxes. First, to eliminate the temperature-dependence of the MT emissions, the speciated fluxes were used to calculate temperature-normalized (to $30°C$) emissions $E_0$ using the equation $E = E_0\exp[\beta(T - 30°C)]$ (Guenther et al., 1993), where $E$ are the total MT emissions, $\beta$ is an empirically fitted coefficient (found for the Norunda 2020 campaign to be $\beta = 0.094$), and $T$ is the temperature in $°C$. Next, for the two-way ANOVA analysis, the flux data was sorted according to the monthly period (i.e., June, July, or August), and day of each sampling period (i.e., day 1, 2, or 3) when each gradient sample was collected. This allowed for the comparison of MT fluxes between sampling days and between sampling periods. The p-values from the analysis indicated that there was no significant ($p < 0.05$) trend in the measured flux between the monthly sampling periods for the speciated MT compounds observed, with the exception of a weak negative trend in the emission of $\Delta^3$-carene when comparing the three monthly sampling periods, in particular, between June and August ($p<0.0309$). This indicates that at a ecosystem-scale level of emissions from the Norunda forest canopy, following from the gradient samples, at least during the summer months of June to August, no statistically significant variation was noted in the relative mixture of MT compounds emitted with respect to the time of season.

**4 Discussion**

We conducted a measurement campaign in a boreal forest in order to estimate speciated monoterpene concentrations and fluxes during Scandinavian summer. The flux of MT compounds at Norunda was observed to primarily consist of α-pinene, followed by $\Delta^3$-carene (at approximately 60–70% the rate of α-pinene). This is similar to the preponderance of α-pinene, followed by $\Delta^3$-carene, observed by Hakola et al. (2012) above the boreal forest at the Hyytiälä research station in Finland, a predominantly Scots pine forest with nearby Norway Spruce, as well. Speciated MT emissions showed no significant differences during summer-seasonal cycle. Only a slight weakening of the temperature-normalized emission rate for $\Delta^3$-carene was observed over the course of the 2020 measurements (June – August). It should be noted that observations did not extend to the spring and autumn, hence further attention to the seasonal behavior of individual MT species in BVOC emission and climate models may be warranted in future investigations.

These gradient method flux measurements allow the speciated flux to be evaluated at an ecosystem-level scale, which sidesteps a potential issue for understanding ecosystem-atmosphere MT exchange, the large variability in speciated MT emissions that can exist among pine or spruce groups. Such variability, as observed in chamber measurement studies, can occur even among members of the same tree species and population (Bäck et al., 2012; Hakola et al., 2017).

The observed MT emissions were significantly temperature-dependent (fitting well to the storage-based pool emission algorithm of Guenther et al. (1993)) and that most of the forest MT emission originates from plant storage structures rather than from de novo synthesis as is the case for certain boreal tree species (e.g., Ghirardo et al., 2010). This is consistent with the forest tree species composition and the fact that MT mixing ratios were higher at night, with the lower boundary layer height and stable atmosphere, despite that nighttime MT emission rates are lower (e.g., Davison et al. 2009, Seco et al. 2013). The influence of chemical degradation on the surface exchange rates of isoprene, MT, and SQT was also investigated. While the impact of chemical degradation on isoprene exchange rates was negligible (<5%; less that EC flux uncertainties), a significant influence on the nighttime MT exchange rate was observed, with the exchange rate being on average ca. 10.8 % (varying from 6.8 to 14.1 %) greater than measured MT flux. This was far more evident with the overall SQT exchange rates, which diurnally were on average ca. 160 % (varying 130 to 180 %) greater during day and ca. 190 % (varying from 140 to 230 %) greater during night than the measured SQT flux.

The impact of MT and SQT chemical speciation on inferred SER rates should also be noted. Since the various individual MT and SQT compounds react at different rates with OH, $O_3$, and $NO_3$, these differences in turn affect the total MT and SQT fluxes that are ultimately observed above canopy using Vocus PTR-ToF-MS. Conversely, the estimation of SER rates from total MT and total SQT flux observations is therefore dependent on the relative mixture of emitted compounds considered. For example, for $\Delta^3$-carene (the 2$^{nd}$ most commonly observed MT compound during the 2020 Norunda campaign, following a-pinene) the reaction rates for the radicals OH, $O_3$, and $NO_3$ are $8.8\times10^{-11}$, $3.7\times10^{-17}$, and $9.1\times10^{-12}$ $cm^3$ $molec^{-1}$ $s^{-1}$, respectively (Atkinson, 1997). When these values are implemented in our quantification of chemical degradation effects on the MT SER rates, in lieu of the reaction rate constants for α-pinene, it yields nighttime SER values that are ca. 12.6 % (varying 7.7 to 16.9 %) greater than measured flux. This is an increase from the nighttime SER estimate based on the reaction rates of α-pinene of ca. 10.8 % (varying 6.8 to 14.1 %) for this SER-to-flux comparison. Also present are MT compounds that are far more reactive than either α-pinene or $\Delta^3$-carene. For example, relative to individual MT compounds, it was observed that the combined concentration of b-myrcene and d-limonene was exceeded only by α-pinene and $\Delta^3$-carene. The OH, $O_3$, and $NO_3$ rate constants for β-myrcene are $2.2\times10^{-10}$, $4.7\times10^{-16}$, and $1.1\times10^{-11}$ $cm^3$ $molec^{-1}$ $s^{-1}$, while for d-limonene they are $1.7\times10^{-10}$, $2\times10^{-16}$, and $1.22\times10^{-11}$ $cm^3$ $molec^{-1}$ $s^{-1}$, respectively (Atkinson, 1997). When we substitute these reaction rate constants into our SER evaluation procedure, for our SER-to-flux comparison, we find for b-myrcene during nighttime ca. 24.6 % (varying from 16.7 to 33.3 %) and during daytime ca.17.5 % (varying 13.1 to 21.8 %). Meanwhile, for d-limonene, we find during nighttime ca. 19.3 % (varying from 12.7 to 26.4 %) and during daytime ca. 10.8 % (varying from 7.9 to 13.4 %). These values are ca. 2.4 and 1.8 times greater during nighttime, and ca. 3.8 and 2.3 times greater during daytime, for b-myrcene and d-limonene, respectively, than the corresponding increases found for α-pinene. For the average concentration apportionment, over the course of the 2020 Norunda campaign, of the identified MT compounds α-pinene, $\Delta^3$-carene, b-myrcene, d-limonene, β-pinene, and camphene, the mean values of their reactivities for OH, $O_3$, and $NO_3$ are $8.9\times10^{-11}$, $9.79\times10^{-17}$, and $6.83\times10^{-12}$ $cm^3$ $molecule^{-1}$ $s^{-1}$, respectively. When using these reaction rate constants instead of those of α-pinene for the MT SER evaluation,

this yields a diurnal SER estimate for total MT that is very close (~$\Delta$1%) to the estimate formed using the OH, O$_3$, and NO$_3$ reactivities of a-pinene, with an nighttime increase in SER from the measured MT fluxes of 12 % (varying from 7.7 to 15.7 %), and a daytime increase of 6.1 % (varying from 4.3 to 7.8 %). This indicates that a-pinene likely acts as a good proxy for the mixture of MT emissions from this and similar boreal forests when attempting to infer the impact of chemical degradation on observed fluxes as collected using total MT measurement instruments such as PTR-ToF-MS.

For the speciated MT fluxes it has been observed that, despite the relatively large uncertainty involved with the speciated MT fluxes (compared to total MT EC flux measurements), the fraction of MT flux from the most common MT compound, α-pinene, is somewhat higher than the fraction of their concentration relative to the other MT compounds (see Fig. 2 & 10). This condition is also found in other studies (e.g., Rinne et al. 2000). A potential explanation of this observation is the influence of chemical degradation, particularly with respect to more reactive compounds such as myrcene and d-limonene, during turbulent transport within and above canopy. Another potential influence on speciated MT flux versus concentration observations is from transport from outside the flux tower footprint. The observed concentrations used for gradient measurements are the result of emissions, sinks, chemical transformation and transport, whereas the observed fluxes reflect the emissions and sinks solely in the flux footprint of the tower.

## 5 Implications

A significant implication of this work is the role that seasonal changes can have on the speciation of MT compounds and subsequently their impact on air chemistry. BVOCs like MT influence the tropospheric ozone budget (Archibald et al. 2020). MTs also contribute to the formation and/or growth of atmospheric SOA due to the gas/particle partitioning of their reaction products in the troposphere. The structure of MTs has a significant role in their reactivity. For example, endocyclic monoterpenes (e.g., limonene, α-pinene, and 3-carene) have a greater aerosol formation potential and tend to react faster than compounds with exocyclic double bonds (e.g.,β-pinene and camphene). b-myrcene, an acyclic monoterpene with three double-bonds, has a significant impact on the overall MT reactivity of MTs investigated by TD sampling, despite being at lower concentrations than many of the observed MT compounds. For example, in July, while less than 14% of total MT concentration, b-myrcene made up more than half (52 %) of overall observed MT ozone reactivity. Fast-reacting MTs can represent a significant fraction of total reactivity even at low concentrations (e.g., Yee et al., 2018). Subsequently, even small seasonal changes in their concentration can represent large changes in the reactivity of the overall MT population emitted.

From July to August, the average of the OH, ozone, and NO$_3$ reaction rate coefficients for the observed MT mixture decreased. This was mainly driven by month-to-month trends in the relative abundance of 3-carene and b-myrcene. While overall reactivity of MT towards OH, ozone, and NO$_3$ increased from July to august, due to an increase in overall MT concentration, that increase is less than would be expected if changes in speciation were neglected, particularly for ozone (from July to

August, for example, a 31±18 % overestimate of ozone's MT reactivity). The relative average decrease in total oxidative capacity between July and August sampling periods amounts to $16 \pm 10$ %, when evaluated at the average ozone and OH concentrations investigated during the TD sampling periods. This is primarily due to the impact of speciation changes on ozone's oxidation capacity (~24 $\pm$ 11 % decrease for ozone, vs. ~10 $\pm$ 3 % decrease for OH). At night, when $NO_3$ can form (modeled at 1 pptv), and when average OH and ozone concentration are lower, this drop in total oxidative capacity was $12 \pm 3$ % (decrease for $NO_3$'s capacity was $8 \pm 3$ %). While this is within the uncertainty of modeled OH, ozone, and $NO_3$ concentrations utilized in our evaluation of chemical degradation, it is still a significant difference that warrants further attention when considering the impacts of MTs on air chemistry at the ecosystem-scale for boreal forests in BVOC emission modeling.

While not the main focus of this investigation, the impact on SQT speciation is likewise significant. For example, at the 37 m height used for TD sampling, the measured concentration of longifolene greatly exceeds that of α-humulene at a ratio of 80%-to-20%. However, the reactivity of α-humulene towards ozone is more than $2.3 \times 10^4$ times greater than that of longifolene. An approximate calculation using the same chemical degradation evaluation presented in this investigation indicates the converse for the speciation of these SQT emissions, indicating that α-humulene emission dominates that of longifolene at a ratio of nearly 90%-to-10%. As can be seen, this presents a useful potential tool for investigating the ecosystem-scale speciation of SQT emissions.

Another potential implication is the determination of SQT surface exchange rates at the ecosystem-scale using a combination of measured fluxes and modeled F/E ratio. The breakdown of SQT has long-been a limiting factor in evaluating surface emissions based on observed fluxes (e.g., Duhl et al., 2008; Helmig et al., 2006; Pollmann et al., 2005). The determination of chemical degradation's impact on the surface exchange of reactive BVOC fluxes can be evaluated using measurements of ozone and the radicals OH and $NO_3$. While OH measurements have classically been difficult to conduct (e.g., Heard, 2006; Heard and Pilling, 2003; Stone et al., 2012), many stations regularly monitor ozone, often the leading contributor (by an order of magnitude (see Fig. A2)) to the SQT chemical degradation that attenuates observed SQT fluxes. In parallel with GC-MS or more recent tools such as ultrafast GC (e.g., Materić et al., 2015) for determining SQT speciation at the PTR eddy-covariance inlet height, combining PTR-ToF-MS eddy-covariance flux observations with chemical degradation analysis offers a promising avenue for evaluating ecosystem-scale surface exchange rates of SQTs and other highly reactive BVOC compounds.

**Code and data availability**

Station atmospheric and ecosystem data from the ICOS Norunda station is publicly available at https://www.icos-sweden.se/norunda. The campaign data and scripts are available from the authors upon request. The original Julia code for TOF-Tracer2 (e.g., Fischer et al., 2021), which was modified to process the raw Vocus PTR-ToF-MS spectral data, can be

found at https://github.com/lukasfischer83/TOF-Tracer2. The Matlab code for the eddy-covariance package InnFLUX (e.g., Striednig et al., 2020) can be found at https://www.atm-phys-chem.at/innflux.

## Author contributions

JR, RCP, and TH initiated the study. JR and TH supervised the study and acquired the primary funding to support this research. CW, CM and RCP conducted the Vocus PTR-ToF-MS campaign measurements, and TH assisted with Vocus field calibration preparation and measurements. RCP performed the TD gradient sampling and laboratory analysis at Lund University. JC assisted with post-processing of GC-MS laboratory files following TD sample analysis. RCP conducted the main study analysis and prepared the manuscript and figures, with contributions from all co-authors.

## Competing interests

A co-author is a member of the editorial board of Atmospheric Chemistry and Physics.

Disclaimer: The views and opinions expressed are those of the author(s) only and do not necessarily reflect those of the European Union. Neither the European Union nor the granting authority can be held responsible for them.

## Acknowledgements

This work was supported by the Swedish Research Council Formas (2017-01474). The Norunda research station received funding by the Swedish Research Council VR, Grant 2019-00205. We would like to thank the staff, A. Båth, I. Lehner, and Dr. M. Mölder, at the ICOS Norunda (SE-Nor) research station for their help and support. We further thank staff member Dr. M. Mölder for his assistance with the sonic anemometer profile on the Norunda tower. We would also like to thank Dr. Katerina Sindelarova for her assistance in quantifying the boreal contribution to the global and northern hemispheric summertime MT emission inventory. This work was also supported by the Swedish Environmental Protection Agency (Naturvårdsverket), Knut and Alice Wallenberg foundation, European Comission H2020 project FORCeS, and ERC project INTEGRATE. Airborne LiDAR was acquired by Kljun et al. with support from the British Natural Environment Research Council (NERC/ARSF/FSF grant EU10-01 and NERC/GEF grant 933). The post-processing of ATD-GC-MS files by J. K. Chan was supported by the Danish National Research Foundation (VOLT, DNRF168). This research has been partly supported by the "Greenhouse Gas Fluxes and Earth System Feedbacks" (GreenFeedBack) project from the European Union's Horizon Europe Framework Programme for Research and Innovation (project no. 101056921), funded by the European Union.

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

**Appendix sections:**

## Determination of displacement height $d$ and the $\Gamma$ correction factor

The displacement height $d$ was determined using the velocity $u$ and friction velocity $u_*$ data from the vertical profile of sonic anemometers on the Norunda flux tower, using the approach described in Meelis et al. (1999), as well as with the empirical relation d = 0.86h found therein (where h is canopy height). The $u$ and $u_*$ data was filtered according to the selection criteria: $u(87\,m) - u(44\,m) > 0.2\,m\,s^{-1}$ and $u_* > 0.1\,m\,s^{-1}$. Good agreement for the displacement height d was found for d = 24 m.

To correct the SLG flux measurements of speciated MT flux for the influence of the roughness sublayer (RSL) above canopy, the Γ correction factor was determined by integrating the profile for γ(z) above the canopy (and within the RSL) from the lower TD sampling height (37 m) to upper sampling height (60 m).

The γ-coefficients above the canopy were calculated from the sensible heat flux $H$, temperature $T$, and friction velocity $u_*$ from the sonic anemometer profile according to $\gamma = \Phi_h^{MO}(\zeta)/\Phi_h^{meas}(z)$, where $\Phi_h^{MO}$ is the dimensionless gradient for sensible heat according to Monin-Obukhov similarity theory (where ζ is the stability parameter (z-d)/L), $\Phi_h^{meas}(z)$ is the dimensionless gradient for sensible heat according to measurements on the flux tower, $k$ is the von Kármán constant, $\rho$ is the density of air, and $c_p$ is the specific heat capacity of dry air. To minimize errors in the derivative of θ(z) when calculating $\Phi_h^{meas}(z)$, the gradient $d\theta/dz$ was determined by fitting the profile sonic data for potential temperature θ(z) using the equation $\theta = a + b\ln(z) + c\ln^2(z)$ (e.g., Mölder et al., 1999).

Measured γ values during the TD sampling periods on June 8-10, July 22-24, and August 16-18, between 9:30 CEST to 17:00 CEST, were collected. This vertical γ-profile above forest canopy was then used to fit the coefficients for a continuous profile for γ(z). We applied the following formula

$\gamma \cong 1 + Ae^{-B(z-d)}$, (A1)

where A and B are undetermined coefficients. This fitting formula derives from consideration of Garratt (1980) and Harman and Finnigan (2007). Following nonlinear regression, the fitted coefficients were found to be A = 5.26 and B = 0.12 (ΔA = 1.01, ΔB = 0.02). A plot of the vertical γ-coefficient profile between the TD sampling levels at $z_1$=37 m and $z_2$=60 m during the 2020 Norunda campaign is shown in Figure A1 (the fitted curve for eq. A1 appears in blue).

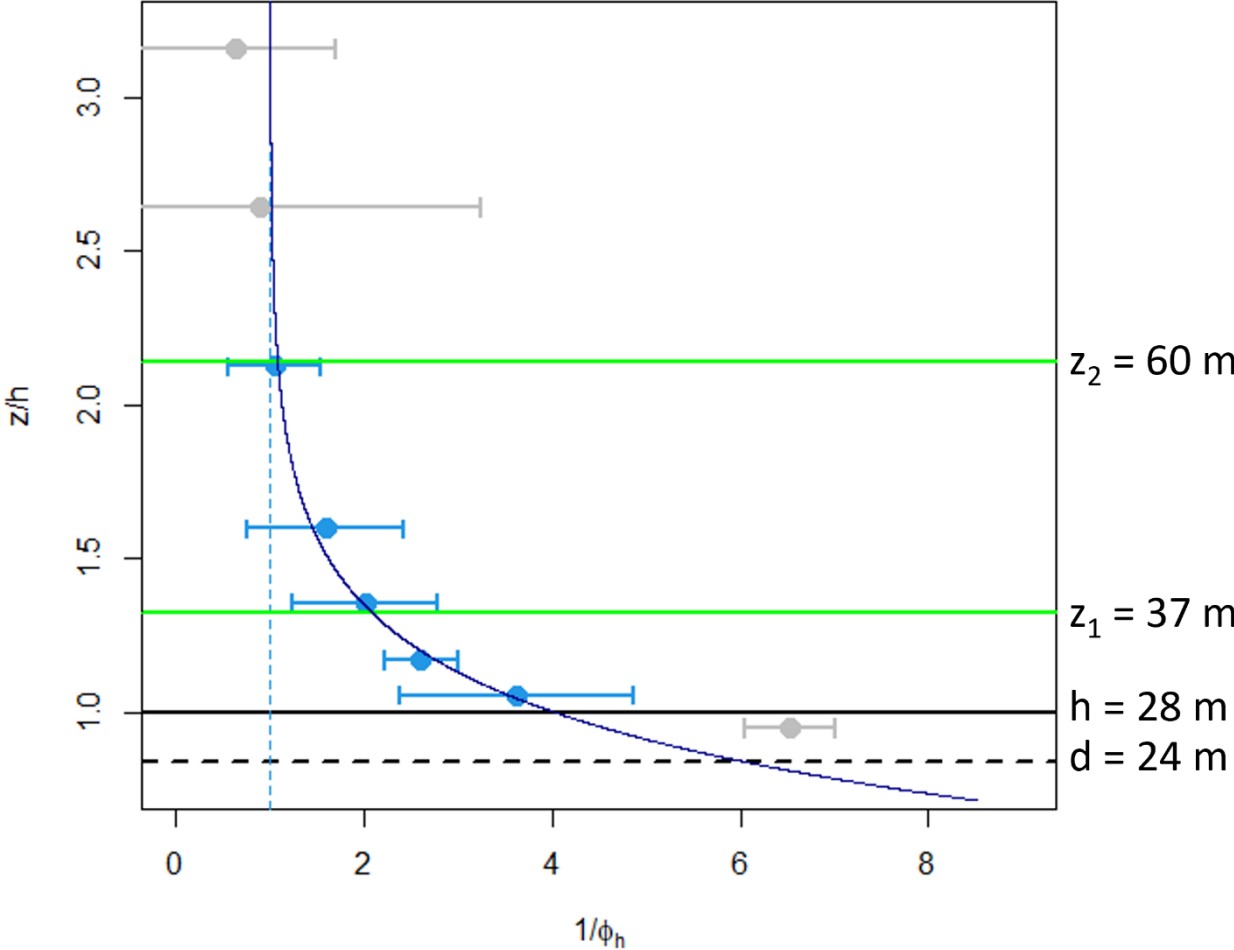

Figure A1: Vertical profile of $\gamma$ above the Norunda forest canopy during daytime TD sampling. The x-axis shows the value of $\gamma = 1/\phi_h = \Phi_h^{MO}(\zeta)/\Phi_h^{meas}$, and the yaxis shows the normalized height z/h. Points indicate median $\gamma$ (± standard error of median) measured from $\Phi_h^{MO}(\zeta)/\Phi_h^{meas}$ during for the TD sampling periods (9:30 to 16:30 CEST; June 8-10, July 22024, & Aug 16-18). Blue points (near or within RSL) indicate y used to fit function for the RSL layer. Dark blue curve indicates regression fit of $\gamma = 1 + Ae^{-B(z-d)}$ to the measured γ-profile values (A = 5.26, B = 0.12). Vertical dashed line indicates γ = 1. Horizontal

lines indicate (dashed black) displacement height d, (solid black) canopy height h, and (green) the $z_1$=37 m and $z_2$=60 m TD sampling heights.

Using $z_1$ = 37 m, $z_2$ = 60 m, $d$ = 24 m, and fitted coefficients A = -5.26 and B = 0.12, the integral of equation A1 was then used to determine a mean enhancement factor $\Gamma = \frac{1}{z_2 - z_1} \int_{z_1}^{z_2} \gamma(z) dz$. For the Norunda 2020 SLG

flux measurements, this yielded Γ=1.36±0.09.

$$\Gamma(z_1 = 37 \ m, z_2 = 60 \ m) = \frac{1}{z_2 - z_1} \int_{z_1}^{z_2} \gamma(z) dz = \frac{A}{B} \left[ \frac{e^{-B(z_2-d)} - e^{-B(z_1-d)}}{z_2 - z_1} \right] + 1 \cong \underline{1.36}. \ \text{(A2)}$$

The uncertainty of this fitted estimate was determined using the standard errors of coefficients A and B,

and propagation of uncertainty $\Delta\Gamma = \sqrt{\left(\frac{\partial\Gamma}{\partial A}\Delta A\right)^2 + \left(\frac{\partial\Gamma}{\partial B}\Delta B\right)^2} \cong 0.09$.

**Effect of chemical degradation on isoprene, MT, and SQT surface exchange rates (model calculations)**

The effect of chemical degradation on isoprene, MT, and SQT surface exchange rates was explored using the modeling supplement of Rinne et al. (2012), which made use of a stochastic Lagrangian transport chemistry model, and by parameterizing the reaction rates for ozone, OH and $NO_3$, to estimate the surface
exchange rates for the 2020 Norunda campaign's terpenoid EC flux dataset.

The impact of chemical degradation on measured fluxes of reactive compounds is closely related to the ratio the mixing timescale $\tau_t$ to the compound's chemical timescale $\tau_c$, also known as the Damköhler number $Da$ (Damköhler 1940), which can be written as

$$Da = \frac{\tau_t}{\tau_c}. \ \text{(A3)}$$

In the case of $Da \ll 1$, the flux at the measurement height closely corresponds to the emission rate. For relatively large $Da$ values, however, the compound will be impacted by chemical degradation before reaching the measurement height, and the observed flux will be lower than the primary emission.

To estimate the flux-to-emission ratio (F/E) from measured Da-number ratios, lookup tables were used from the supplementary materials provided by Rinne et al. 2012. These tables relate the canopy-top
Damköhler number ($Da_h$) to the flux-to-emission ratio (F/E) for specific normalized heights z/h above the forest canopy-top height h. In this model, the LAI of the canopy, modeled according to the typical profile of a Scotts pine forest, is 3.5. For comparison, LAI at Norunda is 3.6 (± 0.4) m$^2$ m$^{-2}$ (Petersen et al., 2023). Using an approximation for the mixing timescale of $\tau_t = z/u_*$, the canopy-top Damköhler number ($Da_h$) was calculated using the relation

$$Da_h = \frac{h}{u_* \tau_c}, \ \text{(A4)}$$

Where the friction velocity $u_*$ was measured at 37 m on the station flux tower (see example station BVOC inlet setup in Fig. 1 of the main text), $h$ is the canopy height (28 m), and $\tau_c$ is the chemical lifetime of a given BVOC compound. The chemical lifetime $\tau_c$ can be calculated as

$$\tau_c = \left(k_{X,OH}[OH] + k_{X,O_3}[O_3] + k_{X,NO_3}[NO_3] + k_{X,photolysis}\right)^{-1}, \ \text{(A5)}$$

where $[OH]$, $[O_3]$, and $[NO_3]$ are the concentrations of OH, $O_3$, and $NO_3$, respectively, and where $k_{X,OH}$, $k_{X,O_3}$, and $k_{X,NO_3}$ are the reaction rate constants between the compound of interest $X$ (i.e., isoprene, MT, SQT, ect.) and the radicals OH, $O_3$, and $NO_3$, respectively. Finally, $k_{X,photolysis}$ is the photolysis rate of compound $X$. As $k_{X,photolysis}$ for isoprene, MT, and SQT is much less than the rate of reactions with radicals OH, $O_3$, and $NO_3$, as in Rinne et al. (2012), it is dropped from subsequent calculations.

The reaction rate constants for isoprene, α-pinene and β-caryophyllene with respect to OH, $O_3$, and $NO_3$ used for the Norunda 2020 campaign SER analysis are shown in table A3.

For the 2020 Norunda campaign SER analysis, modeling of the diurnal cycle of OH and $NO_3$, as well as local ozone data from the nearby Norunda-Stenen monitoring site, were used to calculate the timeseries of $\tau_c$ for isoprene, MT, and SQT, to then calculate $Da_h$ for applying the Rinne et al. (2012) F/E lookup tables.

The radicals OH and $NO_3$ were diurnally simulated for the $Da_h$ calculations. The modeling choices for OH and $NO_3$ were informed by the HUMPPA-COPEC-2010 measurement campaign in Hyytiälä, Finland, and the settings implemented in Rinne et al. (2012). The OH concentration was assumed to vary diurnally between $0.3 \times 10^6$ and $1.2 \times 10^6$ molecules $cm^{-3}$ over a Gaussian peak profile ($t_{mean}$ = 12:00 CEST, $\sigma_t$ = 3.5 hrs). The uncertainty ΔOH was assumed to be 40% of concentration.

$NO_3$ was assumed to vary as a step function, with a daytime concentration of zero and a nighttime concentration of $2.5 \times 10^7$ molecules $cm^{-3}$ (1 pptv), consistent with the breakdown of $NO_3$ under direct sunlight. The $NO_3$ concentration was assumed to linearly ramp between daytime and nighttime values during a 30-min widow before the daily sunrise and after daily sunset. The uncertainty Δ$NO_3$ was assumed to be 50% of concentration.

The ozone data used for the $\tau_c$-calculations came from the Norunda-Stenen monitoring station (1.4 km east of Norunda flux tower). To accurately reflect above-canopy nighttime concentrations, as model ozone data was collected well below canopy height, a nighttime minimum concentration was set at $5 \times 10^{11}$ molecules $cm^{-3}$ (ca. 20 ppbv). This conforms with previous campaign studies of the diurnal vertical ozone profile within and above the Norunda forest summer canopy (Petersen et al. 2023), where a significant
nighttime sink was observed below canopy height (28 m). The uncertainty Δ$O_3$ was assumed to be 25% of concentration.

The diurnal mean reaction rates of ozone, OH, and $NO_3$ with isoprene, as well as with α-pinene and β-caryophyllene (used as examples for MT and SQT, respectively), are shown in Fig. A2 below.

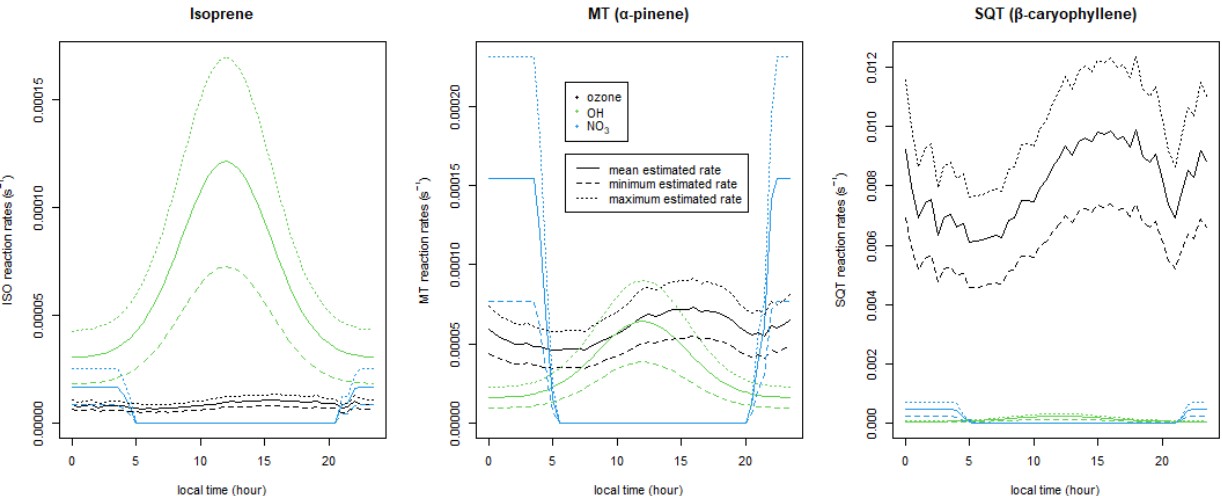

**Figure A2: Diurnal mean reaction rates of ozone (black), OH (green), and NO₃ (blue) for isoprene, MT (α-pinene) and SQT (β-caryophyllene). Lines indicate the mean (solid) reaction rates, as well as the minimum (dashed) and maximum (dotted) -estimated values of the reaction rates based on the concentrations of OH, O₃, and NO₃ as well as the uncertainties ΔOH, ΔO₃, and ΔNO₃.**

For the estimate of F/E from $Da_h$, the lookup tables were used from the supplementary materials provided by Rinne et al. 2012. These tables relate the canopy-top Damköhler number ($Da_h$) to the flux-to-emission ratio (F/E) for specific normalized heights z/h above the forest canopy-top height h. In this model, the LAI of the canopy, modeled according to the typical profile of a Scotts pine forest, is 3.5.

The F/E modeling also includes the output for different parameterizations, such as canopy-source emission vs ground-source emission, as well as the relative scaling of oxidation rates above vs below-canopy. Several examples of the estimated diurnal F/E ratios for the 2020 Norunda campaign under these different parameterizations are shown below (Fig. A3).

The ground source, for example, is likely appropriate for certain types of forest floor SQT emission (cite field studies) In addition, daytime mixing may lead to equal or similar oxidation levels above and below
canopy, as the canopy is well-mixed, while nighttime oxidation rates can become very stratified within stable canopy air, as was frequently observed for vertical profile of ozone within the Norunda canopy at night (see Fig. 7 therein). For the SER estimates presented in the main text, the source is modeled as being in the canopy, and the relative oxidation rates above and below canopy are modeled as being equal. For SQT (and for predominantly a canopy source), this can be addressed by interpolating between table
*lookup_C_o025.txt* (i.e., nighttime stratified ozone) and table *lookup_C_o100.txt* (i.e., daytime well-mixed ozone) using the Monin-Obukhov atmosphere stability length $L$.

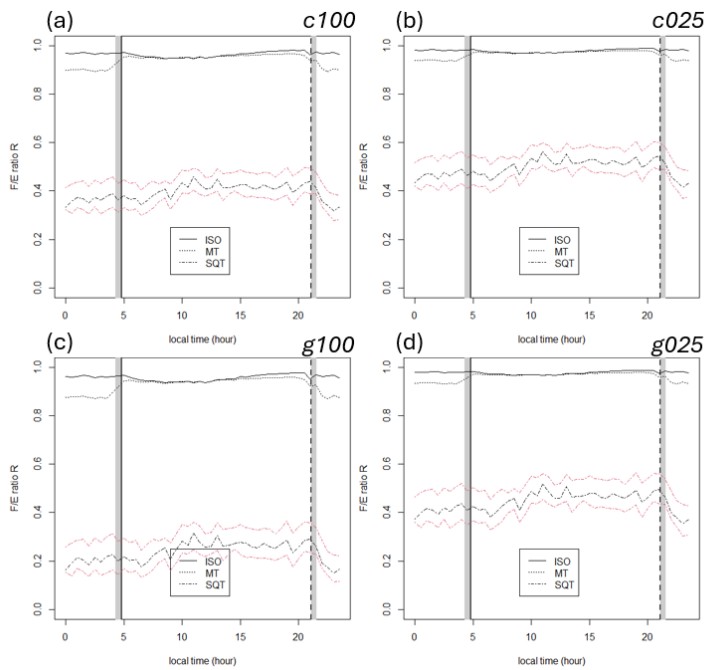

**Figure A3: Diurnal F/E ratios for isoprene, MT (α-pinene) and SQT (β-caryophyllene) for a range of different model parameterizations. (a, b) parameterized for canopy source. (c, d) parameterized for ground source. (a, c) below-canopy oxidation rate equals that of above-canopy. (b, d) below-canopy**
**oxidation rate 0.25x that of above-canopy. The estimated F/E ratios for the minimum and maximum of the ozone, OH, and NO$_3$ reactions rates are shown as dashed red lines.**

**Impact of chemical speciation on estimated surface exchange rate uncertainty**

Having described an approach for inferring surface exchange rates from measured fluxes, we now aim to investigate the
conditions when applying a proxy or group-average estimate for the R ratio is justified when estimating emission rates from chemically degraded flux measurements of total SQT and nighttime total MT by PTR-MS instrumentation. Using the ratio R = F/E (described in section 2.8), we can write the total SQT flux measured by the Vocus PTR-ToF-MS as $F = \sum_i F_i =$

$\sum_i E_i R_i = E \sum_i \chi_i R_i$, where $\chi_i$ is the fraction $E_i/E$ that each compound of the total mixture contributes to the total emission (with $\sum_i \chi_i = 1$). We can then define an effective R ratio for describing the total mixture as $R_{true} = \sum_i \chi_i R_i$, an emission-weighted average of the compound-specific R values for all compounds contributing to the total group emission rate. In practice, even for the relatively well-behaved MT mixture, our knowledge about $\chi_i$ is limited. For estimating the surface exchange rate it can, instead, be useful to use an estimate for R based on a compound commonly observed to dominate total emissions of a terpenoid group such as MT or SQT or, alternatively, use an estimate for the approximate emission rates based on previous chamber emission measurements at the surrounding site. The true and estimated total emission are $E_{true} = F/R_{true}$ and $E_{est} = F/R_{est}$, where $R_{est}$ is the assumed group average or compound-specific proxy used for estimating the total emission rate. The relative error in the estimated total emission rate is $\epsilon = |E_{est} - E_{true}|/E_{true} = \left| \frac{1}{R_{avg}} - \frac{1}{R_{true}} \right| / \left( \frac{1}{R_{true}} \right)$, which simplifies to

$$\epsilon = |R_{true} - R_{est}|/R_{est} \quad (A5)$$

We proceed from here to estimate the impact that individual compounds in the mixture might have on the estimated total emission rate. While $R_{true}$ might be illusive in practice, we can test the influence of individual compounds against an idealized case. For a test case where we can specify the emitted proportions, and in the case that the estimate for R is based on a single compound, Eq. A5 simplifies to

$$\left| \frac{R_{mix} - R_k}{R_k} \right| = \left| \frac{(\sum_i \chi_i R_i) - R_k}{R_k} \right| = \left| \frac{(\sum_{i \neq k} \chi_i R_i) - (1 - \chi_k) R_k}{R_k} \right| = \left| \frac{(\sum_{i \neq k} \chi_i R_i) - (\sum_{i \neq k} \chi_i) R_k}{R_k} \right| = \left| \sum_{i \neq k} \chi_i \frac{R_i - R_k}{R_k} \right| \quad (A6)$$

Where $\chi_k$ and $R_k = R_{est}$ are the emitted proportion and R ratio, respectively, of the proxy compound used for estimating the total surface exchange rate, and $R_{mix}$ is the weighted average of the R ratio based on the specified emitted proportions of the test case mixture. We can apply Eq. A6 to a simple test based on three compounds.

For this three-compound test, we assume that the SQT mixture is similar to that observed in chamber emission measurements at Norunda in July & August 2014 (e.g., Wang et al. 2018) and at a similar boreal forest by Hellén et al. (2018). The compounds, proportions, and calculated R ratio are summarized in the table below. Based on observations, the proportion of longifolene is likely much lower (<1%). From Kim et al. (2011), for an air temperature of 298 K and at sea level pressure, we calculate the reaction rate coefficients for β-farnesene to be $k_{OH} = 2.9 \times 10^{-16}$ cm$^3$ molecule$^{-1}$ s$^{-1}$ and $k_{ozone} = 6.9 \times 10^{-16}$ cm$^3$ molecule$^{-1}$ s$^{-1}$. For the daytime OH and ozone concentrations presented in table A3, the chemical lifetime of β-farnesene during the day is ca. 16 minutes.

| compound | proportion Xi | calculated ratio Ri |
|---|---|---|
| β-caryophyllene | 85% | 0.49 |
| Longifolene | 5% | 0.992 |
| β-farnesene | 10% | 0.885 |

Applying this to Eq. A6, we find that applying β-caryophyllene as a proxy leads to an offset of about 13 %. In comparison, the influence of uncertainty from modeled concentrations of daytime OH (±40%) and ozone (±25%) on the calculated surface exchange rates from the chemical degradation analysis 12%) is about the same.

In a more general view, since $R_{true}$ can be expressed as a linear weighted average over the full set of compounds making up the true emission, we can divide it between a subset of known compounds and its complement subset. Let $S$ be the subset of known compounds making up a linear estimate for $R_{est}$ and let its complement, $S^c$, be the remaining, missing group of unknown compounds. We can then define $\chi_{est} = \sum_{j \in S} \chi_j$ and $\chi_{miss} = \sum_{k \in S^c} \chi_k = 1 - \chi_{est}$ where $\sum_i \chi_i = \chi_{est} + \chi_{miss} = 1$. From this step, we can break up $R_{true}$ into its known and unknown parts, with $R_{est} = \frac{1}{\chi_{est}} \sum_{j \in S} \chi_j R_j$ , $R_{miss} = \frac{1}{\chi_{miss}} \sum_{k \in S^c} \chi_k R_k$, and $R_{true} = \sum_i \chi_i R_i = \sum_{j \in S} \chi_j R_j + \sum_{k \in S^c} \chi_k R_k = \chi_{miss} R_{miss} + \chi_{est} R_{est}$. Applying Eq. A5 to these yields

$$\left| \frac{R_{true} - R_{est}}{R_{est}} \right| = \left| \frac{(\chi_{miss} R_{miss} + \chi_{est} R_{est}) - R_{est}}{R_{est}} \right| = \chi_{miss} \left| \frac{R_{miss} - R_{est}}{R_{est}} \right| < \varepsilon$$

Hence, based on assumptions regarding $R_{est}$, we can therefore estimate roughly the average chemical lifetime that the unresolve components of total SQT emission must have in order to fall outside the specified amount of acceptable uncertainty $\varepsilon$, based on the inequality $|R_{miss} - R_{est}| < \frac{\varepsilon}{\chi_{miss}} R_{est}$.

For example, if we are confident that at least 85% of total SQT emissions consists of β-caryophyllene, use it as a proxy for our total SQT estimate as in sections 2.8 and 3.3.1, and are willing to accept an uncertainty bound $\varepsilon = 15$ %, then solving the above inequality for $R_{miss}$ places the constraint $R_{miss} < 2 \times R_{est} = 0.98$. From this upper-bound on R, we can estimate the corresponding value of $D_{ah}$, and subsequently $\tau_{c,miss}$, by working back through the R-vs-$D_{ah}$ lookup tables of the chemical degradation analysis outlined earlier in the appendix (in this case, yielding $D_{ah} < 0.005$). Using Eq. A4, we can estimate the associated upper-bound for the average chemical lifetime from $\tau_c < \frac{h}{u_* Da_h}$. In this example (using table A3 for daytime friction velocity $u_* = 0.65$ m$^{-2}$ s$^{-1}$ and canopy height h = 28 m), to fall within $\varepsilon = 15$ %, the average chemical lifetime of the missing emission components is bounded by $\tau_{c,miss} < 2.4$ hr.

**Appendix figure: Daily mean concentrations and fluxes of MT compound species**

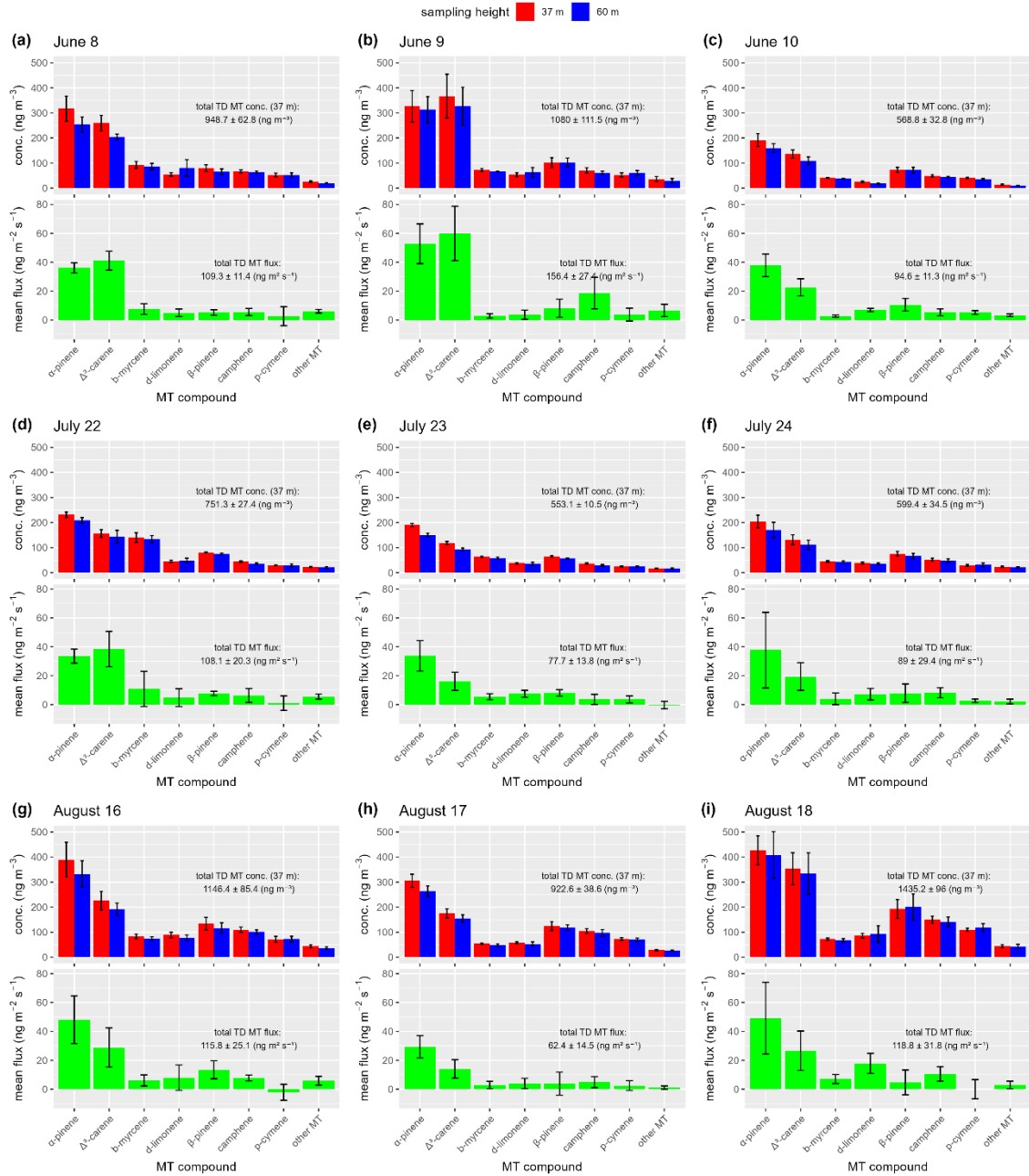


**Figure A4: Daily mean (09:30-17:00 CEST) concentrations of MT compound species at 37 (red) and 60 m (blue) on the station flux tower at the ICOS Norunda boreal forest, as well as daily mean speciated MT fluxes (green), during June 8 – 10 (a – c), July 22 - 24 (d - f) and August 16 - 18 (g - i). Vertical error bars indicate the standard mean error.**

## Comparison of friction velocity and sensible heat flux between 37 and 60 m

For Table A1, the data was divided into three wind-direction classes based on the two prevailing wind directions (between 245°-310° and 310°- 80°) during the campaign TD sampling on the Norunda flux tower on June 8-10, July 22-24, and August 16-18. The heterogeneity of the forest tree heights can be observed from the lidar height map presented in Fig. 3.

> Table A1 caption: The ratios of momentum and sensible heat fluxes measured at 60 m to those at 37 m (i.e., $r_{u_*} = \overline{u_{*,60m}/u_{*,37m}}$ and $r_H = \overline{w'T'_{60m}/w'T'_{37m}}$) during daytime (8:30am to 17:30pm) during the 2020 Norunda campaign season. L is the Monin-Obukhov length.

### Unstable: $-1000 < L < 0$

| wind direction | N samples | $r_{u_*}$ | $r_H$ |
| --- | --- | --- | --- |
| 80-245 | 509 | 0.98(±0.005) | 0.93(±0.007) |
| 245-310 | 198 | 1(±0.008) | 0.96(±0.011) |
| 310-80 | 261 | 0.97(±0.008) | 0.92(±0.009) |

### Neutral: $|L| > 1000$

| wind direction | N samples | $r_{u_*}$ | $r_H$ |
| --- | --- | --- | --- |
| 80-245 | 45 | 0.99(±0.015) | 0.95(±0.026) |
| 245-310 | 17 | 0.98(±0.040) | 0.86(±0.053) |
| 310-80 | 12 | 0.84(±0.022) | 0.85(±0.051) |

### Stable: $0 \leq L < 0$

| wind direction | N samples | $r_{u_*}$ | $r_H$ |
| --- | --- | --- | --- |
| 80-245 | 14 | 0.97(±0.030) | 1.04(±0.044) |
| 245-310 | 4 | 1(±0.080) | 1.01(±0.138) |
| 310-80 | 9 | 0.94(±0.066) | 0.97(±0.076) |

Table A2 caption: Summary of VOC concentration calibration coefficients and parameters for the concentration (ppbv) -calibrated timeseries data for the 2020 Norunda field campaign measurements. For each calibrated compound, the summary includes the compound name, quantitative ion formula, observed mass-to-charge ratio m/z [Th], molecular polarizability ($\alpha$), permanent dipole moment (D), calculated proton-transfer reaction rate constant (kPTR), fraction f of the compound's total signal attributed to the quantitative ion, and average of the compound sensitivity (attenuated) during the full 2020 Norunda field

| compound name | quantitative ion formula | m/z (Th) | molecular polarizability $\alpha$ ($10^{-24}$ cm$^3$) | permanent dipole moment D (Debye) | $K_{PTR}$ (E/N = 110 Td) ($10^{-9}$ cm$^3$ molec$^{-1}$ s$^{-1}$) | f (%) | sensitivity (cps ppbv$^{-1}$) |
|---|---|---|---|---|---|---|---|
| Isoprene | $C_5H_8H^+$ | 69.07 | 9.99 | 0.25 | 1.93 | 88.1 | 193 |
| a-Pinene | $C_{10}H_{16}H^+$ | 137.133 | 18.67 | 0.06 | 2.48 | 60.2 | 218 |
| Sesquiterpenes | $C_{15}H_{24}H^+$ | 205.195 | 26.5 | 1 | 2.96 | 20 | 97 |

Table A3 caption: Summary of reaction rate coefficients of terpenoids (isoprene, MTs, and SQTs) for OH, O₃, and NO₃ at 298 K and 1013.25 hPa air pressure at sea level. Reaction coefficients for isoprene and MTs are from Atkinson (1997) and reaction coefficients for SQTs are from Shu and Atkinson (1995). Typical values for their chemical lifetimes during the day ($\tau_{c,day}$), night ($\tau_{c,night}$), as well as the ratio of these day-night timescales ($\tau_{c,day}/\tau_{c,night}$) are also included. Ambient day and night OH, O₃, and NO₃ concentrations for the calculation of $\tau_{c,day}$ and $\tau_{c,night}$, are based on the modeled values used for the chemical degradation analysis. For comparison, typical values of the turbulent mixing timescale in the canopy during day and night ($\tau_{t,day}$ and $\tau_{t,night}$, respectively) are ca. $\tau_{t,day}$ = 40 s, $\tau_{t,night}$ = ca. 1.5 min, and $\tau_{t,day}/\tau_{t,night}$ = 0.4 – 0.5.

| type | compound | $K_{OH}$ (cm$^3$ molecule$^{-1}$ s$^{-1}$) | $K_{ozone}$ (cm$^3$ molecule$^{-1}$ s$^{-1}$) | $K_{NO_3}$ (cm$^3$ molecule$^{-1}$ s$^{-1}$) | $\tau_{c,day}$ (s)[a] | $\tau_{c,night}$ (s)[b] | $\tau_{c,day}/\tau_{c,night}$ |
|---|---|---|---|---|---|---|---|
| ISO | isoprene | $1.01 \times 10^{-10}$ | $1.28 \times 10^{-17}$ | $6.78 \times 10^{-13}$ | 2.1 hr | 5.2 hr | 0.4 |
| MT | α-pinene | $5.37 \times 10^{-11}$ | $8.66 \times 10^{-17}$ | $6.16 \times 10^{-12}$ | 1.8 hr | 1.3 hr | 1.4 |
| MT | Δ³-carene | $8.80 \times 10^{-11}$ | $3.70 \times 10^{-17}$ | $9.10 \times 10^{-12}$ | 1.9 hr | 1 hr | 1.9 |
| MT | myrcene | $2.20 \times 10^{-10}$ | $4.70 \times 10^{-16}$ | $1.10 \times 10^{-11}$ | 23 min | 29 min | 0.8 |
| MT | β-pinene | $7.90 \times 10^{-11}$ | $1.50 \times 10^{-17}$ | $2.51 \times 10^{-12}$ | 2.5 hr | 3 hr | 0.9 |
| MT | limonene | $1.70 \times 10^{-10}$ | $2.00 \times 10^{-16}$ | $1.22 \times 10^{-11}$ | 41 min | 37 min | 1.1 |
| MT | camphene | $5.30 \times 10^{-11}$ | $9.00 \times 10^{-19}$ | $6.60 \times 10^{-13}$ | 4.3 hr | 8.5 hr | 0.5 |
| MT | sabinene | $1.17 \times 10^{-10}$ | $8.60 \times 10^{-17}$ | $1.00 \times 10^{-11}$ | 1.2 hr | 51 min | 1.4 |
| SQT | longifolene | $4.70 \times 10^{-11}$ | $<5.00 \times 10^{-19}$ | $6.80 \times 10^{-13}$ | >4.9 hr | >8.9 hr | 0.6 |
| SQT | α-humulene | $2.80 \times 10^{-11}$ | $1.17 \times 10^{-14}$ | $3.50 \times 10^{-11}$ | 1.4 min | 2.5 min | 0.6 |
| SQT | β-caryophyllene | $2.00 \times 10^{-10}$ | $1.16 \times 10^{-14}$ | $1.90 \times 10^{-11}$ | 1.4 min | 2.6 min | 0.5 |

[a](daytime): OH and ozone concentrations are taken to be $1.2 \times 10^6$ molecules cm$^{-3}$ and $1 \times 10^{12}$ molecules cm$^{-3}$ (ca. 0.05 ppt and 40 ppb), respectively. [b](nighttime): OH and ozone concentrations are taken to be $3 \times 10^5$ molecules cm$^{-3}$ (ppt) and $5 \times 10^{11}$ molecules cm$^{-3}$ (ca 0.01 ppt and 20 ppb), respectively. Nighttime NO₃: $2.5 \times 10^7$ molecules cm$^{-3}$ (ca. 2.5 ppt). [c]Based on $\tau_t = z/u_*$ (e.g., Rinne et al 2012) at inlet height (35 m) ypical friction velocities $u_*$ observed during day and night (ca. 0.65 and 0.3 m s$^{-1}$, respectively) for the 2020 Norunda campaign.

| $\tau_{t,day}$ (s)[c] | $\tau_{t,night}$ (s)[c] | $\tau_{t,day}/\tau_{c,night}$ |
|---|---|---|
| 40 s | 1.5 min | 0.4 – 0.5 |

## Appendix References:

Atkinson, R.: Gas-phase tropospheric chemistry of volatile organic compounds: 1. Alkanes and alkenes, Journal of Physical and Chemical Reference Data, 26, 215-290, 1997.

Atkinson, R.: Atmospheric chemistry of VOCs and NOx, Atmos. Environ., 34, 2063-2101, 2000.

Atkinson, R. and Arey, J.: Gas-phase tropospheric chemistry of biogenic volatile organic compounds: a review, Atmos. Environ., 37, 197-219, 2003.

Aubinet, M., Vesala, T., and Papale, D.: Eddy covariance: a practical guide to measurement and data analysis, Springer Science & Business Media2012a.

Aubinet, M., Feigenwinter, C., Heinesch, B., Laffineur, Q., Papale, D., Reichstein, M., Rinne, J., and Van Gorsel, E.: Nighttime Flux Correction, in: Eddy Covariance: A Practical Guide to Measurement and Data Analysis, edited by: Aubinet, M., Vesala, T., and Papale, D., Springer Netherlands, Dordrecht, 133-157, 10.1007/978-94-007-2351-1_5, 2012b.

Bäck, J., Aalto, J., Henriksson, M., Hakola, H., He, Q., and Boy, M.: Chemodiversity of a Scots pine stand and implications for terpene air concentrations, Biogeosciences, 9, 689-702, 2012.

Bai, J., Guenther, A., Turnipseed, A., Duhl, T., Yu, S., and Wang, B.: Seasonal variations in whole-ecosystem BVOC emissions from a subtropical bamboo plantation in China, Atmos. Environ., 124, 12-21, 2016.

Bianchi, F., Tröstl, J., Junninen, H., Frege, C., Henne, S., Hoyle, C. R., Molteni, U., Herrmann, E., Adamov, A., and
Bukowiecki, N.: New particle formation in the free troposphere: A question of chemistry and timing, Science, 352, 1109-1112, 2016.

Bonn, B., Boy, M., Kulmala, M., Groth, A., Trawny, K., Borchert, S., and Jacobi, S.: A new parametrization for ambient particle formation over coniferous forests and its potential implications for the future, Atmos. Chem. Phys., 9, 8079-8090, 2009.

Borsdorf, H., Bentele, M., Müller, M., Rebmann, C., and Mayer, T.: Comparison of Seasonal and Diurnal Concentration Profiles of BVOCs in Coniferous and Deciduous Forests, Atmos, 14, 1347, 2023.

Boucher, O., Randall, D., Artaxo, P., Bretherton, C., Feingold, G., Forster, P., Kerminen, V.-M., Kondo, Y., Liao, H., and Lohmann, U.: Clouds and aerosols, in: Climate change 2013: the physical science basis. Contribution of Working Group I to the Fifth Assessment Report of the Intergovernmental Panel on Climate Change, Cambridge University Press, 571-658, 2013.

Breitenlechner, M., Fischer, L., Hainer, M., Heinritzi, M., Curtius, J., and Hansel, A.: PTR3: an instrument for studying the lifecycle of reactive organic carbon in the atmosphere, Analytical chemistry, 89, 5824-5831, 2017.

Businger, J. A., Wyngaard, J. C., Izumi, Y., and Bradley, E. F.: Flux-profile relationships in the atmospheric surface layer, Journal of Atmospheric Sciences, 28, 181-189, 1971.

Calogirou, A., Larsen, B. R., Brussol, C., Duane, M., and Kotzias, D.: Decomposition of terpenes by ozone during sampling
on Tenax, Analytical Chemistry, 68, 1499-1506, 1996.

Cellier, P. and Brunet, Y.: Flux-gradient relationships above tall plant canopies, Agr. Forest Met., 58, 93-117, 1992.

Duhl, T. R., Helmig, D., and Guenther, A.: Sesquiterpene emissions from vegetation: a review, Biogeosciences, 5, 761-777, 10.5194/bg-5-761-2008, 2008.

Duhl, T. R., Gochis, D., Guenther, A., Ferrenberg, S., and Pendall, E.: Emissions of BVOC from lodgepole pine in response
to mountain pine beetle attack in high and low mortality forest stands, Biogeosciences, 10, 483, 2013.

Dyer, A. J.: A review of flux-profile relationships, Boundary Layer Meteorol., 7, 363-372, 1974.

Fall, R.: Biogenic emissions of volatile organic compounds from higher plants, in: Reactive hydrocarbons in the atmosphere, Elsevier, 41-96, 1999.

Fazu, C. and Schwerdtfeger, P.: Flux-gradient relationships for momentum and heat over a rough natural surface, QJRMS,
115, 335-352, 1989.

Fehsenfeld, F., Calvert, J., Fall, R., Goldan, P., Guenther, A. B., Hewitt, C. N., Lamb, B., Liu, S., Trainer, M., and Westberg, H.: Emissions of volatile organic compounds from vegetation and the implications for atmospheric chemistry, Global Biogeochem. Cy., 6, 389-430, 1992.

Fischer, L., Breitenlechner, M., Canaval, E., Scholz, W., Striednig, M., Graus, M., Karl, T. G., Petäjä, T., Kulmala, M., and Hansel, A.: First eddy covariance flux measurements of semi-volatile organic compounds with the PTR3-TOF-MS, Atmospheric Measurement Techniques, 14, 8019-8039, 2021.

Friedman, B. and Farmer, D. K.: SOA and gas phase organic acid yields from the sequential photooxidation of seven monoterpenes, Atmos. Environ., 187, 335-345, 2018.

Fuentes, J., Wang, D., Neumann, H., Gillespie, T., Den Hartog, G., and Dann, T.: Ambient biogenic hydrocarbons and isoprene emissions from a mixed deciduous forest, J. Atmos. Chem., 25, 67-95, 1996.

Garratt, J.: Surface influence upon vertical profiles in the atmospheric near-surface layer, QJRMS, 106, 803-819, 1980.

Geron, C., Rasmussen, R., Arnts, R. R., and Guenther, A.: A review and synthesis of monoterpene speciation from forests in the United States, Atmos. Environ., 34, 1761-1781, 2000.

Ghirardo, A., Koch, K., Taipale, R., Zimmer, I., Schnitzler, J.-P., and Rinne, J.: Determination of de novo and pool emissions of terpenes from four common boreal/alpine trees by $^{13}CO_2$ labelling and PTR-MS analysis, Plant Cell Environ., 33, 781-792, 2010.

Goldstein, A. H., Daube, B., Munger, J., and Wofsy, S.: Automated in-situ monitoring of atmospheric non-methane hydrocarbon concentrations and gradients, J. Atmos. Chem., 21, 43-59, 1995.

Griffin, R. J., Cocker III, D. R., Flagan, R. C., and Seinfeld, J. H.: Organic aerosol formation from the oxidation of biogenic hydrocarbons, J. Geophys. Res. Atmos., 104, 3555-3567, 1999.

Guenther, A.: Upscaling biogenic VOC emissions from leaves to landscapes, Biology, Controls and Models of Tree Volatile Organic Compound Emissions, edited by: Niinemets, U. and Monson, R., Springer Tree Physiology series, 2012.

Guenther, A., Jiang, X., Heald, C. L., Sakulyanontvittaya, T., Duhl, T. a., Emmons, L., and Wang, X.: The Model of Emissions of Gases and Aerosols from Nature version 2.1 (MEGAN2. 1): an extended and updated framework for modeling biogenic emissions, Geosci. Instrum. Dev., 5, 1471-1492, 2012.

Guenther, A., Baugh, W., Davis, K., Hampton, G., Harley, P., Klinger, L., Vierling, L., Zimmerman, P., Allwine, E., and Dilts, S.: Isoprene fluxes measured by enclosure, relaxed eddy accumulation, surface layer gradient, mixed layer gradient, and mixed layer mass balance techniques, J. Geophys. Res. Atmos., 101, 18555-18567, 1996.

Guenther, A., Hewitt, C. N., Erickson, D., Fall, R., Geron, C., Graedel, T., Harley, P., Klinger, L., Lerdau, M., and McKay, W.: A global model of natural volatile organic compound emissions, J. Geophys. Res., 100, 8873-8892, 1995.

Guenther, A. B., Zimmerman, P. R., Harley, P. C., Monson, R. K., and Fall, R.: Isoprene and monoterpene emission rate variability: model evaluations and sensitivity analyses, J. Geophys. Res. Atmos., 98, 12609-12617, 1993.

Hakola, H., Hellén, H., Hemmilä, M., Rinne, J., and Kulmala, M.: In situ measurements of volatile organic compounds in a boreal forest, Atmos. Chem. Phys., 12, 11665-11678, 10.5194/acp-12-11665-2012, 2012.

Hakola, H., Tarvainen, V., Bäck, J., Ranta, H., Bonn, B., Rinne, J., and Kulmala, M.: Seasonal variation of mono-and sesquiterpene emission rates of Scots pine, Biogeosciences, 3, 93-101, 2006.

Hakola, H., Tarvainen, V., Praplan, A. P., Jaars, K., Hemmilä, M., Kulmala, M., Bäck, J., and Hellén, H.: Terpenoid and carbonyl emissions from Norway spruce in Finland during the growing season, Atmos. Chem. Phys., 17, 3357-3370, 2017.

Hallquist, M., Wenger, J. C., Baltensperger, U., Rudich, Y., Simpson, D., Claeys, M., Dommen, J., Donahue, N., George, C., and Goldstein, A.: The formation, properties and impact of secondary organic aerosol: current and emerging issues, Atmos. Chem. Phys., 9, 5155-5236, 2009.

Harman, I. N. and Finnigan, J. J.: A simple unified theory for flow in the canopy and roughness sublayer, Boundary Layer Meteorol., 123, 339-363, 10.1007/s10546-006-9145-6, 2007.

Heard, D. E.: Atmospheric field measurements of the hydroxyl radical using laser-induced fluorescence spectroscopy, Annu. Rev. Phys. Chem., 57, 191-216, 2006.

Heard, D. E. and Pilling, M. J.: Measurement of OH and HO2 in the Troposphere, Chem. Rev., 103, 5163-5198, 10.1021/cr020522s, 2003.

Heinritzi, M., Dada, L., Simon, M., Stolzenburg, D., Wagner, A. C., Fischer, L., Ahonen, L. R., Amanatidis, S., Baalbaki, R., Baccarini, A., Bauer, P. S., Baumgartner, B., Bianchi, F., Brilke, S., Chen, D., Chiu, R., Dias, A., Dommen, J., Duplissy, J., Finkenzeller, H., Frege, C., Fuchs, C., Garmash, O., Gordon, H., Granzin, M., El Haddad, I., He, X., Helm, J., Hofbauer, V., Hoyle, C. R., Kangasluoma, J., Keber, T., Kim, C., Kürten, A., Lamkaddam, H., Laurila, T. M., Lampilahti, J., Lee, C. P., Lehtipalo, K., Leiminger, M., Mai, H., Makhmutov, V., Manninen, H. E., Marten, R., Mathot, S., Mauldin, R. L., Mentler, B., Molteni, U., Müller, T., Nie, W., Nieminen, T., Onnela, A., Partoll, E., Passananti, M., Petäjä, T., Pfeifer, J., Pospisilova, V.,

Quélever, L. L. J., Rissanen, M. P., Rose, C., Schobesberger, S., Scholz, W., Scholze, K., Sipilä, M., Steiner, G., Stozhkov, Y., Tauber, C., Tham, Y. J., Vazquez-Pufleau, M., Virtanen, A., Vogel, A. L., Volkamer, R., Wagner, R., Wang, M., Weitz, L., Wimmer, D., Xiao, M., Yan, C., Ye, P., Zha, Q., Zhou, X., Amorim, A., Baltensperger, U., Hansel, A., Kulmala, M., Tomé, A., Winkler, P. M., Worsnop, D. R., Donahue, N. M., Kirkby, J., and Curtius, J.: Molecular understanding of the suppression of new-particle formation by isoprene, Atmos. Chem. Phys., 20, 11809-11821, 10.5194/acp-20-11809-2020, 2020.

Heiskanen, J., Brümmer, C., Buchmann, N., Calfapietra, C., Chen, H., Gielen, B., Gkritzalis, T., Hammer, S., Hartman, S., and Herbst, M.: The integrated carbon observation system in Europe, Bulletin of the American Meteorological Society, 103, E855-E872, 2022.

Hellén, H., Praplan, A. P., Tykkä, T., Ylivinkka, I., Vakkari, V., Bäck, J., Petäjä, T., Kulmala, M., and Hakola, H.: Long-term measurements of volatile organic compounds highlight the importance of sesquiterpenes for the atmospheric chemistry of a boreal forest, Atmos. Chem. Phys., 18, 13839-13863, 10.5194/acp-18-13839-2018, 2018.

Helmig, D., Ortega, J., Guenther, A., Herrick, J. D., and Geron, C.: Sesquiterpene emissions from loblolly pine and their potential contribution to biogenic aerosol formation in the Southeastern US, Atmos. Environ., 40, 4150-4157, 2006.

Hiltunen, R. and Laakso, I.: Gas chromatographic analysis and biogenetic relationships of monoterpene enantiomers in Scots pine and juniper needle oils, Flavour Fragrance J., 10, 203-210, 1995.

Hodzic, A., Kasibhatla, P. S., Jo, D. S., Cappa, C. D., Jimenez, J. L., Madronich, S., and Park, R. J.: Rethinking the global secondary organic aerosol (SOA) budget: stronger production, faster removal, shorter lifetime, Atmos. Chem. Phys., 16, 7917-7941, 2016.

Högström, U., Bergström, H., Smedman, A.-S., Halldin, S., and Lindroth, A.: Turbulent exchange above a pine forest, I: Fluxes and gradients, Boundary Layer Meteorol., 49, 197-217, 1989.

Holzke, C., Dindorf, T., Kesselmeier, J., Kuhn, U., and Koppmann, R.: Terpene emissions from European beech (shape Fagus sylvatica~ L.): pattern and Emission Behaviour Over two Vegetation Periods, J. Atmos. Chem., 55, 81-102, 2006.

Horst, T. W.: The Footprint for Estimation of Atmosphere-Surface Exchange Fluxes by Profile Techniques, Boundary Layer Meteorol., 90, 171-188, 10.1023/A:1001774726067, 1999.

Janson, R. W.: Monoterpene emissions from Scots pine and Norwegian spruce, J. Geophys. Res. Atmos., 98, 2839-2850, 1993.

Jensen, A. R., Koss, A. R., Hales, R. B., and de Gouw, J. A.: Measurements of volatile organic compounds in ambient air by gas-chromatography and real-time Vocus PTR-TOF-MS: calibrations, instrument background corrections, and introducing a PTR Data Toolkit, Atmospheric Measurement Techniques, 16, 5261-5285, 2023.

Kajos, M., Rantala, P., Hill, M., Hellén, H., Aalto, J., Patokoski, J., Taipale, R., Hoerger, C., Reimann, S., and Ruuskanen, T.: Ambient measurements of aromatic and oxidized VOCs by PTR-MS and GC-MS: intercomparison between four instruments in a boreal forest in Finland, Atmospheric Measurement Techniques, 8, 4453-4473, 2015.

Kaplan, J. O., Folberth, G., and Hauglustaine, D. A.: Role of methane and biogenic volatile organic compound sources in late glacial and Holocene fluctuations of atmospheric methane concentrations, Global Biogeochem. Cy., 20, 2006.

Karl, T., Potosnak, M., Guenther, A., Clark, D., Walker, J., Herrick, J. D., and Geron, C.: Exchange processes of volatile organic compounds above a tropical rain forest: Implications for modeling tropospheric chemistry above dense vegetation, J. Geophys. Res. Atmos., 109, 2004.

Kiendler-Scharr, A., Wildt, J., Maso, M. D., Hohaus, T., Kleist, E., Mentel, T. F., Tillmann, R., Uerlings, R., Schurr, U., and Wahner, A.: New particle formation in forests inhibited by isoprene emissions, Nature, 461, 381-384, 10.1038/nature08292, 2009.

Kim, D., Stevens, P. S., and Hites, R. A.: Rate constants for the gas-phase reactions of OH and O3 with β-ocimene, β-myrcene, and α-and β-farnesene as a function of temperature, The Journal of Physical Chemistry A, 115, 500-506, 2011.

Kirkby, J., Duplissy, J., Sengupta, K., Frege, C., Gordon, H., Williamson, C., Heinritzi, M., Simon, M., Yan, C., Almeida, J., Tröstl, J., Nieminen, T., Ortega, I. K., Wagner, R., Adamov, A., Amorim, A., Bernhammer, A.-K., Bianchi, F., Breitenlechner, M., Brilke, S., Chen, X., Craven, J., Dias, A., Ehrhart, S., Flagan, R. C., Franchin, A., Fuchs, C., Guida, R., Hakala, J., Hoyle, C. R., Jokinen, T., Junninen, H., Kangasluoma, J., Kim, J., Krapf, M., Kürten, A., Laaksonen, A., Lehtipalo, K., Makhmutov, V., Mathot, S., Molteni, U., Onnela, A., Peräkylä, O., Piel, F., Petäjä, T., Praplan, A. P., Pringle, K., Rap, A., Richards, N. A. D., Riipinen, I., Rissanen, M. P., Rondo, L., Sarnela, N., Schobesberger, S., Scott, C. E., Seinfeld, J. H., Sipilä, M., Steiner, G., Stozhkov, Y., Stratmann, F., Tomé, A., Virtanen, A., Vogel, A. L., Wagner, A. C., Wagner, P. E., Weingartner, E., Wimmer, D., Winkler, P. M., Ye, P., Zhang, X., Hansel, A., Dommen, J., Donahue, N. M., Worsnop, D. R., Baltensperger, U.,

Kulmala, M., Carslaw, K. S., and Curtius, J.: Ion-induced nucleation of pure biogenic particles, Nature, 533, 521-526, 10.1038/nature17953, 2016.

Kljun, N., Calanca, P., Rotach, M., and Schmid, H.: A simple parameterisation for flux footprint predictions, Boundary Layer Meteorol., 112, 503-523, 2004.

Kljun, N., Calanca, P., Rotach, M., and Schmid, H. P.: A simple two-dimensional parameterisation for Flux Footprint Prediction (FFP), Geosci. Instrum. Dev., 8, 3695-3713, 2015.

Komenda, M. and Koppmann, R.: Monoterpene emissions from Scots pine (Pinus sylvestris): field studies of emission rate 1410 variabilities, J. Geophys. Res. Atmos., 107, ACH 1-1-ACH 1-13, 2002.

Krechmer, J., Lopez-Hilfiker, F., Koss, A., Hutterli, M., Stoermer, C., Deming, B., Kimmel, J., Warneke, C., Holzinger, R., and Jayne, J.: Evaluation of a new reagent-ion source and focusing ion–molecule reactor for use in proton-transfer-reaction mass spectrometry, Analytical chemistry, 90, 12011-12018, 2018.

Kulmala, M., Suni, T., Lehtinen, K., Dal Maso, M., Boy, M., Reissell, A., Rannik, Ü., Aalto, P., Keronen, P., and Hakola, H.: 1415 A new feedback mechanism linking forests, aerosols, and climate, Atmos. Chem. Phys., 4, 557-562, 2004.

Lagergren, F., Eklundh, L., Grelle, A., Lundblad, M., Mölder, M., Lankreijer, H., and Lindroth, A.: Net primary production and light use efficiency in a mixed coniferous forest in Sweden, Plant Cell Environ., 28, 412-423, 2005.

Lee, A., Goldstein, A. H., Kroll, J. H., Ng, N. L., Varutbangkul, V., Flagan, R. C., and Seinfeld, J. H.: Gas-phase products and secondary aerosol yields from the photooxidation of 16 different terpenes, J. Geophys. Res. Atmos., 111, 2006a.

Lee, A., Goldstein, A. H., Keywood, M. D., Gao, S., Varutbangkul, V., Bahreini, R., Ng, N. L., Flagan, R. C., and Seinfeld, J. H.: Gas-phase products and secondary aerosol yields from the ozonolysis of ten different terpenes, J. Geophys. Res. Atmos., 111, 2006b.

Lindfors, V. and Laurila, T.: Biogenic volatile organic compound (VOC) emissions from forests in Finland, Boreal Environ. Res., 5, 95-113, 2000.

Lindinger, W., Hansel, A., and Jordan, A.: On-line monitoring of volatile organic compounds at pptv levels by means of proton-transfer-reaction mass spectrometry (PTR-MS) medical applications, food control and environmental research, Int. J. Mass Spectrom. Ion Proc., 173, 191-241, 1998.

Manninen, A.-M., Tarhanen, S., Vuorinen, M., and Kainulainen, P.: Comparing the variation of needle and wood terpenoids in Scots pine provenances, Journal of chemical ecology, 28, 211-228, 2002.

Materić, D., Lanza, M., Sulzer, P., Herbig, J., Bruhn, D., Turner, C., Mason, N., and Gauci, V.: Monoterpene separation by coupling proton transfer reaction time-of-flight mass spectrometry with fastGC, Analytical and bioanalytical chemistry, 407, 7757-7763, 2015.

McFiggans, G., Mentel, T. F., Wildt, J., Pullinen, I., Kang, S., Kleist, E., Schmitt, S., Springer, M., Tillmann, R., Wu, C., Zhao, D., Hallquist, M., Faxon, C., Le Breton, M., Hallquist, Å. M., Simpson, D., Bergström, R., Jenkin, M. E., Ehn, M., Thornton, 1435 J. A., Alfarra, M. R., Bannan, T. J., Percival, C. J., Priestley, M., Topping, D., and Kiendler-Scharr, A.: Secondary organic aerosol reduced by mixture of atmospheric vapours, Nature, 565, 587-593, 10.1038/s41586-018-0871-y, 2019.

McKeen, S., Gierczak, T., Burkholder, J., Wennberg, P., Hanisco, T., Keim, E., Gao, R. S., Liu, S., Ravishankara, A., and Fahey, D.: The photochemistry of acetone in the upper troposphere: A source of odd-hydrogen radicals, Geophys. Res. Lett., 24, 3177-3180, 1997.

Mohr, C., Thornton, J. A., Heitto, A., Lopez-Hilfiker, F. D., Lutz, A., Riipinen, I., Hong, J., Donahue, N. M., Hallquist, M., Petäjä, T., Kulmala, M., and Yli-Juuti, T.: Molecular identification of organic vapors driving atmospheric nanoparticle growth, Nature Communications, 10, 4442, 10.1038/s41467-019-12473-2, 2019.

Mölder, M., Grelle, A., Lindroth, A., and Halldin, S.: Flux-profile relationships over a boreal forest—roughness sublayer corrections, Agr. Forest Met., 98, 645-658, 1999.

Monks, P. S.: Gas-phase radical chemistry in the troposphere, Chemical Society Reviews, 34, 376-395, 2005.

Müller, J. F.: Geographical distribution and seasonal variation of surface emissions and deposition velocities of atmospheric trace gases, J. Geophys. Res. Atmos., 97, 3787-3804, 1992.

Müller, M., Graus, M., Ruuskanen, T. M., Schnitzhofer, R., Bamberger, I., Kaser, L., Titzmann, T., Hörtnagl, L., Wohlfahrt, G., and Karl, T.: First eddy covariance flux measurements by PTR-TOF, Atmospheric Measurement Techniques, 3, 387, 2010.

Niinemets, Ü.: Mild versus severe stress and BVOCs: thresholds, priming and consequences, Trends Plant Sci., 15, 145-153, 2010.

Noe, S., Hüve, K., Niinemets, Ü., and Copolovici, L.: Seasonal variation in vertical volatile compounds air concentrations within a remote hemiboreal mixed forest, Atmos. Chem. Phys., 12, 3909-3926, 2012.

Pagonis, D., Sekimoto, K., and de Gouw, J.: A library of proton-transfer reactions of H3O+ ions used for trace gas detection, Journal of the American Society for Mass Spectrometry, 30, 1330-1335, 2019.

Peñuelas, J. and Staudt, M.: BVOCs and global change, Trends Plant Sci., 15, 133-144, 2010.

Persson, Y., Schurgers, G., Ekberg, A., and Holst, T.: Effects of intra-genotypic variation, variance with height and time of season on BVOC emissions, Meteorol. Z, 25, 377-388, 2016.

Petersen, R. C., Holst, T., Mölder, M., Kljun, N., and Rinne, J.: Vertical distribution of sources and sinks of volatile organic compounds within a boreal forest canopy, Atmos. Chem. Phys., 23, 7839-7858, 2023.

Pfannerstill, E. Y., Arata, C., Zhu, Q., Schulze, B. C., Woods, R., Harkins, C., Schwantes, R. H., McDonald, B. C., Seinfeld, J. H., Bucholtz, A., Cohen, R. C., and Goldstein, A. H.: Comparison between Spatially Resolved Airborne Flux Measurements and Emission Inventories of Volatile Organic Compounds in Los Angeles, Environ. Sci. Technol., 57, 15533-15545, 10.1021/acs.est.3c03162, 2023.

Pollmann, J., Ortega, J., and Helmig, D.: Analysis of atmospheric sesquiterpenes: Sampling losses and mitigation of ozone interferences, Environ. Sci. Technol., 39, 9620-9629, 2005.

Pryor, S. C., Hornsby, K. E., and Novick, K. A.: Forest canopy interactions with nucleation mode particles, Atmos. Chem. Phys., 14, 11985-11996, 10.5194/acp-14-11985-2014, 2014.

Rannik, Ü.: On the surface layer similarity at a complex forest site, J. Geophys. Res. Atmos., 103, 8685-8697, 1998.

Rantala, P., Taipale, R., Aalto, J., Kajos, M. K., Patokoski, J., Ruuskanen, T. M., and Rinne, J.: Continuous flux measurements of VOCs using PTR-MS—reliability and feasibility of disjunct-eddy-covariance, surface-layer-gradient, and surface-layer-profile methods, Boreal Environ. Res., 19, 87-107, 2014.

Read, K., Carpenter, L., Arnold, S., Beale, R., Nightingale, P., Hopkins, J., Lewis, A., Lee, J., Mendes, L., and Pickering, S.: Multiannual observations of acetone, methanol, and acetaldehyde in remote tropical Atlantic air: Implications for atmospheric OVOC budgets and oxidative capacity, Environ. Sci. Technol., 46, 11028-11039, 2012.

Riipinen, I., Yli-Juuti, T., Pierce, J. R., Petäjä, T., Worsnop, D. R., Kulmala, M., and Donahue, N. M.: The contribution of organics to atmospheric nanoparticle growth, Nature Geoscience, 5, 453-458, 10.1038/ngeo1499, 2012.

Rinne, J., Bäck, J., and Hakola, H.: Biogenic volatile organic compound emissions from the Eurasian taiga: current knowledge and future directions, Boreal Environ. Res., 14, 2009.

Rinne, J., Hakola, H., Laurila, T., and Rannik, Ü.: Canopy scale monoterpene emissions of Pinus sylvestris dominated forests, Atmos. Environ., 34, 1099-1107, 2000a.

Rinne, J., Ammann, C., Pattey, E., Paw U, K. T., and Desjardins, R. L.: Alternative Turbulent Trace Gas Flux Measurement Methods, in: Springer Handbook of Atmospheric Measurements, Springer, 1505-1530, 2021.

Rinne, J., Tuovinen, J.-P., Laurila, T., Hakola, H., Aurela, M., and Hypén, H.: Measurements of hydrocarbon fluxes by a gradient method above a northern boreal forest, Agr. Forest Met., 102, 25-37, 2000b.

Rinne, J., Markkanen, T., Ruuskanen, T., Petäjä, T., Keronen, P., Tang, M., Crowley, J., Rannik, Ü., and Vesala, T.: Effect of chemical degradation on fluxes of reactive compounds–a study with a stochastic Lagrangian transport model, Atmos. Chem. Phys., 12, 4843-4854, 2012.

Roberts, J. M., Flocke, F., Stroud, C. A., Hereid, D., Williams, E., Fehsenfeld, F., Brune, W., Martinez, M., and Harder, H.: Ground-based measurements of peroxycarboxylic nitric anhydrides (PANs) during the 1999 Southern Oxidants Study Nashville Intensive, J. Geophys. Res. Atmos., 107, ACH 1-1-ACH 1-10, 2002.

Ruuskanen, T., Taipale, R., Rinne, J., Kajos, M., Hakola, H., and Kulmala, M.: Quantitative long-term measurements of VOC concentrations by PTR-MS: annual cycle at a boreal forest site, Atmos. Chem. Phys. Disc., 9, 81-134, 2009.

Schween, J., Dlugi, R., Hewitt, C., and Foster, P.: Determination and accuracy of VOC-fluxes above the pine/oak forest at Castelporziano, Atmos. Environ., 31, 199-215, 1997.

Shu, Y. and Atkinson, R.: Atmospheric lifetimes and fates of a series of sesquiterpenes, J. Geophys. Res. Atmos., 100, 7275-7281, 1995.

Simpson, D., Winiwarter, W., Börjesson, G., Cinderby, S., Ferreiro, A., Guenther, A., Hewitt, C. N., Janson, R., Khalil, M. A. K., and Owen, S.: Inventorying emissions from nature in Europe, J. Geophys. Res. Atmos., 104, 8113-8152, 1999.

Simpson, I., Thurtell, G., Neumann, H., Den Hartog, G., and Edwards, G.: The validity of similarity theory in the roughness sublayer above forests, Boundary Layer Meteorol., 87, 69-99, 1998.

Sindelarova, K., Markova, J., Simpson, D., Huszar, P., Karlicky, J., Darras, S., and Granier, C.: High-resolution biogenic global emission inventory for the time period 2000–2019 for air quality modelling, Earth Syst. Sci. Data, 14, 251-270, 10.5194/essd-14-251-2022, 2022.

Sindelarova, K., Granier, C., Bouarar, I., Guenther, A., Tilmes, S., Stavrakou, T., Müller, J. F., Kuhn, U., Stefani, P., and Knorr, W.: Global data set of biogenic VOC emissions calculated by the MEGAN model over the last 30 years, Atmos. Chem. Phys., 14, 9317-9341, 10.5194/acp-14-9317-2014, 2014.

Singh, H. B., O'hara, D., Herlth, D., Sachse, W., Blake, D., Bradshaw, J., Kanakidou, M., and Crutzen, P.: Acetone in the atmosphere: Distribution, sources, and sinks, J. Geophys. Res. Atmos., 99, 1805-1819, 1994.

Spracklen, D. V., Bonn, B., and Carslaw, K. S.: Boreal forests, aerosols and the impacts on clouds and climate, Philosophical Transactions of the Royal Society A: Mathematical, Physical and Engineering Sciences, 366, 4613-4626, 2008.

Stolzenburg, D., Fischer, L., Vogel, A. L., Heinritzi, M., Schervish, M., Simon, M., Wagner, A. C., Dada, L., Ahonen, L. R., and Amorim, A.: Rapid growth of organic aerosol nanoparticles over a wide tropospheric temperature range, Proceedings of the National Academy of Sciences, 115, 9122-9127, 2018.

Stone, D., Whalley, L. K., and Heard, D. E.: Tropospheric OH and HO 2 radicals: field measurements and model comparisons, Chemical Society Reviews, 41, 6348-6404, 2012.

Striednig, M., Graus, M., Märk, T. D., and Karl, T. G.: InnFLUX–an open-source code for conventional and disjunct eddy covariance analysis of trace gas measurements: an urban test case, Atmospheric Measurement Techniques, 13, 1447-1465, 2020.

Taipale, R., Kajos, M. K., Patokoski, J., Rantala, P., Ruuskanen, T. M., and Rinne, J.: Role of de novo biosynthesis in ecosystem scale monoterpene emissions from a boreal Scots pine forest, Biogeosciences, 8, 2247-2255, 2011.

Tarvainen, V., Hakola, H., Hellén, H., Bäck, J., Hari, P., and Kulmala, M.: Temperature and light dependence of the VOC emissions of Scots pine, Atmos. Chem. Phys., 5, 989-998, 2005.

Tingey, D. T., Manning, M., Grothaus, L. C., and Burns, W. F.: Influence of light and temperature on monoterpene emission

rates from slash pine, Plant Physiol., 65, 797-801, 1980.

Tunved, P., Hansson, H.-C., Kerminen, V.-M., Ström, J., Dal Maso, M., Lihavainen, H., Viisanen, Y., Aalto, P. P., Komppula, M., and Kulmala, M.: High natural aerosol loading over boreal forests, Science, 312, 261-263, 2006.

Wang, M., Schurgers, G., Arneth, A., Ekberg, A., and Holst, T.: Seasonal variation in biogenic volatile organic compound (BVOC) emissions from Norway spruce in a Swedish boreal forest, Boreal Environ. Res., 22, 353-367, 2017.

Wang, M., Schurgers, G., Hellén, H., Lagergren, F., and Holst, T.: Biogenic volatile organic compound emissions from a boreal forest floor, Boreal Environ. Res., 23, 249-265, 2018.

Went, F. W.: Blue hazes in the atmosphere, Nature, 187, 641-643, 1960.

Yee, L. D., Isaacman-VanWertz, G., Wernis, R. A., Meng, M., Rivera, V., Kreisberg, N. M., Hering, S. V., Bering, M. S., Glasius, M., Upshur, M. A., Gray Bé, A., Thomson, R. J., Geiger, F. M., Offenberg, J. H., Lewandowski, M., Kourtchev, I.,

Kalberer, M., de Sá, S., Martin, S. T., Alexander, M. L., Palm, B. B., Hu, W., Campuzano-Jost, P., Day, D. A., Jimenez, J. L., Liu, Y., McKinney, K. A., Artaxo, P., Viegas, J., Manzi, A., Oliveira, M. B., de Souza, R., Machado, L. A. T., Longo, K., and Goldstein, A. H.: Observations of sesquiterpenes and their oxidation products in central Amazonia during the wet and dry seasons, Atmos. Chem. Phys., 18, 10433-10457, 10.5194/acp-18-10433-2018, 2018.

Yuan, B., Koss, A. R., Warneke, C., Coggon, M., Sekimoto, K., and de Gouw, J. A.: Proton-transfer-reaction mass

spectrometry: applications in atmospheric sciences, Chem. Rev., 117, 13187-13229, 2017.

Zhao, G., Chen, Y. Y., Holsen, T. M., and Dhaniyala, S.: Numerical simulations of the sampling performance of a large particle inlet, Journal of Aerosol Science, 90, 63-76, 10.1016/j.jaerosci.2015.08.006, 2015.