# Peer review of "BVOC and speciated monoterpene concentrations and fluxes at a"

_EGUsphere, 2024_

## Referee Comment (RC1)

**Review of**

**BVOC and speciated monoterpene concentrations and fluxes at a Scandinavian boreal forest**

by Ross Charles Petersen et al.; MS No.: egusphere-2024-3410.

**General comments**

Despite the relatively large importance of boreal forests in the global biosphere and their interaction with the atmosphere e.g. by emissions of BVOC still only few data sets provide direct BVOC flux measurements. This work contributes with a long time series of BVOC flux measurements and speciation of individual monoterpenes. It is a valuable contribution to our understanding of BVOC fluxes in boreal forests in summer and well within the scope of the journal. Therefore, I recommend publication in ACP after major revisions as indicated below.

**Specific comments**

The abstract should give the concentrations and fluxes of the most relevant BVOC observed including their uncertainties during day and night.

Define in the introduction the difference between fluxes, emissions, and potentially different emission 'surfaces'.

The results section contains several short parts without sufficient discussion or links to the other sections. Consider restructuring this to achieve a better scientific storyline. You could revise the discussion of the uncertainties e.g. by combining them in the method section so that the reader can understand the significance of the following results better. Section 3.2. contains information on the gradient method which could find place either in the method section or the appendix.

Consider adding a comparison to an emission model like MEGAN2.1 (Günter et al., 2012). This could be useful to rationalize your observations and to potentially improve the model.

Consider adding a comparison of the terpenoids with other BVOC or their sum. This would be useful for a better general understanding but also of potential shifts in emissions of different BVOC depending on changing environmental parameters. In this context also the speciation of the SQT would a useful information you could add. A comprehensive discussion on how the different environmental factors impact the different BVOC is somewhat missing.

For a comprehensive discussion of the BVOC chemistry in the forest a comparison of your observations with a model like the chemical transport model SOSAA (Zhou et al., 2017 https://doi.org/10.5194/acp-17-14309-2017, 2017) could be useful.

The discussion and implementation section should be focused on the main novel results with respect to the actual state of the art.

**Abstract**

L21 10 Hz is not high frequency; please reformulate.

L28 Quantify the major MT fluxes and relate them to the total BVOC flux and typical values in comparable forest.

L29 Specify the summer shifts in D3-carene emissions in the abstract.

**1 Introduction**

L35 Please use OH, and $NO_3$ radicals throughout the manuscript.

L43 Use peroxyacyl nitrates as general term since peroxyacetyl nitrate is a specific compound.

L46 Use BVOC and AVOC for biogenic and anthropogenic VOC and not anthropogenic BVOC (check the text further on).

L46 Try to be quantitative in your statements according to the references. Globally up to 90% of the VOC are BVOC.

L63 The sentence is unclear, please reformulate: "Isoprene is mainly involved in influencing production and lifetime of tropospheric ozone (Atkinson and Arey, 2003) and can be rather ineffective at increasing SOA yields in the troposphere with respect to MT."

L73 Specify typical chemotypes that have been identified in previous work for the Scots pine and Norway spruce.

L92 Mention also other potentially important environmental parameters (e.g. RH, soil moisture, ...) and discuss how they can impact BVOC emissions.

L98ff Be more specific on the criteria needed for good eddy covariance measurements.

L103-104 give a reference

L105 consider using "relatively fast" instead of "high frequency". Explain that you need to measure faster than the typical atmospheric fluctuations (e.g. 5 Hz of faster).

L105-107 The sentence is to long. Point out to what extend the method can be used to determine BVOC fluxes on ecosystem scales instead of one specific tower with a limited footprint. See e.g. Pfannerstill et al., *Environ. Sci. Technol.* 2023, 57, 41, 15533–15545.

L115 Add a citation for the MEGAN model.

*2 Methods*

Explain how the environmental parameters like radiation, RH or soil moisture were measured.

Figure 1: Please show the extend of trees around the tower in summer 2020 as this is hard to see on figure 3.

L168 & L173 avoid giving estimates like "ca. 4.7 s" but give the value and an uncertainty e.g.: (4.7±?) s.

L183 Explain the differences in sensitivities and e.g. relative sensitivities for isoprene, monoterpenes ,and sesquiterpenes of your calibrations vs. Jensen et al., 2023.

L189 Can you say how large the typical mass drift was within 6 minutes?

L213 Did you compare the innFLUX results with a different software? E.g. one of those participating in the recent comparison by Lan et al., AMT,17, 2649, 2024, https://amt.copernicus.org/articles/17/2649/2024/.

L219 Rather "Adsorption sampling and TD-GC-MS analysis", or?

L288 Discuss the systematic uncertainty of the gradient based flux determination employing a two point profile measurement. This should add to the uncertainty of the concentration measurement.

L289 Adsorption sampling, or?

L291 Explain all abbreviations when used first e.g. ATD-GC-MS.

L320 Is this method really resulting in landscape-scale emission information. Please be precise in the wording in the following and what you can really learn from the analysis of tower measurement on the landscape-scale.

L331 Explain how you determined the canopy temperature.

L314 Rinne et al. (2012) pointed to the large uncertainties of this methodology. How do you justify using it in a two point "gradient" analysis?

L319 Explain why you used this approach instead of comparing e.g. with the MEGAN model?

**3 Results**

L341 Give the environmental conditions like precipitation or soil water content in a quantitative manner.

L345 Give indicators for the variability.

L345-349 Give a description of the additional measurement methods (e.g. ozone) in the method section.

Figure 2: I guess the upper panel shows mixing ratios of isoprene, monoterpenes and sesquiterpenes but not of ions, or? The pie charts are hard to read. You may better plot them larger and separately. Another panel with wind speed and direction as well as precipitation would be useful. The environmental conditions for the June measurement days are not given.

Figure 2a: It would be nice to have a comparison of the MT concentrations derived with the two independent methods. Consider adding the mean total MT concentrations measured by TD-GC-MS for each of the three daytime periods by symbols with uncertainties.

Figure 2b: It would be nice to have a comparison of the MT fluxes derived with the two different methods. Consider adding the mean fluxes estimated by the SLG method for each of the three daytime periods by symbols with uncertainties.

L364/365 You didn't sample gradients. Please revise the sentence.

L365 Describe the flux footprint model in the method section.

Figure 3: The caption doesn't explain the figure. Only green lines but no red or blue. What is the grayscale background showing?

L373 The VOCUS inlet was at 35 m, or?

L392-393 Please add the information on when the different conditions were prevailing and how this has influence on the results presented.

L399 Isoprene emission should be replaced by isoprene fluxes or concentrations.

L405 Be more specific which time period you mean, the summer season, a defined campaign period or just the time for which you did the EC flux measurements?

L424-425 it would be helpful for the reader to have the estimated chemical lifetimes for SQT, MT, Isoprene.

L429 Figures 5e&h show concentrations but not fluxes. Revise the wording regarding emissions.

L432 It would be useful to add the observed behaviors of other BVOC of biological relevance, e.g. acetone, acetaldehyde, hexanol, in comparison with the isoprene, MT, and SQT in figure 5.

L436 Explain what the surface is. Did you calculate the exchange rates for canopy level emissions?

L438 Define the SER vs. E in R=F/E. Are they the same?

L440 Please explain which reactions are included. Can you identify from the off-line GC-MS analysis the major SQT species and justify if using the relatively high rate coefficient of β-caryophyllene especially with ozone for all SQT is justified?

L447 Why don't you give an average value for nighttime F/E with uncertainties as for the other cases?

Figure 7: Improve the visibility of the different lines and axis lables.

L476 Explain why you don't use e.g. the MEGAN model here for comparison.

L508 Explain how you choose the other VOCs and why you don't treat the biogenic ones in a similar way as the terpenoids before. It may be useful in terms of understand changing plant physiology with environmental conditions.

L527 Consider adding the speciated SQT here as this another interesting group of indicators for the plant physiology.

L533: Figure A4: Please give the data with the same Y-axis to allow for better comparison. It would also help the reader if the order of the compounds would be the same as in figure 9. Please add the sum of the MT values and the values from the on-line analysis (VOCUS) for the same time periods. Make clear that these are daytime (09:30-17:00) and not daily data!

L535-537 You don't give uncertainties here. I this change really significant?

L538 Consider summarizing the data in a table including the sum of the MT concentrations.

L545 Figure9: Consider adding the sum of the MT values and the values from the on-line analysis (VOCUS) for the same time periods (daytime). Make clear that these are daytime (09:30-17:00) and not daily data!

L553-555 Is this a comparison of the mean daytime values and what are the uncertainties given? Here the accuracies should be given.

Figure 10: Rescale to make the variation over daytime better visible. The sum values given in the text have higher values for the on-line measurements but the pots show the opposite.

Can you identify a significant daytime variation the relative abundancies of the individual MT or SQT?

L564 Figure 9 shows the off-line data. Please revise to figure 5. However, this value is hard to see there.

L564-590 The uncertainties discussed in this section should be given and discussed in the sections were the results are presented first or in a concise way in the method section.

**Discussion**

L594 Please give the results like '2/3rds' with uncertainties and averaging period.

L596 You did show differences in fluxes of MT e.g. in figure 9 and A4. Aren't they significant?

L601-605 To long sentence. How do your observations compare with chamber data?

L619 A chemical speciation typically doesn't influence SER, or? Please reformulate.

L620-644 The main issue of this section should be mentioned under 2.8 so that the reader understand the following results better and the details should be added to the appendix. Although you find not a big difference in using the reactivity of a-pinene for all MT for day and night this may not be the case for b-caryophyllene and all SQT for which the chemical degradation is also much more relevant. You should add a corresponding analysis to the method part (& appendix) and discuss the consequences in the discussion section.

L650ff A quantitative comparison of all major sources of uncertainty for the flux determination should be given.

**Implications**

L655-665 You measured only in one summer season and observed almost no changes in e.g. MT composition. Wouldn't the overall change of BVOC levels with temperature for changing seasons be anyway much more significant for chemical reactivity?

L666 Do you mean the reactivity of the MT mixture decreased?

L672ff The uncertainties given for the changes of oxidative capacity seem to be to small.

L681-687 As mentioned before you should give the speciation of the SQT and the corresponding composition based flux estimates in the results section.

L691-693 You didn't measure OH and $NO_3$ radicals. Please reformulate.

L695-699 To long sentence. Please revise.

**Code and data availability**

L701 Consider making e.g. the timelines of VOC data also available e.g. via a suitable database.

**Technical corrections**

L201,255,261,... Please give all equations a number starting with the first one.

L228-L230 write $MnO_2$

L315 Please use subscripts for all chemical formulars in the manuscript: $NO_3$.

L479 $E_0$

Figure 8: Give the fit functions in a better readable form. E.g. the regression coefficient & function in the plot. The parameters are already given in the text. Give the uncertainties for the temperature dependencies and add correct units on the temperature axis.

L483-485 $\beta$ in $°C^{-1}$; also in the following.

L533 Add a dot after figure 9.

L547 Figure A4 not A8.

L596 Delete 'clear' in this sentence.

L628 β-myrcene instead of b-myrcene

L637 $O_3$, $NO_3$

L695 Delete one ‚to'.

L1140ff Give the table captions on top of the tables.

---

## Author Response (AR1)

*Response to the comment of anonymous referee #1*

**Review of**

**BVOC and speciated monoterpene concentrations and fluxes at a Scandinavian boreal forest**

by Ross Charles Petersen et al.; MS No.: egusphere-2024-3410.

**General comments**

Despite the relatively large importance of boreal forests in the global biosphere and their interaction with the atmosphere e.g. by emissions of BVOC still only few data sets provide direct BVOC flux measurements. This work contributes with a long time series of BVOC flux measurements and speciation of individual monoterpenes. It is a valuable contribution to our understanding of BVOC fluxes in boreal forests in summer and well within the scope of the journal. Therefore, I recommend publication in ACP after major revisions as indicated below.

**Specific comments**

The abstract should give the concentrations and fluxes of the most relevant BVOC observed including their uncertainties during day and night.

The standard emission ($E_o$) rates (i.e., temperature-independent) may be more practical for the abstract. Aside from any chemical degradation that may be occurring, there are large day-night temperature differences that significantly affect MT and SQT emission from plant tissue storage pools (i.e., $E = E_o e^{[\beta(T-30°C)]}$ – see Eq in line L338 in previous manuscript draft and discussion in that section). Additionally, since isoprene is produced via *de novo* synthesis it has essentially no flux at night and, as noted throughout manuscript, the observed concentrations and fluxes of SQT are heavily impacted by chemical degradation. Meanwhile, the Eo and β values are much more likely to be useful to other researchers such as BVOC emission modelers. Moreover, this concentration and flux information is presented in detail in the results section. *Added $E_o$ value for MT with uncertainties*.

Define in the introduction the difference between fluxes, emissions, and potentially different emission 'surfaces'.

The relevance of chemical degradation for observed fluxes above forest canopy has been noted in the introduction of the previous draft of the manuscript (L86-L87). This point was

expanded upon in the revised manuscript. Typically, there is a correspondence between measured fluxes and emission. Full ecosystem-scale emission assessments above canopy for many trace gases (including VOCs) also frequently do not distinguish between emissions originating in the canopy from those originating from the understory vegetation, forest floor, etc.

However, the chemical degradation of short-lived VOCs below the flux measurement level can lead to significantly lower measured flux relative to the actual emission rate from the vegetation surfaces (e.g., Ciccioli et al., 1999; Rantala et al., 2014; Rinne et al., 2012; Spanke et al., 2001). Differentiating between measured flux, emission, and surface exchange becomes important and necessary when the time required for air to mix out of the canopy becomes significantly longer than chemical lifetime (i.e., $\tau_t / \tau_c \geq 1$).

*To incorporate these differences into the introduction, the following revision sentence has been incorporated into paragraph 5 of the introduction section to address these key concepts:*

*"Evaluating the ecosystem-atmosphere exchange of reactive BVOCs in the absence of chemical degradation, and hence isolating the roles of surface emission, deposition, and physical transport from its effects, represents an important goal for separating the relative influences on BVOC ecosystem-scale surface exchange and physical transport processes from atmospheric chemistry."*

The results section contains several short parts without sufficient discussion or links to the other sections. Consider restructuring this to achieve a better scientific storyline. You could revise the discussion of the uncertainties e.g. by combining them in the method section so that the reader can understand the significance of the following results better. Section 3.2. contains information on the gradient method which could find place either in the method section or the appendix.

*For following Referee #1's suggestion, several changes to the manuscript section material have been made. The description of the gradient-method uncertainty analysis (section 2.7), for example, has been expanded.*

To address this point, a test for evaluating the potential assumptions described by referee #1 with respect to the inference of total SQT emission rates from the Norunda total SQT flux measurements from the OH, Ozone, and NO3 reaction rate coefficients, has now been included in the latest revision of the manuscript.

*Such a test of potential underlying assumptions has now been included in the revised material for the manuscript appendix.*

Consider adding a comparison to an emission model like MEGAN2.1 (Günter et al., 2012). This could be useful to rationalize your observations and to potentially improve the model.

Several years (2020 – 2022) of BVOC flux data at Norunda have been collected. A direct comparison to an emission model in another work may be performed in the future.

Consider adding a comparison of the terpenoids with other BVOC or their sum. This would be useful for a better general understanding but also of potential shifts in emissions of different BVOC depending on changing environmental parameters. In this context also the speciation of the SQT would a useful information you could add. A comprehensive discussion on how the different environmental factors impact the different BVOC is somewhat missing.

While a larger comparison of the terpenoids with other VOCs was considered for this work, previous investigations at Norunda (e.g., Petersen et al 2023) have already explored this. The opportunity was taken here in this manuscript to focus in as much detail as possible on the terpenes (particularly MTs).

For a comprehensive discussion of the BVOC chemistry in the forest a comparison of your observations with a model like the chemical transport model SOSAA (Zhou et al., 2017 https://doi.org/10.5194/acp-17-14309-2017, 2017) could be useful.

The authors have read the study by Zhou et al. (2017) and have considered it while drafting this manuscript. It may be of considerable use in evaluating the 2021 data & the future 2022 Norunda BVOC forest clearcutting manuscript (upcoming).

The discussion and implementation section should be focused on the main novel results with respect to the actual state of the art.

*Further focus placed in the discussion and implication sections*. The Vocus is a cutting-edge PTR-ToF-MS instrument and allowed novel comparisons between speciated MT measurements and to perform the estimation of surface exchange rates from chemically degraded flux measurements. To our knowledge, the presented analysis for estimating the surface exchange rates of nighttime MT and SQT for a boreal forest, based on observed fluxes and the analysis of chemical degradation to produce a practical inference of the ecosystem-scale exchange rate, is unique.

**Abstract**

L21 10 Hz is not high frequency; please reformulate.

The referee is correct, as the use of the term "high frequency" varies depending on the context of its usage. For example, in radio astronomy, "high frequency" is 3-30 Megahertz.

In the context of BVOC instrumentation and specifically eddy-covariance flux data analysis (e.g., cite examples here), The Vocus PTR-ToF-MS can be considered a very "fast sampling" instrument, and the actual cycling rate of the Vocus's ToF-MS is several orders of magnitude faster than the 10 Hz-averaged spectra produced by the Vocus. Moreover, use of the term "high-frequency" for the EC flux analysis of PTR-ToF-MS BVOC measurements is relatively common for this topic (e.g., Striednig et al., 2020). However, in the case of the 10 Hz sample averaging used for the EC flux analysis, this verbage perhaps should be made very clear in the text (i.e., introduction, methods), as it is important to distinguish the usage of "high-frequency" in this case from other scientific applications. As such, we agree from the reviewer that it is best to remove the term as it is from the Abstract and rephrase in general in the text.

*To address the referee's comment, the sentence was rephrased in the abstract to omit the term "High-frequency" by replacing "High-frequency (10 Hz) measurements" with "Measurements (10 Hz sampling)". In addition, in the rest of the text, the following changes to phrasing were made at the following locations of the previous manuscript draft:*

*In terms of line numbering of previous manuscript draft -*

*L99: "high-frequency" was changed to "fast (10 – 20 Hz)"*

*L105: The sentence "The capability of PTR-ToF-MS to conduct high-frequency…" was changed to "The high sampling rate capability of PTR-ToF-MS (>10 Hz)…"*

*L199: "high-frequency" was changed to ", fast (10 – 20 Hz)"*

*L209:"high-frequency" was not changed in this instance, as it is consistent with the technical language used to discuss this type of EC flux attenuation for BVOC fluxes (e.g., Rantala et al., 2014; Striednig et al., 2020).*

L28 Quantify the major MT fluxes and relate them to the total BVOC flux and typical values in comparable forest.

Key details regarding speciated MT flux, such as the main compounds contributing to MT flux were α-pinene and $\Delta^3$-carene, and the June-August decease for 3-carene emission, have already been included.

*To provide temperature-independent context for the total MT fluxes ->The full campaign regression result for the standard emission rate (Eo) of MT (pool emission algorithm with β from Guenther et al. 2012 – for Norunda 2020, yields "386 (± 5) ng m$^{-2}$ s$^{-1}$ for β = 0.1 $^{\circ}C^{-1}$") was added to the abstract.*

Details such as flux averages have been presented within the manuscript, but as commented in the specific comments section, the abstract may not be the best place to put down the day + night values of the fluxes. (1) concentrations + fluxes would add clutter to an abstract already at the word-limit during first submission (2) there are a few practical considerations that limit the utility of such flux averages for readers. Isoprene doesn't emit at night (since de novo) so there is no nighttime flux for it to report. SQT flux is significantly influenced by chemical degradation (a component of our chemical degradation analysis, but limits the information utility for many BVOC researchers).

L29 Specify the summer shifts in D3-carene emissions in the abstract.

*To address the referee's comment, the following addition was made to the sentence:*

*", featuring a decrease in its relative fraction among observed MT compounds from June to August sampling periods,"*

**1 Introduction**

L35 Please use OH, and NO3 radicals throughout the manuscript.

Revisions made to ensure consistent "OH" and $NO_3$" radical notation throughout the manuscript.

*In the introduction section (L42 in previous manuscript draft), the text "hydroxyl radical" has been revised to "OH". Throughout the revised manuscript, the text for "$NO_3$" has been checked and revised when needed to ensure that "3" is in subscript.*

L43 Use peroxyacyl nitrates as general term since peroxyacetyl nitrate is a specific compound.

*Replaced "peroxyacetyl nitrate (PAN) compounds" with "peroxyacyl nitrates (PAN)"*

L46 Use BVOC and AVOC for biogenic and anthropogenic VOC and not anthropogenic BVOC (check the text further on).

*At L46-47, replaced "anthropogenic BVOC" with "anthropogenic VOC"*

L46 Try to be quantitative in your statements according to the references. Globally up to 90% of the VOC are BVOC.

*Following the referee's comment, the sentence was revised to the following in order to be more quantitative regarding the proportion of BVOC to AVOC for emissions globally:*

*"Globally, biogenic VOC (BVOC) emissions are several times greater than anthropogenic emissions, accounting for up to ca. 90% of total VOC emissions worldwide (Guenther et al., 1995; Müller, 1992)."*

L63 The sentence is unclear, please reformulate: "Isoprene is mainly involved in influencing production and lifetime of tropospheric ozone (Atkinson and Arey, 2003) and can be rather ineffective at increasing SOA yields in the troposphere with respect to MT."

*The sentence was revised to the following "Isoprene is mainly involved in influencing production and lifespan of tropospheric ozone (Atkinson and Arey, 2003) but is relatively ineffective at enhancing tropospheric SOA yields compared to MT."*

L73 Specify typical chemotypes that have been identified in previous work for the Scots pine and Norway spruce.

The compounds α-pinene and $\Delta^3$-carene are the chemotypes that are typically discussed for these two boreal tree species in the BVOC literature. *An additional reference for Norway spruce has been added to the sentence.*

Janson R. 1993. Monoterpene emissions from Scots pine and Norwegian spruce. J. Geophys. Res. 98: 2839-2850.

L92 Mention also other potentially important environmental parameters (e.g. RH, soil moisture, ...) and discuss how they can impact BVOC emissions.

Referee #1 is correct that common environmental parameters other than photosynthetic light and temperature can have potentially important impacts on BVOC emissions, even if under constitutive conditions they are might be not be as readily apparent in comparison. While parameters such as relative humidity vapor pressure deficit can influence stomatal conductance, for many MTs and SQTs (specifically, those with a large Henry's law constant – i.e., not very soluble in water) they are relatively insensitive compared to other more soluble BVOCs such as methanol.

There are many possible environmental parameters ($CO_2$, nutrient availability, oxidative exposure, etc) that can be potentially important for BVOC emissions, but relative brevity of this introduction paragraph it is a prudent consideration.

With this in mind -> A manner in which water availability can have a significant and distinct influence on MT and SQT emission rates is from the lack of it (i.e., drought stress*). To allude to the importance of water availability in general, while highlighting the importance of non-constitutive influences for BVOC emission models in the future, the following revision was made to the made to the sentence text:*

*Quantifying fluxes is also important for accurately parameterizing the functional dependencies of BVOC emissions on environmental parameters, such as temperature and solar radiation, **as well as non-constitutive influences on ecosystem-scale emissions such as drought and disturbance stress,** for regional and global atmospheric chemistry models (e.g. Rinne et al., 2002; Taipale et al., 2011).*

*Note: A precipitation timeseries was added to Figure 2 panel (e) as well.*

L98ff Be more specific on the criteria needed for good eddy covariance measurements.

*Highlighted that 10 – 20 Hz measurements are required to employ the full EC flux method. Also referenced Striednig et al. 2020 to the reader (directed to section 2.4 for details) for its summary of criteria for EC flux approach. Specifically, why eddy fluctuations of 0.1 – 5 Hz need to be resolved (using ≥10 Hz sampling), has been included in the introduction text.*

L103-104 give a reference

*The following references were added:* (Guenther, 2012; Peñuelas and Staudt, 2010; Rinne et al., 2009).

Guenther, A.: Upscaling biogenic VOC emissions from leaves to landscapes, Biology, Controls and Models of Tree Volatile Organic Compound Emissions, edited by: Niinemets, U. and Monson, R., Springer Tree Physiology series, 2012.

Rinne, J., Bäck, J., and Hakola, H.: Biogenic volatile organic compound emissions from the Eurasian taiga: current knowledge and future directions, Boreal Environ. Res., 14, 2009.

Peñuelas, J. and Staudt, M.: BVOCs and global change, Trends Plant Sci., 15, 133-144, 2010.

L105 consider using "relatively fast" instead of "high frequency". Explain that you need to measure faster than the typical atmospheric fluctuations (e.g. 5 Hz of faster).

*See reply to reviewer #1's comment for L21 above – sentence was revised and use of the phrase "high-frequency" in it was eliminated.*

*The following sentence was added to the revised paragraph:*

*A high sampling rate is essential to resolve fluctuations from small, short-lived eddies (0.1–5 Hz) that drive turbulent transport, as lower rates can lead to significant attenuation of measured fluxes due to unresolved turbulence.*

L105-107 The sentence is to long. Point out to what extend the method can be used to determine BVOC fluxes on ecosystem scales instead of one specific tower with a limited footprint. See e.g. Pfannerstill et al., Environ. Sci. Technol. 2023, 57, 41, 15533–15545.

The sentence was broken up into two sentences (with revised text), and a new (third) sentence was inserted between them. This new sentence addresses Reviewer #1's request

to point out the extent to which EC-based flux methods can be used to determine BVOC fluxes at larger ecosystem scales (i.e., airborne flux measurements) rather than just a single flux tower.

*The revised text is the following:*

*"The high sampling rate capability of PTR-ToF-MS (>10 Hz) makes it well-suited for measuring BVOC fluxes using the EC method. Additionally, EC-based methods utilizing PTR-ToF-MS can be implemented for various mobile platforms, including aircraft, for spatially resolved landscape-scale flux assessments over wide areas (Pfannerstill et al., 2023). When combined with the high sensitivity and accuracy of modern instrumentation (e.g., Krechmer et al., 2018), PTR-ToF-MS stands as one of the most effective tools currently available for measuring ecosystem-scale BVOC fluxes."*

L115 Add a citation for the MEGAN model.

*Added a citation for the MEGAN model (Guenther et al., 2012)*

**2 Methods**

Explain how the environmental parameters like radiation, RH or soil moisture were measured.

The measurement of these environmental parameters on the station tower is a standard part of Norunda's operations and an ICOS atmospheric and ecosystem station, and these details can be found in the ICOS station manual. *The radiation measurement used to measure canopy temperature is in the revised text.*

Figure 1: Please show the extend of trees around the tower in summer 2020 as this is hard to see on figure 3.

*Following referee #1's comment, Figure 1 has been modified to show a color-version of the canopy height map from Figure 3. This map of canopy heights (now panel 1a) shows the extent of trees (to a distance of 1500 m) around the station tower. The location showing Norunda's relative position in Sweden map (now panel b) has also been remade. Finally, the diagram showing the BVOC inlet and infrastructure setup on the Norunda tower is now panel c.*

[Figure]

*To account for the new additions, the Figure 3 caption was also revised to the following:*

*"Figure 1: Forest map, station location, and BVOC inlet setup for ICOS Norunda. (a) A map of tree heights surrounding the station flux tower (out to 1500 m radially from tower base) for the Norunda forest. (b) Location and coordinates of ICOS station Norunda in Sweden. (c) BVOC inlet, infrastructure, and instrumentation setup for Vocus PTR-ToF-MS measurements on the Norunda tower (BVOC inlet at 35 m). Shown are the heights of the on-site collection of 3D sonic anemometers (blue diamonds) and BVOC inlet (red cross) at the station flux tower. The canopy top height was at approximately 28 m. Sonic-profile anemometers were located at 1.8, 4.4, 14.8, 20.8, 26.6, 29.6, 32.8, 35, 37.9, 44.8, 59.5, 74,88.5, and 101.8 m on the Norunda tower. The instrument shed contained the Vocus PTR-ToF-MS and zero-air generator for the Vocus. A blower was used to pull air through the tower inlet."*

L168 & L173 avoid giving estimates like "ca. 4.7 s" but give the value and an uncertainty e.g.: (4.7±?) s.

*In the revised manuscript text, "4.7 s" has been replaced with "4.7 ± 1 s".*

The presented time delay (4.7 ± 1 s) is based on (#1: mean) the empirically calculated value for the time delay, based on flowrate and Teflon tubing dimensions, and (#2: uncertainty) the diurnal variability (sinusoidal) of calculated lagtimes (i.e., maximizing cross-covariance functions) after regression-based correction for the gradual clock drift between Vocus and sonic datalogger clocks (e.g., Schallhart et al., 2018 - see Fig. 1 therein) over the course of the campaign.

The main source of uncertainty was found to be the variability of the flowrate over the course of the day-night cycle.

L183 Explain the differences in sensitivities and e.g. relative sensitivities for isoprene, monoterpenes ,and sesquiterpenes of your calibrations vs. Jensen et al., 2023.

The Mohr group (to whom the Vocus belongs) arrived at similar numbers for their analysis of the Norunda 2020 campaign calibration data. The Norunda 2021 campaign calibration yielded sensitivities for isoprene, MT, and SQT were much closer to (though still short of) the sensitivities of Jensen et al. (2023). The calibration in 2020 and 2021 used the same calibration standard gas bottle. The relative ratios of these sensitivities, in both years, are the same/similar, and consistent with the relative ratios presented in Jensen et al. (2023) as well. From 2020 to 2021, the fragmentation rates during calibration were also consistent.

Overall, the calibration procedure as outlined in Jensen et al. (2023) appears to have worked very well, so it is puzzling why the final sensitivies are low. To illustrate, plots of the Sexp(kptr) fitting, with the compounds present within the cal bottle corrected for measured fragmentation rates, and the transmission curve are presented below.

2020 cal. example

Several other papers present a similar magnitude for their Vocus sensitivities relative to the Norunda values as well (e.g., Li et al., 2020; Zhang et al., 2024). Based on our evaluations, it appears to have been a general attenuation of typical sensitivity performance for a Vocus. The reason for this is unknown, but it is possible that this issue is due misalignment within the Vocus following instrument transport, an issue after servicing of the instrument, etc.

*To address the quantified difference between the Norunda 2020 Vocus's sensitivity performance and typical Vocus performance illustrated in Jensen et al. 2023, the text "(attenuated)" have been added to the Table A2 caption.*

L189 Can you say how large the typical mass drift was within 6 minutes?

It really depends on the time of day and conditions. Very small in general, the main issue is that not correcting for it can interfere with the peak-fitting procedure. Attached is an

example plot from the VOCUS-data processing for one campaign day (Aug. 7) for an example peak (m/z+=107).

*Note: a small typographic correction to the text: mass scale calibration was performed once every 600 spectra (at 10 Hz sampling), hence recalibration every 1 minute (not 6).*

Figure RC1-1: example plot illustrating the Vocus-data mass-scale recalibration during processing for one campaign day (Aug. 7). Shown is an an example peak at nominal m/z+ 107. For the Norunda 2020 campaign Vocus-data processing, mass-scale recalibration was performed for every 1-minute interval. Colored peaks show the normalized signal intensity peak at 107 for each 1-minute interval of collected Vocus data on Aug 7, 2020.

L213 Did you compare the innFLUX results with a different software? E.g. one of those participating in the recent comparison by Lan et al., AMT,17, 2649, 2024, https://amt.copernicus.org/articles/17/2649/2024/.

No, InnFLUX was the only software used to process the calibrated VOCUS measurement traces for EC fluxes. InnFLUX is a flexible program based in Matlab that was originally designed specifically with PTR-MS -based (i.e., BVOC) EC flux analysis in-mind (Striednig et al., 2020), so it was deemed the leading choice for this study.

L219 Rather "Adsorption sampling and TD-GC-MS analysis", or?

Section title changed to "Adsorption sampling and ATD-GC-MS analysis", where "ATD", as described in the section, refers to automated thermal desorption.

L288 Discuss the systematic uncertainty of the gradient based flux determination employing a two point profile measurement. This should add to the uncertainty of the concentration measurement.

As referee #1 points out, identifying and quantifying systematic error sources is an important step for evaluating SLG flux uncertainties. To a certain extent, such as the sampling pump flow rate and the laboratory calibration uncertainties, this has been discussed in the previously submitted manuscript draft. Referee #2 also requested some literature-based estimates of the systematic errors associated with flux-gradient approaches such as the SLG method.

The SLG uncertainties presented in the manuscript were evaluated based on the detailed evaluation of SLG-related uncertainties for BVOC flux measurements presented in Rinne et al. 2000.

As indicated in the first paragraph of section 2.6 (i.e., $F_C = -K \frac{\delta c}{\delta z}$ for the gradient methods in general), consideration of the sources of uncertainty can be divided into those of (1) the gradient itself (i.e., the measured concentration difference $c(z_1) - c(z_2)$) and (2) the turbulent exchange coefficient K as arrived at using Monin-Obukhov (MO) similarity theory (Foken, 2006). During this evaluation, one should also take care to distinguish between random and systematic errors of (1) and (2).

While the manuscript's previous version of the section 2.7 text describes the measurement uncertainties for (1) fairly well, the analysis treatment performed for the random and systematic errors relating to (2) in the final SLG flux uncertainty estimates was not adequately described in the manuscript text. To address this point, additional information related to the uncertainties of the turbulent exchange coefficient K has been included.

A detailed breakdown of all potential sources of uncertainty for the SLG flux approach are outlined in Tabel 3 of Rinne et al. (2000)(table included below).

**Table 3**
Uncertainties of the VOC fluxes measured at Kenttärova

| Source | Correction/error estimate |
|---|---|
| 1. Hydrocarbon gradient | Monoterpene: 50 pptv→70–170% |
| Sampling and analysis | Isoprene 10 pptv→200–300% |
| 2. Turbulent exchange coefficient | |
|   2.1 Eddy covariance measurements of the buoyancy flux ($B$) and friction velocity ($u_*$) | |
|     2.1.1 Density fluctuations due to the latent heat flux | $B$: Negligible, Kaimal and Gaynor (1991); $u_*$: not affected |
|     2.1.2 Frequency response of the sensors | $B$, $u_*$: Negligible |
|     2.1.3 Spatial separation of the sensors | $B$: Not affected; $u_*$: corrected, Moore (1986) |
|     2.1.4 Line averaging of the sensors | $B$, $u_*$: corrected, Moore (1986) |
|     2.1.5 Flow distortion | Transducer shadowing corrected, Kaimal and Gaynor (1991), otherwise unknown |
|     2.1.6 Random noise of the measurements system | 2.1.6+2.1.7=20% |
|     2.1.7 Sampling error | (random) |
|   2.2 Parametrizations | |
|     2.2.1 Businger-Dyer formulas and von Kármán constant | 25% |
|     2.2.2 Displacement height | 25% |
|     2.2.3 Roughness sublayer | Corrected |

After corrections (outlined in the example table above), the main sources of error contributing to the uncertainty of SLG BVOC flux measurements by the turbulent exchange coefficient K are -

*EC measurements of buoyancy flux and friction velocity $u_*$, consisting mainly from random noise of the measurements system and sampling error (random: total ~20%).

*Variability of estimated K due to the range of values reported in the literature for the related empirical constants (treated collectively as a single parameter set) parameterizing universal gradient-flux relationships – i.e., the parametrization of the Businger-Dyer formulas and von Kármán constant used for turbulent exchange coef K incorporated into Eq. 2. (systematic: ~25%)

*As an independent parameter, the parameterization of the displacement height d. For this manuscript, the displacement height of the Norunda canopy during the 2020 campaign was evaluated using the station tower data and existing 14-level sonic anemometer profile (e.g., Rantala et al. 2014). (systematic: ~10%).

*To address the referee comments, the text of section 2.7 was revised to the following:*

*"For the two-point SLG gradient method, as alluded to by the form of $F_C = -K \frac{\delta \bar{c}}{\delta z}$ and Eq. 2, the sources of uncertainty can be divided into those from the gradient (i.e., the measured concentration difference $\bar{c}(z_1) - \bar{c}(z_2)$) and those from to the turbulent exchange coefficient K. A detailed evaluation of SLG-related uncertainties for BVOC flux measurements is presented in Rinne et al. (2000).*

*The adsorption sampling and analysis of the BVOCs represents the largest single source of uncertainty in the flux calculation. This is due to the relatively small difference in concentrations between sampling heights as compared to the uncertainty of the concentration measurements themselves. The two error sources which can be evaluated for the chemical gradient measurements are the sampling uncertainty and the analysis*

*uncertainty from the ATD-GC-MS (e.g., Kajos et al., 2015). Measurement results from the ATD-GC-MS include values for peak area mean and standard deviation, as well as signal-to-noise ratio, which were used in the uncertainty analysis. Sampling uncertainty during field measurements includes the sampling pump flow rate (typically ±5% of set flow rate), whereas sources of uncertainty in the analysis include the preconditioned tube background, as well as the ATD-GC-MS instrumental uncertainty and standard calibration uncertainty. From the combined (in quadrature) uncertainties of TD sampling and laboratory analysis, a total estimated uncertainty of ±15% is assumed for each monoterpene compound. The total uncertainty of the measured concentration difference, $\bar{c}(z_1) - \bar{c}(z_2)$, is then determined by summing the uncertainties of $\bar{c}(z_1)$ and $\bar{c}(z_2)$ in quadrature.*

*In addition to these concentration-related uncertainties, the random and systematic uncertainties associated with the turbulent exchange were also considered. Following the approach of Rinne et al. (2000), the principal uncertainties of the turbulent exchange coefficient can be further divided into those originating from the Norunda tower flux measurements used in calculating K and those arising from the parametrization of K. For the former, uncertainties in K are dominated by measurement noise and sampling error in the EC-derived friction velocity and buoyancy flux, contributing an estimated random error of ±20%. For the latter, systematic uncertainties in $K$ primarily arise from the use of universal flux–gradient relationships. While consensus exists on their functional form, a range of values for the empirical constants used to parameterize these relationships has been reported in the literature (e.g., Businger et al., 1971; Dyer, 1974; Wieringa, 1980; Högström, 1988; Oncley et al., 1996). In practice, alongside the von Kármán constant,  the constants used in parameterizing the Businger–Dyer relationships are not determined independently from each other and hence, in principle, should be treated as a single parameter set. Variability among reported parameter sets produces up to 25% systematic uncertainty in calculated estimates of K. Evaluated directly from Norunda station data (e.g., Rantala et al. 2014), the zero-plane displacement height d was treated as an independent parameter and assessed to contribute an estimated systematic error of ±10%. The final uncertainty of the SLG flux is then assessed by applying the standard propagation of error method for summing up these four key uncertainties for each SLG flux estimate.*
*"*

L289 Adsorption sampling, or?

In this sentence, "chemical" changed to "adsorption"

L291 Explain all abbreviations when used first e.g. ATD-GC-MS.

Following the referee's comment, the introduction of all abbreviations upon their first appearance was checked and included. In section 2.5, "ATD" and "GC-MS" were both defined in quick succession (sentences 1 & 3 of paragraph 3, respectively).

*To improve the clarity of the paragraph as well as introduce the full abbreviation "ATD-GC-MS", the starting text of sentence 4 was changed from "For the gas chromatography," to "For the gas chromatography portion of the ATD-GC-MS analysis,".*

L320 Is this method really resulting in landscape-scale emission information. Please be precise in the wording in the following and what you can really learn from the analysis of tower measurement on the landscape-scale.

The referee is correct that precise wording here is important. Specifically, the EC BVOC flux measurements from the tower at Norunda reflect *ecosystem*-scale information about the boreal forest there. This distinction is also important for understanding what can be learned from hybrid regression analysis of MT flux observations regarding *de novo* vs. storage-pool emission. Since true landscape-scale EC BVOC flux measurements, such as those conducted from aircraft (e.g., Pfannerstill et al., 2023), cover multiple (or at least varying) ecosystems across a wide area, applying the hybrid algorithm regression analysis to landscape-scale observations likely would muddy the result + interpretation in comparison to ecosystem-scale observations from (for example) a single boreal forest. *The text of the sentence has been revised from "landscape-scale" to "ecosystem-scale". The same change has been done at L588, L678, L686, L689 and L698.*

L331 Explain how you determined the canopy temperature.

The canopy surface temperature was measured from 55 m on the Norunda flux tower using a precision SI-111 infrared radiometer (Campbell Scientific Inc., Logan UT, USA).

*The following sentence was added to the start of section 2.2: "The canopy temperature of the Norunda forest was measured using a precision SI-111 infrared radiometer (Campbell Scientific Inc., Logan UT, USA) mounted at 55 m on the station flux tower."*

L314 Rinne et al. (2012) pointed to the large uncertainties of this methodology. How do you justify using it in a two point "gradient" analysis?

In this study, as indicated at the end of the sentence at L314, the chemical degradation analysis presented in Rinne et al. (2012) was not applied here to the gradient-method flux measurements - only the EC flux results.

The only use of it with relation to the two-height TD sampling was to investigate the impact it might have on an underpinning assumption of the SLG method (i.e., no sources/sinks between 37 and 60 m). This is described in the previous manuscript draft at L465-L474 (last paragraph of section 3.3.1). The ratio R = F/E provided by the chemical degradation analysis, for both heights (i.e., $R_1 = F_1/E$ and $R_2 = F_2/E$, since E is the same for both), provided a way to probe this assumption based on $R_2/R_1 = F_2/F_1$.

The impact on MT was negligible and, as expected, the effect of chemical degradation between the two intervening heights ($F_{2,SQT}/F_{1,SQT}$ = ca. 0.62 calculated) would have made the SLG method inapplicable to estimating SQT fluxes.

L319 Explain why you used this approach instead of comparing e.g. with the MEGAN model?

The primary goal was to investigate the standardized emission rates for Norunda, as EC PTR-ToF-MS fluxes (moreover, EC BVOC fluxes in general) had not been performed there before, despite it being a well-established boreal forest station. Moreover, the underlying emission algorithms used in the MEGAN model (e.g., Guenther et al., 1996; Guenther et al., 1995; Guenther et al., 2012; Guenther et al., 1993) are the same as those used here to investigate the standard emission rates and temperature parameter β in the manuscript.

The emission fitting investigation, however, was still performed with some consideration of MEGAN. For example, as part of the terpenoid regression fittings of $E_o$ in the results section (see 2.9), , the β coefficient values set as parameters for MT and SQT in the MEGAN model of Guenther et al. (2012) were applied (0.1 and 0.17 $°C^{-1}$ for the MTs and SQTs, respectively – see table 4 therein). This was done to investigate the pool-emission Eo regression when β was held fixed (i.e., limiting the degrees of freedom down to one variable – temperature – for the pool algorithm $E = E_o e^{[\beta(T - 30°C)]}$ to examine how well the regression performed, examine $R^2$ of the fit, and compare it to a regression where β was allowed to vary as a fitting variable as well)

**3 Results**

L341 Give the environmental conditions like precipitation or soil water content in a quantitative manner.

*The qualitative sentence regarding soil water content was removed from section 3.1 and the Norunda station precipitation data during the 2020 Norunda campaign has been included as a timeseries in Figure 2.*

L345 Give indicators for the variability.

The variability (±standard deviation) for temperature and PPFD for the campaign TD sampling periods has now been included. The reader can observe the changes in temperature and photosynthetic light by observing the 30-min temperature and PPFD timeseries data for the 2020 Norunda campaign in panel d of figure 2 (temperature shown in blue and PPFD in yellow). An additional source of information regarding diurnal variability is also provided in figure 5c, which shows the mean and standard deviation of temperature and the net radiation on a 24-hour basis for the entire 2020 campaign.

*To address the referee's comment, the mean ± sd for temperature and PPFD for the campaign TD sampling periods has been calculated and included in the text.*

L345-349 Give a description of the additional measurement methods (e.g. ozone) in the method section.

A description of the ozone instrument used by the Norunda-stenen monitoring station was added to the sentence. The new sentence is the following: "*Ozone monitoring was available, throughout the campaign from the nearby Norunda-stenen station, via a Model E400 Teledyne ozone analyzer, located 1.4 km east of the Norunda tower.*"

Figure 2: I guess the upper panel shows mixing ratios of isoprene, monoterpenes and sesquiterpenes but not of ions, or? The pie charts are hard to read. You may better plot them larger and separately. Another panel with wind speed and direction as well as precipitation would be useful. The environmental conditions for the June measurement days are not given.

Yes, the upper panel of figure 2 does show the mixing ratios as determined by gas standard calibration and ion signal traces (i.e., cps to ppbv). *The legend labeling for isoprene, MT, and SQT in the figure has been updated for clarity.*

*Figure 2 was remade with larger pie charts. In addition, a new panel showing wind speed, wind direction, and precipitation for July and August has been included as panel e.*

*To address reviewer #2's comments regarding nmol $m^{-2}$ $s^{-1}$ vs ng $m^{-2}$ $s^{-1}$, separate righthand-side y-axis scales were added to the flux panel in figure 2 for the ease of readers more accustomed to working with one or the other unit. The added scales (green, blue, red) show the mass-fluxes of isoprene, MT, and SQT (thess mass-flux axes are placed left-to-right in order of increasing mass-flux scale range – a combined righthand-side axis label for them is included in grey).*

As this figure focuses on the combined results of the TD and Vocus measurements, the June environmental conditions are not displayed, but are instead described in the section text of the manuscript.

Figure 2a: It would be nice to have a comparison of the MT concentrations derived with the two independent methods. Consider adding the mean total MT concentrations measured by TD-GC MS for each of the three daytime periods by symbols with uncertainties.

The addition of these points to Figure 2a was considered, but it was ultimately chosen not to do so based on several considerations. The addition of more visual material to the figure was considered to be cluttering, while a similar but more detailed comparison had already been included in Figure 10.

Figure 2b: It would be nice to have a comparison of the MT fluxes derived with the two different methods. Consider adding the mean fluxes estimated by the SLG method for each of the three daytime periods by symbols with uncertainties.

Similar to conclusion of comment reply above. Additionally, due to the greatly enhanced uncertainties of the SLG method, the error bars of such points if included would be extremely large and were considered to be very distracting from the Vocus EC-flux timeseries for total MT that is presented in Figure 2b.

L364/365 You didn't sample gradients. Please revise the sentence.

*In the sentence, "each TD tube sampling of the BVOC gradient" was revised to "paired TD tube sampling, used for assessing of the BVOC gradient,".*

L365 Describe the flux footprint model in the method section.

The flux footprint model was developed by Kljun et al. (2015) and is available to download online for the R programming language (https://footprint.kljun.net/).

*At the end of L365, the following sentence was included: "The Flux Footprint Prediction (FFP) is a two-dimensional parameterization for the flux footprint based on a scaling approach to its crosswind distribution (e.g., Kljun et al., 2004; Kljun et al., 2015)*

Figure 3: The caption doesn't explain the figure. Only green lines but no red or blue. What is the grayscale background showing?

The reply to referee #1's comment can be split into two parts:

(part 1 – caption/text)

The figure/caption discrepancy noted by reviewer #1 (and reviewer #2) between Figure 3 and caption text is due to an editing error during previous manuscript preparation. The figure shows the geometric-mean height footprint (at 47.1 m, footprint shown in green) for the gradient between 37 and 60 m, rather than separate flux footprints for each height.

Originally, two flux footprints were plotted in each panel for Fig 3. – red for 37 m and blue for 60 m. This was deemed less than ideal, since the paired TD sampling measured concentration at those heights - not flux. Hence, for figure 3, using either concentration footprints for 37 and 60 m, or the flux footprint at the geometric-mean height (47.1 m) of the sampled gradient, was deemed more appropriate.

In regards to using the later, as discussed in Horst (1999) for profile-based flux techniques, the use of geometric-mean height is a well-supported choice for evaluating flux footprints in two-point gradient-flux measurements. It was also observed that using two sets of footprint contours made the figure difficult to read. Therefore, it was chosen for the geometric-mean height flux footprints to be illustrated in the panels of Figure 3. *Section text and caption have been updated accordingly.*

*The section text has been revised to the following:*

*"To accurately interpret BVOC fluxes derived from concentration gradient measurements, it is important to assess the gradient-flux method footprint for the two TD sampling levels (37 m and 60 m). A flux footprint analysis at their geometric-mean height (47.1 m), as suggested by Horst (1999) for two-height gradient-profile flux estimates, was conducted for each daily period corresponding to TD tube BVOC gradient sampling on the Norunda flux tower. Each footprint was calculated using the flux footprint model developed by Kljun et al. (2015). The Flux Footprint Prediction (FFP) is a two-dimensional parameterization for the flux footprint based on a scaling approach to its crosswind distribution (e.g., Kljun et al., 2004; Kljun et al., 2015). It was found that the footprints, particularly for ca. 85th percentile and below footprint contours (depicted in Fig. 3), in general compared well with each other in terms of the forest area and composition covered. Since the geometric-mean height for the SLG estimates is above the Vocus inlet height (35 m), the total extent of the estimated SLG footprints tended to be slightly larger than the EC flux footprint. During the campaign TD measurements, approximately 90 % of the flux measured by the Vocus tower inlet at the 35 m level originated within 350 m of the tower itself. For comparison, at 47.1 m, approximately 90 % of the observations originated from within 420 m of the tower."*

*The figure caption has been revised to the following:*

*"Figure 3: Daytime (from 9:00 CEST to 17:00 CEST) average footprint estimates for SLG-derived fluxes that use the two TD BVOC sampling heights (37 and 60 m) on the Norunda flux tower. Footprint contour lines (green) are shown in 10% increments from 10% to 90%.Displayed footprints assessed at geometric-mean height (47.1 m) of TD sampling levels, following Horst (1999) for footprint estimation of SLG-method surface fluxes under unstable atmospheric stratification above-canopy (see Fig. 5f). The panels show these*

*footprints for (a-c) June 8, 9, and 10, (d-f) July 22, 23, and 24, and (g-f) August 16, 17, and 18, respectively."*

(2 – grayscale background)

The grayscale background displays the Norunda forest canopy height as determined by airborne LiDAR measurements. Air-borne LiDAR was previously acquired at ICOS Norunda by Kljun et al. with support from the British Natural Environment Research Council (NERC/ARSF/FSF grant EU10-01 and NERC/GEF grant 933).

L373 The VOCUS inlet was at 35 m, or?

The Vocus Inlet was 35 m, while the thermal desorption BVOC sampling for the gradient-method flux estimates was conducted at 37 and 60 m. *Text at L373 in the previous manuscript draft revised accordingly to "35 m".*

L392-393 Please add the information on when the different conditions were prevailing and how this has influence on the results presented.

A diurnal plot of the inverse Obukhov length, a measure of atmospheric stability, is presented in Fig. 5f.

[Figure]

As can be seen in this panel, atmospheric instability (i.e., good mixing conditions for gradient-method flux measurements) prevailed during the time periods of daytime TD sampling used for gradient-method flux estimates during the campaign. The timing of the other stability conditions is also succinctly summarized in Fig. 5f.

*The following text was added to the paragraph: "Atmospheric instability above canopy (i.e., Obukhov length $L^{-1}$ < -1000; see Fig. 5f) prevailed during the times of daytime TD sampling. Stable atmospheric conditions were generally observed at night, while near-neutral conditions were typically observed during the transition in stability following sunrise and preceding sunset."*

L399 Isoprene emission should be replaced by isoprene fluxes or concentrations.

*The first paragraph of section 3.3 was revised to more clearly distinguish in the text between isoprene concentration and isoprene flux - In the second sentence of the paragraph, "value" was changed to "concentration values", and later "isoprene emission" was replaced with "isoprene flux". In the fourth paragraph, "monoterpene" was also replaced with "MT" for consistency of abbreviations in the manuscript.*

L405 Be more specific which time period you mean, the summer season, a defined campaign period or just the time for which you did the EC flux measurements?

The Vocus was deployed from July 21 to August 27 (as indicated in the caption of Figure 5), and represents the total of Vocus measurements for the campaign. This was, however, after the June-period of TD sampling.

*To clarify in this section of text the specific period to which the diurnal plots in Fig. 5 represent, we have revised the sentence to the following:*

*"Fig. 5 shows the diurnal variation in isoprene, total MT, and total SQT from the Vocus PTR-ToF-MS measurements collected (July 21 to August 27) during the 2020 Norunda field campaign."*

L424-425 it would be helpful for the reader to have the estimated chemical lifetimes for SQT, MT, Isoprene.

In the previous manuscript draft, the range of typical chemical lifetimes of SQT compounds was already discussed at L434. However, the chemical lifetimes $\tau_c$ are also calculated as part of the chemical degradation analysis (see Appendix equation A5).

*Appendix table A3 has been developed further to feature typical values for the chemical lifetimes during the day ($\tau_{c,day}$), chemical lifetime at night ($\tau_{c,night}$), as well as the ratio of these two timescales ($\tau_{c,day}/\tau_{c,night}$). This revised table now also includes the day and night mixing timescales, as well as their ratio, to provide a point of reference for the reader when*

*considering diurnal changes in the Damköhler number Da.*

| type | compound | $k_{OH}$ (cm$^3$ molecule$^{-1}$ s$^{-1}$) | $k_{ozone}$ (cm$^3$ molecule$^{-1}$ s$^{-1}$) | $k_{NO_3}$ (cm$^3$ molecule$^{-1}$ s$^{-1}$) | $\tau_{c,day}$ (s)[a] | $\tau_{c,night}$ (s)[b] | $\tau_{c,day}$ / $\tau_{c,night}$ |
|---|---|---|---|---|---|---|---|
| ISO | isoprene | $1.01 \times 10^{-10}$ | $1.28 \times 10^{-17}$ | $6.78 \times 10^{-13}$ | 2.1 hr | 5.2 hr | 0.4 |
| MT | α-pinene | $5.37 \times 10^{-11}$ | $8.66 \times 10^{-17}$ | $6.16 \times 10^{-12}$ | 1.8 hr | 1.3 hr | 1.4 |
| MT | Δ$^3$-carene | $8.80 \times 10^{-11}$ | $3.70 \times 10^{-17}$ | $9.10 \times 10^{-12}$ | 1.9 hr | 1 hr | 1.9 |
| MT | myrcene | $2.20 \times 10^{-10}$ | $4.70 \times 10^{-16}$ | $1.10 \times 10^{-11}$ | 23 min | 29 min | 0.8 |
| MT | β-pinene | $7.90 \times 10^{-11}$ | $1.50 \times 10^{-17}$ | $2.51 \times 10^{-12}$ | 2.5 hr | 3 hr | 0.9 |
| MT | limonene | $1.70 \times 10^{-10}$ | $2.00 \times 10^{-16}$ | $1.22 \times 10^{-11}$ | 41 min | 37 min | 1.1 |
| MT | camphene | $5.30 \times 10^{-11}$ | $9.00 \times 10^{-19}$ | $6.60 \times 10^{-13}$ | 4.3 hr | 8.5 hr | 0.5 |
| MT | sabinene | $1.17 \times 10^{-10}$ | $8.60 \times 10^{-17}$ | $1.00 \times 10^{-11}$ | 1.2 hr | 51 min | 1.4 |
| SQT | longifolene | $4.70 \times 10^{-11}$ | $<5.00 \times 10^{-19}$ | $6.80 \times 10^{-13}$ | >4.9 hr | >8.9 hr | 0.6 |
| SQT | α-humulene | $2.80 \times 10^{-11}$ | $1.17 \times 10^{-14}$ | $3.50 \times 10^{-11}$ | 1.4 min | 2.5 min | 0.6 |
| SQT | β-caryophyllene | $2.00 \times 10^{-10}$ | $1.16 \times 10^{-14}$ | $1.90 \times 10^{-11}$ | 1.4 min | 2.6 min | 0.5 |

[a](daytime): OH and ozone concentrations are taken to be 1.2 x 10$^6$ molecules cm$^{-3}$ and 1 x 10$^{12}$ molecules cm$^{-3}$ (ca. 0.05 ppt and 40 ppb), respectively. [b](nighttime): OH and ozone concentrations are taken to be 3 x 10$^5$ molecules cm$^{-3}$ (ppt) and 5 x 10$^{11}$ molecules cm$^{-3}$ (ca 0.01 ppt and 20 ppb), respectively. Nighttime NO$_3$: 2.5 x 10$^7$ molecules cm$^{-3}$ (ca. 2.5 ppt). [c]Based on $\tau_t = z/u_*$ (e.g., Rinne et al 2012) at inlet height (35 m) typical friction velocities $u_*$ observed during day and night (ca. 0.65 and 0.3 m s$^{-1}$, respectively) for the 2020 Norunda campaign.

| $\tau_{t,day}$ (s)[c] | $\tau_{t,night}$ (s)[c] | $\tau_{t,day}$ / $\tau_{c,night}$ |
|---|---|---|
| 40 s | 1.5 min | 0.4 – 0.5 |

*The Table A3 caption was also revised to the following: "Table A3 caption: Summary of reaction rate coefficients of terpenoids (isoprene, MTs, and SQTs) for OH, O$_3$, and NO$_3$ at 298 K and 1013.25 hPa air pressure at sea level. Reaction coefficients for isoprene and MTs are from Atkinson (1997) and reaction coefficients for SQTs are from Shu and Atkinson (1995). Typical values for their chemical lifetimes during the day ($\tau_{c,day}$), night ($\tau_{c,night}$), as well as the ratio of these day-night timescales ($\tau_{c,day}/\tau_{c,night}$) are also included. Ambient day and night OH, O$_3$, and NO$_3$ concentrations for the calculation of $\tau_{c,day}$ and $\tau_{c,night}$, are based on the modeled values used for the chemical degradation analysis. For comparison, typical values of the turbulent mixing timescale in the canopy during day and night ($\tau_{t,day}$ and $\tau_{t,night}$, respectively) are ca. $\tau_{t,day}$ = 40 s, $\tau_{t,night}$ = ca. 1.5 min, and $\tau_{t,day}/\tau_{t,night}$ = 0.4 – 0.5.*

L429 Figures 5e&h show concentrations but not fluxes. Revise the wording regarding emissions.

Reference to Figure 5 should be for the panels displaying MT and SQT flux (d & g) and not those displaying concentration (e & h). *Figure 5 reference changed to "Fig. 5 d & g".*

L432 It would be useful to add the observed behaviors of other BVOC of biological relevance, e.g. acetone, acetaldehyde, hexanol, in comparison with the isoprene, MT, and SQT in figure 5.

Due to limited space and structure considerations, instead addressed in section 3.3.3 (Other VOCs) text in the manuscript.

However, the concentration behavior of these non-terpenoids have been considered and examined before at Norunda during previous BVOC investigations there (e.g., Petersen et al. 2023). Reference to the Norunda BVOC investigation of these non-terpenoid VOCs by Petersen et al. (2023) has already been included in the text.

While additional information regarding other, non-terpenoid VOCs was previously considered and even included in earlier drafts, this text was pared down to anchor manuscript text structure in the main focus of the campaign: isoprene, total and specific MTs (in particular), and total SQT.

L436 Explain what the surface is. Did you calculate the exchange rates for canopy level emissions?

*The following revised sentence was added to the text of the first paragraph of section 3.3.1:*

*"For ecosystem-atmosphere exchange, whereas above-canopy fluxes quantify the cumulative effect of within-canopy processes, including chemical degradation, the surface exchange rate characterizes the net emission and deposition occurring at the ecosystem's soil and vegetation surfaces in the absence of this chemical sink."*

Yes, the exchange rates were calculated using the above-canopy observed BVOC fluxes (fluxes from Vocus measurements & eddy-covariance flux method).

L438 Define the SER vs. E in R=F/E. Are they the same?

Yes, they are the same as practically-applied here in the manuscript to observed above-canopy fluxes and in text discourse of Rinne et al. 2012), but it is important to distinguish between emission (one-way; i.e., pure source) and exchange (i.e., emission – deposition).

In ideal conditions, when all assumptions made during the derivation of the flux equation from the scalar conservation equation are valid, the above-canopy flux in the absence of any chemical degradation (i.e., R =1) simply equals the surface exchange. For our measured above-canopy BVOC fluxes (where the application of the EC method also presumes the same as above), the measurement includes the impact of any deposition within-canopy alongside chemical degradation. Hence F/R implicitly describes surface exchange.

With regards to E = F/R (i.e., R = F/E) in the chemical degradation modeling conducted by Rinne et al. (2012), there is no explicit deposition sink in the model (deposition to canopy surfaces is not parameterized and the simulated particles modeling the transfer of reactive gases via the stochastic motion of air parcels are perfectly reflected at the forest floor). So in literal sense for the model, SER and E in F/E in the model are the same. In a practical sense, if the deposition rate D is constant (relatively; in the 30-minute period of an EC flux measurement), E can be considered $E_{net} = E – D = SER$, in which case emission *E* and $E_{net}$ (i.e., SER) are not equivalent.

It is also important that the label 'SER' is clearly defined/introduced in the text. To clarify this point in the manuscript text, the following revised text change was included in section

2.8, when the concept of estimating surface exchange rates is introduced with Eq. 8:  *the text "surface exchange rate (E)" has been revised to "surface exchange rate (SER)"*

L440 Please explain which reactions are included. Can you identify from the off-line GC-MS analysis the major SQT species and justify if using the relatively high rate coefficient of β caryophyllene especially with ozone for all SQT is justified?

As described in the methods and Appendix sections, the investigation of chemical degradation described in the manuscript was performed calculating the Damköhler number $Da = \tau_t / \tau_c$, where $\tau_t$ is the mixing timescale and $\tau_c$ is the chemical lifetime of the compound being considered. The Damköhler number values were then used with tabled modeling results from the Rinne et al. (2012) Supplement to lookup the resulting F/E ratio that would be expected for it based on their previous modeling.

Chemical lifetimes were calculated based OH, $O_3$, and $NO_3$ reaction rate constants using the equation

$$\tau_c = \left(k_{X,\mathrm{OH}}[\mathrm{OH}] + k_{X,\mathrm{O}_3}[\mathrm{O}_3] + k_{X,\mathrm{NO}_3}[\mathrm{NO}_3] + k_{X,photolysis}\right)^{-1},$$

where, as discussed in the following sentence of L440 in the previous draft, the OH, $O_3$, and $NO_3$ reaction rate constants presented in section 3.3.1 are those for isoprene, α-pinene, and β-caryophyllene. As chemical degradation was a late addition to the study, earlier SQT identification was not attempted during laboratory GC-MS analysis and subsequent data postprocessing. However, in previous studies (e.g., Wang et al., 2018), β-caryophyllene was identified as a major contributor at the Norunda site. It was also used in Rinne et al. (2012), which was a similar site to Norunda, being also a mixed pine/spruce forest. Moreover, β-caryophyllene is known to be significantly emitted at sites with similar pine/spruce composition. In addition, α-humulene was also common during the campaign, and has a similarly high reaction rate coefficient (hence day and night chemical lifetimes – see revised Table A3) as β-caryophyllene. Being previously observed at Norunda is large relative quantities and being consistent with its previous modeling use, it was therefore deemed acceptable to use its reaction rate constant values as a suitable input for the total SQT flux determined from the PTR-ToF-MS EC-flux measurements.

L447 Why don't you give an average value for nighttime F/E with uncertainties as for the other cases?

Previous presentation had been kept from earlier text versions through manuscript editing process. *Now expressed as "F/E = 0.90 ± 0.03".*

Figure 7: Improve the visibility of the different lines and axis lables.

*Figure 7 has been revised to improve the visibility of the different lines and axis labels.*

[Figure]

L476 Explain why you don't use e.g. the MEGAN model here for comparison.

The primary goal was to investigate the standardized emission rates for Norunda, as EC PTR-ToF-MS fluxes (moreover, EC BVOC fluxes in general) had not been performed there before, despite Norunda being a well-established boreal forest station. Moreover, the underlying emission algorithms used in the MEGAN model (e.g., Guenther et al., 1996; Guenther et al., 1995; Guenther et al., 2012; Guenther et al., 1993) are the same as those used here to investigate the standard emission rates and temperature parameter β in the manuscript.

L508 Explain how you choose the other VOCs and why you don't treat the biogenic ones in a similar way as the terpenoids before. It may be useful in terms of understand changing plant physiology with environmental conditions.

The other VOCs presented (acetaldehyde, toluene, phenol, acetic acid, hexanol, as well as methyl vinyl ketone and metharcolein (MVK+MACR)) as concentrations were chosen based on their previous inclusion in Petersen et al. 2023 and Holst et al. 2010. In each of those papers, a general evaluation of ecosystem VOC exchange using PTR-quad-MS measurements was considered. The main focus of this current manuscript was regarding isoprene, MT, and SQT, particularly their EC fluxes as well as speciated MT measurements, yet since the opportunity was presented by the capabilities of the PTR-ToF-MS to investigate these other VOCs, some information was included on them in the manuscript as well.

The observed characteristics of other VOCs, such as methanol, acetaldehyde, and acetone, are highlighted in previous investigations at Norunda (e.g., Petersen et al. 2023 – see figure 11 therein regarding water-soluble VOC sink behavior at night and examples observed from the vertical concentration profiles within and below the forest canopy). Since past work at Norunda had already probed the physiological connection to these other non-terpenoid VOCs, more emphasis was placed in this manuscript on detailed terpenoid measurements (particularly their EC flux + speciated MT) instead.

Referee #2 has noted that the section is tangential to the main aims of the paper and should be removed from the text. In an attempt to strike a balance between the comments and suggestions from both referees, the section text was reduced to a single paragraph and revised for brevity. Text was revised to the single following paragraph:

*Acetaldehyde (m/z+ = 45.034) exhibited a mean daily concentration of 0.7 ppbv. Concentrations of toluene (m/z+ = 93.07) were generally low during daytime (~12 pptv) and increased during nighttime (~30 pptv), This behavior by toluene is consistent with the build-up of anthropogenic background emissions during night in the shallow nocturnal boundary layer (Karl et al., 2004). Similar behavior was found for the mass peak at m/z+ = 95.049 (i.e., phenol), which had a concentration minimum during daytime (~9 pptv) and maximum during nighttime (~40 pptv). Acetic acid (m/z+ = 61.028) was typically lowest after sunrise (~10 pptv), gradually increasing throughout the day and peaking before sunset (~33 pptv), then declining overnight. The exception to this trend occurred when high nighttime canopy concentrations coincided with similar peaks in acetone and acetaldehyde. The dirurnal signal for m/z+ = 41.039 and m/z+ = 103.112, representing the PTR-protonated hexanol fragment and the hexanol parent ion, respectively, followed a similar pattern to acetone. The minimum in hexanol concentration (~50 pptv) typically occurred in the morning following sunrise and peaked after sunset (~130 pptv). Methyl vinyl ketone and methacrolein (MVK+MACR, m/z+ = 71.049), two important intermediate products from the photochemical oxidation of isoprene, averaged 7 pptv daily.*

L527 Consider adding the speciated SQT here as this another interesting group of indicators for the plant physiology.

The primary focus of the TD sampling was to identify MT compounds above the canopy. Reviewer #1 is correct that speciated SQT would have value. Unfortunately, extensive SQT information was not extracted during ATD-GC-MS and subsequent data postprocessing analysis. This was largely due to project planning (before a chemical degradation analysis was included) and the anticipated limitations created by open-air ozone.

Relative to the MTs, inferring highly detailed plant physiological SQT characteristics from (ozone-laden) ambient air measurements (for example, detailing the relative speciation of the SQT emission or SLG SQT flux estimation from observed (and significantly degraded) concentrations at 37 and 60 m) was considered to be a difficult proposition.

Low relative signal-to-noise for the ozone-degraded SQTs likely would have also produced cumbersomely large uncertainties.

By the time SQTs were lofted to the TD sampling heights, it was also known that they would be significantly impacted by chemical degradation (ozone, ect.). This degradation would likely mask much of the information that could otherwise be gleaned about plant physiology (at least not without first applying some *a priori* assumptions about the mixture via a chemical degradation analysis). Low relative signal-to-noise for the resulting ozone-degraded SQTs likely would have compounded uncertainties further. Previous branch chamber measurements, which scrubbed ozone from intake, had also previously been conducted at Norunda (e.g., Wang et al., 2017; Wang et al., 2018).

Therefore, following these considerations, acquiring speciated SQT data was not seen as a priority.

**However, one general observation based on the chemical degradation analysis was that the estimated total SQT standard emission rate (see Fig. 8) was roughly 1/3$^{rd}$ of that for total MT. This agrees in general with the proportion for emission potentials presented in Guenther et al. (2012)(i.e., MEGAN) for the plant functional type (see Table 2; Table 3, CLM PFT #2 therein) represented by the Norunda boreal forest. *This detail has been added to the results & discussion.*** Future work focusing specifically on SQTs and reconstructing their original speciation at time of emission may be able to glean additional information about associated plant physiological processes.

Laboratory preparation + postprocessing analysis activities, early in the project's post-campaign phase, only sought to identify two SQT compound peaks. Additional peaks might be extracted by future researchers, after significant effort, hence the raw GC files may be of interest to researchers in the future. Unfortunately it is currently beyond the scope of currently available (technical and logistical) resources. For the chemical degradation investigation, β-caryophyllene was also used - in addition to being previously chosen by Rinne et al. (2012) for their modeling work as a typical SQT example for a boreal forest of similar composition to Norunda – because of frequent observations of its dominance in past Norunda SQT emission chamber measurements.

L533: Figure A4: Please give the data with the same Y-axis to allow for better comparison. It would also help the reader if the order of the compounds would be the same as in figure 9.

Please add the sum of the MT values and the values from the on-line analysis (VOCUS) for the same time periods. Make clear that these are daytime (09:30-17:00) and not daily data!

*(1) Figure A4 has been revised to present all panels with the same y-axis scaling. The ordering of compound labels along the panel x-axes has also been revised to match that in Figure 9.*

2) This comparison of summed TD MT and total Vocus MT (for all daytime TD+Vocus sampling overlapping from 09:30-17:00) is given in figure 10.

*Following Referee #1's suggestion, to make it clear that these are daytime (09:30-17:00) and not daily data, in the figure caption the text "Daily mean" was revised to "Daily mean (09:30-17:00 CEST)".*

L535-537 You don't give uncertainties here. I this change really significant?

Yes, though the change is only weakly significant, according to ANOVA (Analysis of Variance) statistical tests.

L538 Consider summarizing the data in a table including the sum of the MT concentrations.

This modification to the data presentation was considered, but ultimately forgone as a concise visual summary of the section data was already provided in Figure 9.

L545 Figure9: Consider adding the sum of the MT values and the values from the on-line analysis (VOCUS) for the same time periods (daytime). Make clear that these are daytime (09:30-17:00) and not daily data!

The inclusion was considered, but ultimately forgone, as the same or similar information is already included in figures (i.e., Figure 10) and text elsewhere in the manuscript.

L553-555 Is this a comparison of the mean daytime values and what are the uncertainties given? Here the accuracies should be given.

Yes, these are the mean daytime values for each monthly TD sampling period. The corresponding average from the Vocus for the same daytimes each month is also presented.

The uncertainties presented are ± 1 standard deviation for the monthly sampling period. As the value is not a single sampling measurement but instead an average of many daytime measurements made each month

Figure 10: Rescale to make the variation over daytime better visible. The sum values given in the text have higher values for the on-line measurements but the pots show the opposite.

Rescaling of Fig. 10 has been done to improve visibility.

The discrepancy in the figure caption was due to an editing error. As can be noted from observing the different sampling intervals of the TD and Vocus measurements, colors in the caption description & plot were reversed. The vocus measurements (red) have 30-minute averages, while the TD-sampled concentrations (black) were conducted at 1-hour intervals.

*The caption text describing the plot colors has been corrected.* In addition, it was noted that the error bars for the summed TD MT concentration were not adequately described. While the error bars for the Vocus total MT concentration (based on 30-min average of 10 Hz sampling) show standard deviation, the error bars of the one-sample TD values show the concentration uncertainty calculated for the summed TD MT points, based on the error analysis for TD air sampling and GC-MS analysis presented in section 2.7. Caption also neglected to state that the summed TD MT points shown were collected at the lower (37 m) TD sampling height. *To address this point, the caption has been revised to the following:*

*"Comparison of MT concentration measurements from the Vocus PTR-ToF-MS (red) and the sum of speciated MT concentrations (black) from the TD sample GC-MS analysis. Summed TD MT displayed were measured at the 37 m TD sampling height. Error bars indicate (red) standard deviation of 30min Vocus MT concentration and (black) uncertainty of summed TD MT concentrations based on the error analysis for TD sampling and analysis presented in section 2.7. Shown are half-hourly timeseries of the MT concentration measurements from both Vocus and manual TD sampling for the (a-c) 2020 July 22 - 24 and (d - f) 2020 August 16 - 18 TD sample periods."*

Can you identify a significant daytime variation the relative abundancies of the individual MT or SQT?

No significant daytime variation in the relative abundancies of the MT or SQT was identified during any individual day of daytime TD sampling.

L564 Figure 9 shows the off-line data. Please revise to figure 5. However, this value is hard to see there.

*Revised to "(see Figs. 2 & 5)".* To provide another visual reference (as the campaign summary figure's panel showing total MT flux has been updated to include a ng m$^{-2}$ s$^{-1}$ scale on the righthand side), Figure 2 is indicated as well.

L564-590 The uncertainties discussed in this section should be given and discussed in the sections were the results are presented first or in a concise way in the method section.

The gradient-flux uncertainty estimates for the TD tube measurement-derived fluxes was already presented earlier in section 3.4.1 (Speciated MT concentrations and fluxes) as part of the barplot results shown in Figure 9. This point was also introduced by the text at the beginning of section 3.4.1.

**Discussion**

L594 Please give the results like '2/3rds' with uncertainties and averaging period.

*The reference to the '2/3rds' approximate proportionality of Δ3-carene relative to α-pinene has been replace with the revision of "at ca. 2/3rds of" to "(at approximately 60–70% the rate of α-pinene)".* This proportionality holds for all June, July, and August sampling periods. This trend has been noted while reviewing the literature regarding MT observations at other similar boreal forest sites. This similarity is noted in the following sentence of the paragraph. The relative proportion of the two may be an indicator of the plant physiology of these boreal forests.

L596 You did show differences in fluxes of MT e.g. in figure 9 and A4. Aren't they significant?

The statistical analysis was applied to investigate the change of each speciated MT flux across the three monthly-sampling periods (June 8-10, July 22-24, and Aug 16-18). The difference between certain compounds during a single period (a-pinene vs b-pinene, for example) *was* significantly different, for example, but the ratios of speciated MT compounds were typically consistent. Aside from 3-carene, the statistical analysis presented in the manuscript did not note a significant change between sampling months. However, the sample size was relatively small compared to the half-hourly Vocus measurements, and it is also important to note that SLG-fluxes have much more random and systematic sources of uncertainty relative to EC flux.

Check the stats code to make sure analysis was for conc. change, and not %-abundance change.

L601-605 To long sentence. How do your observations compare with chamber data?

MT observations compared well with chamber data for Norunda (e.g., Wang et al., 2017; Wang et al., 2018). *The sentence was revised to the following: "These gradient method flux measurements allow the speciated flux to be evaluated at an ecosystem-level scale, which sidesteps a potential issue for understanding ecosystem-atmosphere MT exchange, the large variability in speciated MT emissions that can exist among pine or spruce groups. Such variability, as observed in chamber measurement studies, can occur even among members of the same tree species and population (Bäck et al., 2012; Hakola et al., 2017)."*

L619 A chemical speciation typically doesn't influence SER, or? Please reformulate.

Correct, it is rather that speciation can influence the anticipated SER from observed fluxes. A PTR-based EC flux measurement alone cannot account for this (measuring only total fluxes of MT, SQT, ect) and would require speciation information from other instrumental sources (sampling for GC-MS analysis, ect.).

*To reformulate the phrasing pointed out by the referee, the first two sentences of the paragraph were replaced with the following text:*

*"The impact of MT and SQT chemical speciation on inferred SER rates should also be noted. Since the various individual MT and SQT compounds react at different rates with OH, $O_3$, and $NO_3$, these differences in turn affect the total MT and SQT fluxes that are ultimately observed above canopy using Vocus PTR-ToF-MS. Conversely, the estimation of SER rates from total MT and total SQT flux observations is therefore dependent on the relative mixture of emitted compounds considered."*

L620-644 The main issue of this section should be mentioned under 2.8 so that the reader understand the following results better and the details should be added to the appendix. Although you find not a big difference in using the reactivity of a-pinene for all MT for day and night this may not be the case for b-caryophyllene and all SQT for which the chemical degradation is also much more relevant. You should add a corresponding analysis to the method part (& appendix) and discuss the consequences in the discussion section.

Referee #1 is correct that the relative chemical composition of total SQT is an important consideration.

Potential SQT chemical lifetimes cover a very large range (during daytime, β-caryophyllene & α-humulene ~ 1.4 min, α-farnesene & β-farnesene ~ 30 min, longifolene ~ 5 hrs), and due to the significant impact that chemical degradation can have on many of these SQT compounds by the time they arrive at concentration measurement height, inferring the relative mixture of SQTs at the time of initial emission is far more difficult than it is for MT. Misapplication of the chemical degradation analysis presented can potentially result in large uncertainties.

For the results presented in section 3.3.1, the OH, ozone, and $NO_3$ reaction rate coefficients of α-pinene and β-caryophyllene were implemented to assess surface exchange rates from the measured fluxes of total MT and total SQT, respectively, obtained using the Vocus PTR-ToF-MS. Both compounds are common and frequently dominant examples of their terpenoid classes in the emissions of Norunda and similar boreal forests (cite stuff).

For total MT, the use of α-pinene, the dominant MT compound in all monthly TD sampling periods, is also suggested by the observation that the chemical lifetime and calculated

flux-to-SER ratio for α-pinene were also quite similar to the corresponding weighted averages calculated from the average mixture of MT compounds observed throughout the campaign.

For total SQT, the analysis (presented in 3.3.1 and discussed in section 2.8) made use of β-caryophyllene. In previous leaf and branch-scale chamber measurements of constitutive SQT emission at Norunda (cite) and similar pine/spruce boreal forests (cite), it has been observed that β-caryophyllene, followed by α-humulene, often dominates measured total SQT emission. As β-caryophyllene and α-humulene have quite similar chemical lifetimes (see Table A3), the reaction rate coefficient values of β-caryophyllene towards OH, ozone, and $NO_3$ were used to assess the impact of chemical degradation on total SQT flux measurements.

it is important to note, however, that the dominance of β-caryophyllene might not always be the case, particularly for non-constitutive (stressed) SQT emissions. The emission of β-farnesene has been observed to often dominate over β-caryophyllene, particularly in situations related to non-constitutive emission responses such as herbivory. While not as extreme as with longifolene, the daytime chemical lifetime of β-farnesene (ca. 16 min - see appendix) is still 11-times longer than that of β-caryophyllene. The relative mixture of SQTs emitted from a boreal forest is dependent on tree species and tree population (Bäck et al., 2012; Hakola et al., 2017), and can even be highly variable at the same site, year-to-year, due stress-related impacts such as herbivory (e.g., Wang et al. 2017). Care therefore must be taken when inferring total SQT surface exchange from measured total SQT fluxes that underlying assumptions regarding speciated mixture of SQTs emitted are correct. The impact of this is discussed in the revised appendix.

When inferring total surface exchange rates, therefore, care must be taken that the assumptions regarding speciated mixture of emitted compounds are correct, as misapplication can potentially result in large uncertainties. The impact of this is discussed in the revised appendix.

*A new section of the appendix, entitled "Impact of chemical speciation on estimated surface exchange rate uncertainty" has been added. In this new appendix material, the impact of emitted SQT mixtures on inferred total SQT exchange rates is discussed and an analysis of the impact on method uncertainty is presented.*

*The text of the 2.8 methods section was revised with the following text:*

*"The OH, $O_3$, and $NO_3$ reaction rate constants of isoprene, MT, and SQT for the chemical degradation analysis are from those reported by Atkinson (1997) and Shu and Atkinson (1995). In section 3.3.1, the reaction rate coefficients of α-pinene and β-caryophyllene were*

*implemented to assess surface exchange rates from the measured fluxes of total MT and total SQT, respectively, obtained using Vocus PTR-ToF-MS. Both compounds are common and frequently dominant examples of their terpenoid classes in the emissions of Norunda and similar boreal forests (e.g., Hakola et al., 2006; Hellén et al., 2018; Rinne et al., 2009; Rinne et al., 2012; Wang et al., 2018)." it is important to note, however, that the dominance of a particular compound in total emissions, such as β-caryophyllene among SQTs, might not always be the case, particularly for non-constitutive (stressed) emissions, such as from insect herbivory (e.g., Wang et al., 2017). Care must be taken when inferring total surface exchange rates that underlying assumptions regarding the relative mixture of emitted compounds are correct. A full description of the surface exchange rate calculations, as well as the influence of relative speciation on total exchange rate estimate uncertainties, can be found in the appendix."*

L650ff A quantitative comparison of all major sources of uncertainty for the flux determination should be given.

This has been noted in the revised text material for the section regarding gradient-method error analysis (section 2.7)

**Implications**

L655-665 You measured only in one summer season and observed almost no changes in e.g. MT composition. Wouldn't the overall change of BVOC levels with temperature for changing seasons be anyway much more significant for chemical reactivity?

Referee #1 is correct to latch onto the use of the word "significant" in this sentence, as despite effort to conduct sufficient TD sampling coverage of the 2020 campaign summer season, the total number of TD samples is relatively small.

Having only one season, and only a limited number of TD tubes available for use during the campaign to collect samples (i.e., limited sample size) limits our ability to draw conclusions, in the face sample-size constraints. Yet the consistent variability observed likely reflect seasonal changes in plant physiological emission characteristics, and motivates further investigation in the future. Referee #1 is correct that changes in BVOC levels with temperature would have an impact on chemical reactivity. This is true for MTs, for example, which in turn has a large impact on overall reactivity by increasing/decreasing MT levels (with emission rates following an exponential profile with temperature for storage emission).

However, the relative speciation of the mixture emitted will also have a compounding effect. This point is discussed in the paragraph following L655-665. As the referee notes, we

present here only one season, yet it is already clear that beta varies somewhat between MT compounds (also featured in MEGAN – e.g., Guenther 2012).

*To address referee #1's comment, "significant" has been revised to "important".*

L666 Do you mean the reactivity of the MT mixture decreased?

Not quite, as the concentrations of OH, ozone, $NO_3$, and total MT can all vary. It is also noted in the following line of the text (i.e., L 667-668 of the original manuscript) that overall MT reactivity increased from July to August due to an increase in overall MT concentration. More specifically, for the relative mixture of MT compounds observed in the TD concentration measurements on the tower, the average value of $K_{OH}$, $K_{ozone}$, and $K_{NO_3}$ each decreased.

While total MT reactivity increased from July to August, that increase is not as much as it would have been if the proportions of MT compounds to total MT observed in July had remained fixed.

for a given concentration of OH, ozone, and $NO_3$, the average chemical lifetime $\tau_c$ for the mix of observed MT compounds above the Norunda forest canopy in TD sampling (37 and 60 m) increased from the July to August daytime sampling period (July 22-24 to August 16-18). This point is elaborated upon in the following sentence of the manuscript text.

L672ff The uncertainties given for the changes of oxidative capacity seem to be to small.

**Section discusses 'relative' change in oxidative capacity due to variability of the MT mixture – i.e., July-August comparison (see reply above). Modeled daytime concentrations of OH, ozone and NO3 were held fixed at their average day or night value (depending on whether day or night were being evaluated) for each of the monthly 3-day sampling periods.**

**In this case, following the report of MT reactivity at L714 in earlier manuscript, the uncertainties were mistakenly reported as standard error rather than standard deviation. *The text of the associated ox. capacity uncertainty values have been revised from standard error to standard deviation.***

L681-687 As mentioned before you should give the speciation of the SQT and the corresponding composition based flux estimates in the results section.

This point has already been partially addressed by the reply to referee's earlier comment regarding SQT speciation. For this comment, the reply can be divided into two components:

(1) Speciating SQT measurement to estimate original speciation of SQT emission

(2) Composition-based estimates of SQT flux.

Similar as previous reply, SQT speciation data from TD sampling is very limited following previous ATD-GC-MS laboratory analysis and GC-MS data file postprocesing. Hence an estimate of speciated SQT composition emitted, which would also need to be inferred from observed speciated concentration above canopy using the chemical degradation analysis, is not possible. An estimate based on previous chamber measurements at ICOS Norunda and boreal SQT research literature was the best approach in this case.

We believe that referee #2 is referring to the emission estimate of SQT based on the chemical degradation analysis, rather than an SGL method approach.

For the latter, the issues are compounded. Due to significant chemical degradation between levels, SQT speciated fluxes using the two-height SLG approach is not possible without significant modifications to the method, as illustrated by and discussed in the section text L465 – L474 in the previous manuscript draft. To summarize the problem, one of the underlying assumptions of both gradient and profile methods for estimating flux is that there is no (i.e., chemical) loss of scalar mass during turbulent transport between the measurement heights.

L691-693 You didn't measure OH and NO3 radicals. Please reformulate.

*Corrected the sentence by removing "As demonstrated in this investigation," from the start of it. Also removed "station" from the sentence, because while instrumental OH and NO$_3$ quantification is possible, it is not necessarily part of normal/typical station measurement operations.*

L695-699 To long sentence. Please revise.

*The sentence has been revised to the following:*

*"In parallel with GC-MS or more recent tools such as ultrafast GC (e.g., Materić et al., 2015) for determining SQT speciation at the PTR eddy-covariance inlet height, combining PTR-ToF-MS eddy-covariance flux observations with chemical degradation analysis offers a promising avenue for evaluating ecosystem-scale surface exchange rates of SQTs and other highly reactive BVOC compounds."*

**Code and data availability**

L701 Consider making e.g. the timelines of VOC data also available e.g. via a suitable database.

The 30min-average VOC timeseries data for the Norunda 2020 BVOC campaign will be made available through Zenodo in the final publication.

**Technical corrections**

L201,255,261,… Please give all equations a number starting with the first one.

All equations were updated with a given number (starting with the first one). Numbering of equation references in text updated accordingly.

L228-L230 write MnO2 L315 Please use subscripts for all chemical formulars in the manuscript: NO3.

Changed 'MnO2' to 'MnO$_2$', and used subscripts for all chemical formulas ('NO3' to 'NO$_3$', 'O3' to 'O$_3$', ect) in the manuscript

L479 E$_0$

*Corrected E0 to E$_0$.*

Figure 8: Give the fit functions in a better readable form. E.g. the regression coefficient & function in the plot. The parameters are already given in the text. Give the uncertainties for the temperature dependencies and add correct units on the temperature axis.

*Figure 8 has been revised so that the regression coefficient & function are displayed within their figure panel. The uncertainties for the temperature dependencies and units along the temperature axis have been revised as well.*

L483-485 β in °C−1; also in the following.

*Changed unit label for β to $°C^{-1}$ throughout manuscript.*

L533 Add a dot after figure 9.

*Dot added after 'Fig. 9'*

L547 Figure A4 not A8.

*Text in Figure 9 caption changed from "Fig. A8" to "Fig. A4".*

L596 Delete 'clear' in this sentence.

*Deleted the word 'clear' from the sentence.*

L628 β-myrcene instead of b-myrcene

*Changed "b-myrcene" to "β-myrcene"*

L637 O3, NO3 L695 Delete one ‚to'.

*L637 - Applied subscripts for O3, NO3*

*L695 – Deleted the second instance of 'to' in the sentence*

L1140ff Give the table captions on top of the tables.

*The table captions have been moved to the top of the tables*

***Response to the comment of anonymous referee #2***

**Major comments**

Overall, the manuscript describes well the results from an interesting experiment. BVOC emissions from these boreal ecosystems are significant on both atmospheric chemistry and radiative properties, through secondary organic aerosol formation. Although not thoroughly discussed in the manuscript, BVOC emissions are coupled to chemical ecology and plant-herbivore interactions and the results will be interesting to ecologists and plant physiologists.

The methodology is interesting: it blends one of the newest and most sophisticated instruments in the field (PTR-ToFs-MS eddy covariance) with one of the oldest methods of ecosystem fluxes: the flux gradient approach with thermal desorption tubes. A strength is the comparison of these two methods. One are for improvement: there have been comparisons between gradient fluxes and EC before, specifically for BVOC fluxes. These previous studies should be further cited in the Discussion.

Referee #2 is correct to point out that previous studies have performed comparisons between gradient-flux (as well as profile-flux) methods and EC previously. For example, Rantala et al. (2014) provides excellent background and comparison of disjunct EC (i.e., a modification of typical EC that is based on slower concentration sampling rates of about 1 Hz), surface layer gradient, and surface layer profile methods with respect to BVOC measurements.

*Following referee #2's comment, additional references regarding past comparisons between these two flux methods, with specific emphasis on their application to BVOC flux measurements, have been further considered during revisions of the introduction, discussion, and implications sections.*

While the overall study and manuscript are sound, there are a number of details that need to be addressed. For example, the units of fluxes change midway through the paper. They need to be consistent. Also, there are some indications that the current manuscript was carved out of a larger work. Some comments below focus on internal inconsistencies.

Referee #2 is correct that a consistent choice of units used for the presentation of flux is important.

It has been noted that, depending on the particular research area of BVOC emission studies, it is common for researchers to use one or the other style of flux unit presented in the manuscript --- number-flux (e.g., $\text{nmol m}^{-2}\,\text{s}^{-1}$) or mass-flux (e.g., $\text{ng m}^{-2}\,\text{h}^{-1}$) --- depending on their particular BVOC research focus. For example, researchers studying

chamber-measurement emissions at the leaf and branch-level commonly use mass-flux, while EC flux studies often use number-flux units as well. Reflecting this diversity in the field, both unit types were initially included in various parts of the manuscript in an effort to make the findings intuitively accessible and reader-friendly to researchers from all BVOC subdisciplines.

In consideration of referee #2's comment, efforts have been made in recent revision work to reduce this due to complications it may inadvertently produce and to improve clarity. We have revised the manuscript to improve readability when such issues do occur. For example, so that it can be easily compared within the manuscript and research literature elsewhere, the flux timeseries data for isoprene, total MT, and total SQT displayed in Figure 2 now includes corresponding mass-flux scales for these 3 BVOC fluxes along the righthand-side y-axis.

Regarding referee #2's other observation – they are correct. It was carved out of a larger work – doctoral graduate studies – which generated a significant amount of BVOC measurement and research material. During later drafts, the focus of the manuscript was pared down from discussing multiple non-terpenoid VOCs (~15-20 compounds) to focus mostly on the isoprenoids (isoprene, MTs, and SQTs), with particularly narrow focus placed on detailed and comprehensive investigation of total and speciated MT fluxes. This narrowing of the main focus of the current version manuscript  improved the overall structure and flow of the manuscript text. Based on comments from referee #2 and balancing them with the feedback received from referee #1, these tangential elements have been further pared away, with remaining elements bearing some relevance for biological processes or anthropogenic activity (i.e., disturbance) relating to or influencing isoprene, MT, and/or SQT emission.

**Minor comments**

- Line 41: you mention acetone, but since you are not presenting results on acetone in this paper, the mention seems out of place.

  The primary focus of the manuscript is on the terpenoids, in particular MTs. As noted by referee #1, however, there are other non-terpenoid BVOCs of biological relevance (e.g., acetone, acetaldehyde, hexanol, etc.). These VOCs also have relevance for atmospheric chemistry. While fluxes of these other common VOCs is not presented, it was deemed that mention of them and a brief summary of their observed concentrations by the Vocus PTR-ToF-MS measurements (including acetone) was appropriate.

- Line 75: while the context is mostly clear, it's helpful to insert either plant or tree speciation, since you are also referring to chemical speciation.

  *To improve the clarity of this sentence, the word "chemical" was added in front of "speciation", while "tree" was added in front of subsequent use of "species".*

- Line 81: insert "chemical species-specific" before emission factors.

  *inserted "chemical species-specific" before emission factors.*

- Lines 94-111: there is a lot of great methodological information in this paragraph, but I suggest some reorganization to improve clarity. You start with PTRs, introduce PTR-ToFs, then discuss EC and go back to PTR-ToFs. Perhaps discuss PTRs and then ToFs, and next discuss EC. While many readers know the basics, BVOC papers are also read by ecologists that have more limited knowledge about the methodology.

  Referee #2 comment suggesting possible alternative modifications to the paragraph for the presentation order for PTRs, the improved PTR-ToFs, and EC flux methodology was given significant consideration. Ultimately, it was only partially implemented, as the drawbacks to the structural flow already prepared for the rest of the introduction, as well as the setup for the following paragraph laying out the main aims of the manuscript, was not sufficiently offset by the advantages.

  As introduced in the first sentence of the paragraph, the focus of the section paragraph overall is on the measurement of BVOC concentrations and (particularly) fluxes. The difficulty of the paragraph organization lies in the sequential presentation of information, leading to the outline of the advantages and key drawback of modern EC BVOC measurements (i.e., PTRs can only measure m/z+ - so, for example, total MT but not the individual MT compounds in the way that can be done with GC-MS). This leads directly into the following paragraph and key aims of the manuscript (speciated MT fluxes via the SLG gradient method). The advancements, advantages and limitations of both PTR and EC methods for BVOC flux are closely linked.

  However, to address referee #2's observations regarding non-PTR or EC-specialists, several revisions to the paragraph have been implemented (*see text*) to strengthen the conceptual links for non-specialists. For example, the reason why 10 – 20 Hz sampling is a key requirement is that a significant proportion of turbulent flux is carried by small, relatively short-lived eddies (0.1 – 5 Hz), requiring ≥10 Hz sampling ($2*f_{max}$ - i.e., Nyquist theorem) to resolve the relevant turbulent fluctuations that contribute to flux transport.

- Line 121: do you mean PTR-ToF-MS here? Be clear, since you have already made the distinction above.

  *To improve clarity, changed "PTR-MS" to "PTR-ToF-MS".*

- Line 141 (Fig 1): it would be an improvement to add the gradient inlet heights on this figure.

  Figure 1 was designed to compactly present the setup of the Norunda tower infrastructure without taking up too much "vertical" space.

  The inclusion of all 14 sonics in the figure was considered during the original preparation of Figure 1. However, for the sake of compactness of the figure presentation and relative scaling (the to-scale distance between 37 and 60 m is rather large, and from 37 m to tower-top even more so), for the original figure it was decided to abridge this distance (between 37.9 m and tower top) in the final manuscript version of figure 1.

  This has been compounded by reviewer #1's request for an additional panel (detailed map of tree-heights showing extent of forest surround tower) to be added to figure 1 as well.

  See example figure RC2-1 (located at the end of the authors' reply to reviewer #2) for an illustration of this *to-scale* separation of sonic anemometer heights with respect to the rest of the tower infrastructure setup.

  Difficulties aside, Referee has an excellent point that Figure 1 benefits greatly from the inclusion of the sampling heights (and at the correct relative scale) within the diagram.

The revised manuscript includes an updated Figure 1 that incorporates Referee #2's feedback. Note: to achieve this panel arrangement, the Figure 1 caption was moved to an alternative location (top-right, in space between panels, instead of bottom). An example of the revised figure is included below:

[Figure]

Figure 1: Forest map, station location, and BVOC inlet setup for ICOS Norunda. (a) A map of tree heights surrounding the station flux tower (out to 1500 m radially from tower base) for the Norunda forest. (b) Location and coordinates of ICOS station Norunda in Sweden. (c) BVOC inlet, infrastructure, and instrumentation setup for Vocus PTR-ToF-MS measurements on the Norunda tower (BVOC inlet at 35 m). Shown are the heights of the on-site collection of 3D sonic anemometers (blue diamonds) and BVOC inlet (red cross) at the station flux tower. The canopy top height was at approximately 28 m. Sonic-profile anemometers were located at 1.8, 4.4, 14.8, 20.8, 26.6, 29.6 32.8, 35, 37.9, 44.8, 59.5, 74,88.5, and 101.8 m on the Norunda tower. The instrument shed contained the Vocus PTR-ToF-MS and zero-air generator for the Vocus. A blower was used to pull air through the tower inlet.

*To address the reviewer #2's comment, a list of the sonic anemometer heights has been added to the figure caption.*

- Line 178: give location of company (Ionimed)

  Added location of the company *Ionimed Analytik* - Innsbruck, Austria

- Line 179: give location of company (Ionicon)

- Added location of the company *Ionicon Analytik* - Innsbruck, Austria

- Lines 215-216: give some statistics about the tilt that was calculated.

The application of a tilt-correction to the sonic anemometer data is a standard preprocessing step for analyzing trace gas fluxes using the eddy-covariance flux method. In this manuscript, the directional planar fit method was utilized as described in both Wilczak et al. (2001) and Striednig et al. (2020).

The statistics of the tilt-correction were typical for above-canopy campaign measurements from a tower and were therefore not included. A plot (figure RC2-2) of the tilt-correction information (as generated by InnFLUX during the campaign analysis) is included for the reviewer below:

[Figure]

Figure RC2-2: Tilt correction result for the planar fit method (Norunda 2020 campaign). For the tilt correction, as described in Wilczak (2001), three coordinate rotations are applied sequentially: pitch (α), roll (β), and yaw (γ). The rotations α and β b (shown) are what align the vertical component w of the raw sonic anemometer data with the vertical z axis. The dashed lines show 95% confidence bounds estimated by bootstrapping.

The name of the tilt-correction method implemented and an additional reference for its use/settings for how it was applied (Striednig et al., 2020) was added to the sentence for clarity. To remove any potential ambiguity, the word "instrument" was also changed to "sonic anemometer". *The revised sentence is the following - In particular, a tilt-correction using the directional planar fit method (e.g., Wilczak et al., 2001; Striednig et al., 2020) was performed on the Metek sonic data to align the sonic anemometer's coordinate system with the mean wind streamlines.*

- Lines 216-217: did the calculated delay time agree with the delayed calculated from the tubing flow and geometry (lines 165-169)? Also, using this method of maximizing the correlation coefficient can introduce a bias if there is actually no flux. But since multiple compounds are being measured, this probably was not a problem.

The estimated delay time from covariance maximization agree fairly well (~3 sec) with the time delay calculated just from the tubing flow and geometry (within ~1.5sec). This is a reasonable difference, given that the Vocus computer and sonic data logger internal clocks were not synced to each other for this campaign (i.e., clock separation + desync).

As a component of the matlab flux program InnFLUX, the lag time estimate for each 30min ensemble is produced for each Vocus signal timeseries trace, and the time-delay was consistently the same/similar for strong-signal/high-flux compounds, ect. Additionally, during high flux periods for (e.g.) MT, the estimated lag time is also consistently around the same value.

- Lines 220: were the tubes in some sort of auto-sampler and collected in-situ on the tower? Or was there a sampling line and the tubes were collected at the base of the tower?

The tube samples were collected in-situ on the tower using manual sampling pumps with pre-programmed start-&-end times (30min total) with a specified flow rate (6 L min$^{-1}$). Both sampling tubes and their pumps were raised to their sampling heights just before the start of each sampling period using a rope-&-pulley system.

Below is a figure (Figure RC2-3) illustrating the setup of this in-situ sampling.

[Figure]

Figure RC2-3: Equipment setup used for paired manual BVOC sampling during the summer 2020 Norunda field campaign. (a) Example of equipment used for TD sampling at each height. Specifically, shown are the sampling pumps, TD adsorbent tubes (capped), 1/4" Teflon tubing connections for each pump inlet with Teflon Swagelok fittings (to connect to TD tubes), weather protection bags and carabiners used for attaching the pump and TD tube equipment for each height (37 and 60 m) to the rope, which was then used to lift them up the station tower infrastructure. (c) An example of how the sampling pump and TD tube equipment for each height were attached to the rope for lifting (c) The rope-pulley setup on the tower used for raising the pair of sampling pumps and their tubes to 37m and 60m on the flux tower (samplers circled in green).

- Lines 248-287: are there also co2 measurements at the tower? Could you check your approach by comparing your results to gradients in co2? There are a lot of assumptions in the theory used here, especially around the gamma factor given in Eqn. 3.

  Yes, $CO_2$ measurements were also available on the tower from the ICOS station equipment, and previous testing found the SLG-estimated flux for $CO_2$ vs the station's EC $CO_2$ flux to be comparable in quality to the comparison for Vocus/TD results for total/summed MT. SLG $CO_2$ fluxes with RSL correction have been implemented at multiple boreal sites in the past (e.g., Rannik, 1998; Rannik et al., 2004), including ICOS Norunda (e.g., Mölder et al., 1999). The approach underlying the application of SLG here (including the application of a gamma-factor to account for the roughness sublayer) had been previously tried & tested in other BVOC gradient-flux method studies (e.g., Rantala et al., 2014; Rinne et al., 2000a; Rinne et al., 2000b).

- Lines 289-297 (section 2.7): this is great that you are providing a quantitative error analysis. But given the complexity of the theory presented in the previous section, there is the potential for larger methodological errors. Please provide some literature values that give an estimate of systematic errors that are associated with flux-gradient approaches.

Referee #2's comment has been in large part addressed by a previous reply to a similar comment by referee #1.

*We refer referee #2 to this previous reply regarding a more detailed outline of the uncertainties involved as well as the revised section text.*

An evaluation of such sources of uncertainty, including systematic errors, in the SLG approach has been previously presented in Rinne et al. 2000.

Referee #2 is correct that there is potential for larger methodological errors. As alluded to in the first paragraph of section 2.6, the BVOC flux (and its uncertainty) can be described in a manner known as turbulent dispersion (i.e., $F_C = -K\frac{\delta c}{\delta z}$ for gradient-flux methods in general). In addition to the BVOC gradient itself (i.e., the measured concentration difference $c(z_1) - c(z_2)$), uncertainty also arises from the turbulent exchange coefficient $K$.

With respect the key uncertainties (including systematic error) of the turbulent transfer coefficient $K$, the main sources are the following:

- *EC measurements of buoyancy flux and friction velocity $u_*$, consisting mainly from random noise of the measurements system and sampling error (random: total ~20%).
- *Variability of estimated K due to the range of values reported in the literature for the related empirical constants (treated collectively as a single parameter set) parameterizing universal gradient-flux relationships – i.e., the parametrization of the Businger-Dyer formulas and von Kármán constant used for turbulent exchange coef K incorporated into Eq. 2. (systematic: ~25%)
- *As an independent parameter, the parameterization of the displacement height d. Rinne et al. 2000 reference 25% from a survey of values reported (ranging from d/h = 0.61 to 0.92, for d normalized by canopy height h) in Jarvis (1976). For this manuscript, the displacement height of the Norunda canopy during the 2020 campaign was evaluated using the station tower data and existing 14-level sonic anemometer profile (e.g., Rantala et al. 2014). (systematic: ~10%).

- Line: 336: need to give some more information about the fitting procedure. What statistical approach was used?

Nonlinear least-squares regression was applied to equation 7 (hybrid emission algorithm) in the manuscript (e.g., described in Bates & Watts, 1988). For isoprene (i.e., ., $f_{denovo}$ = 1), eq. 7 reduces to the classical isoprene emission algorithm (e.g., Guenther et al., 1993). For MT or SQT, when a pure-storage emission algorithm fit was being considered (i.e., $f_{denovo}$ = 0), $f_{denovo}$ was fixed at 0 and the hybrid formula simply reduces to the classical MT storage emission algorithm of Guenther et al. (1993). This approach is outlined in the section text.

Non-linear best fitting was performed following the same general algorithm and nls-fitting approach (though not software) outlined in the modeling procedures section of Pio et al. (2005). The fitting procedure was performed in the R programming language using the nls function from the Stats package (he default algorithm is a Gauss-Newton algorithm): https://www.rdocumentation.org/packages/stats/versions/3.6.2/topics/nls

The section 2.9 text was revised with the addition of the following sentence:

"To determine $E_o$, β, and $f_{denovo}$, the fitting of Eq. 7 to the campaign data was performed using nonlinear regression."

- Line 369 (Fig. 3): I am having trouble understanding the foot print information. First, the color scheme is given as red and blue in the figure caption but the figure has green and a very little bit of blue. Second, the green contours are clear, but are there separate blue contours? I can barely see the blue and I am not sure what it represents. Finally, I believe these contours are cumulative flux, but the contour lines should be specific described in the figure caption. I cannot evaluate the statement on lines (365-376) that the two contours agree with each other.

- See reply to the reviewer #1 comment for Fig. 3. The discrepancy noted by Reviewer #2 between figure and text is due to an editing error during previous manuscript preparation. *As detailed in the reply to reviewer #1, The figure 3 caption and text have been revised to correct this editing error*.
- To address the comment specific to referee #2 regarding the contours, *the caption of the figure has been revised to include the following sentence: "Footprint contour lines (green) are shown in 10% increments from 10% to 90%."*

- Line 408 (Fig. 5): give the meaning of the shaded regions for panels (f) and (i).

The two shaded regions (as opposed to the vertical gray stripes indicating the range of sunrise/sunset times) in panels (f) and (i) of Figure 5 indicate the 5th-to-95th (lighter shade) and 25th-to-75th (darker shade) percentile ranges for inverse Obukhov length L-1 (panel f) and ozone concentration in ppbv (panel i), respectively. The meaning of these shaded regions is indicated in the inset legend of panel (f).

It was also noted that the caption did not indicate the meaning vertical grey background bars nor black vertical lines. These convey the mean and range of sunrise/sunset times during the Vocus deployment (July 21 to August 27). To address this omission, the following text was added to the caption: *"In all the panels, mean sunrise and sunset (solid and dotted vertical lines, respectively) for the period of Vocus deployment (July 21 to August 27) are indicated. Vertical bars (grey) indicate the range of sunrise and sunset times during this period."*

- Line 461 (Fig. 7): why are the flux units ng/m2/s instead of the nmol/m2/s used in the other graphs? If there is no specific reason to use different units in this graph, pick one unit and be consistent.

  As noted in earlier replies to comments referees 1 & 2, *manuscript figures have been revised to improve consistency of flux unit presentation.*

- Lines 509-526 (Section 3.3.3): This section is tangential to the rest of the paper. It's great that you have additional data on different chemical species, but you have not prefaced these results in the introduction. Please remove.

  As the main focus of the campaign was terpenoid compound fluxes (isoprene, MTs, and SQTs), it is indeed tangential to the main aims of the paper. The section was intended to convey some brief additional information on biogenic but non-terpenoid VOCs as well as other relevant observations at the station. For example, MVK+MACR are two main products from isoprene photochemical oxidation, and as noted in Petersen et al. 2023 and Karl et al. 2004, can be used to estimate the average time from isoprene emission to detection at the BVOC inlet on the tower. Along these lines, referee #1 notes that this information can have relevance for other biological processes at the regional/local level. In contrast to Referee #2's comment to remove this discussion, Referee #1 also requested an expansion of this information in the text. Section was revised (eliminating redundant phrasing while retaining core information) to attempt to strike a balance between the two referee comments.

  *Text was revised to the single following paragraph:*

*"Acetaldehyde (m/z+ = 45.034) exhibited a mean daily concentration of 0.7 ppbv. Concentrations of toluene (m/z+ = 93.07) were generally low during daytime (~12 pptv) and increased during nighttime (~30 pptv), This behavior by toluene is consistent with the build-up of anthropogenic background emissions during night in the shallow nocturnal boundary layer (Karl et al., 2004). Similar behavior was found for the mass peak at m/z+ = 95.049 (i.e., phenol), which had a concentration minimum during daytime (~9 pptv) and maximum during nighttime (~40 pptv). Acetic acid (m/z+ = 61.028) was typically lowest after sunrise (~10 pptv), gradually increasing throughout the day and peaking before sunset (~33 pptv), then declining overnight. The exception to this trend occurred when high nighttime canopy concentrations coincided with similar peaks in acetone and acetaldehyde. The dirurnal signal for m/z+ = 41.039 and m/z+ = 103.112, representing the PTR-protonated hexanol fragment and the hexanol parent ion, respectively, followed a similar pattern to acetone. The minimum in hexanol concentration (~50 pptv) typically occurred in the morning following sunrise and peaked after sunset (~130 pptv). Methyl vinyl ketone and methacrolein (MVK+MACR, m/z+ = 71.049), two important intermediate products from the photochemical oxidation of isoprene, averaged 7 pptv daily."*

- Line 527: section numbering is not consistent.

  The figure numbering in this section (3.4.1) has been revised (for content and clarity). In this section, "Fig. 2" has been revised to "Fig. 2 & 9". The authors note that figure 2 contains pie charts showing the relative concentrations of MT compounds (from which the greater abundance of α-pinene vs $\Delta^3$-carene can also be observed).

- Line 544 (Fig. 9): again, units are not consistent with Fig. 2.

  Reviewer #2's comment follows from figure 9 using a ng m$^2$ s$^{-1}$ scale for the TD SLG flux results. Following revisions to the units displayed in Fig. 2 (ng m$^2$ s$^{-1}$ scale now displayed along the righthand-side y-axis), the use of ng m$^2$ s$^{-1}$ units in both figures is now consistent, with each displaying a *ng m$^2$ s$^{-1}$* scale.

- Line 551: "Based on comparison with 2022 precut TD measurements" I don't understand what this is referring to.

  The Norunda forest was clearcut in Summer 2022. The manuscript presenting the study of BVOC emissions before, during, and after the 2022 clearcut is currently in preparation. In July 2022, prior to the scheduled 2022 clear-cutting, additional TD sampling was conducted, following the same paired sampling approach (at 37 and 60 m) outlined in this study. The laboratory post-processing of these 2022 campaign

samples (conducted by the Rinnan group at Copenhagen University) was more comprehensive in the number of GC-MS calibration standards available/included.

- Lines 552-555: give ratio of TD/Vocus concentrations for each period. Also, why are the June data not presented?

It is not possible to provide these concentration ratios for June 2020, as the Vocus PTR-ToF-MS was not able to be deployed at Norunda until July 2020. A concentration comparison of the TD (i.e., summed MT) and Vocus (i.e., total MT as measured by the PTR instrument approach) is already provided in the text as well as a description of their relative scaling.

- Lines 572-575: give ratio of TD/Vocus fluxes for each period. Also, why are the June data not presented?

It is not possible to provide these flux ratios for June 2020, as the Vocus PTR-ToF-MS was not able to be deployed at Norunda until July 2020. A flux comparison of the TD-based (i.e., SLG) and Vocus-based (i.e., EC method) flux is already provided in the text as well as a description of their relative scaling. In addition, due to the nature of SLG flux uncertainties and the requirement to sum them in quadrature across all measured MT compounds (again, summed MT for TD vs total MT provided by the PTR instrument approach), the resulting uncertainty range of such a ratio would be extremely large.

- Lines 587-590: you write there is not a "substantial variation" but you should use statistical language. Because of your limited dataset, there might be 'substantial variation' that you do not have the statistical power to detect. You should simply say there is not a statistically significant difference.

Reviewer #2 is correct that the use of statistical language is appropriate for this portion of the text. The wording has been changed from "there does not appear to be in general a substantial variation" to "no statistically significant variation was noted". The text portion "following from the gradient samples" was also moved to earlier in the sentence.

The following additional revisions were made to the manuscript text:

In the co-author list -

The spelling of co-author JC's name was changed from "Jeremy Chan" to "Jeremy K. Chan". The abbreviation for the co-author's name (hereafter "JKC") was correspondingly updated in the text.

The text for the address of co-author JKC's affiliation was updated from

*"Terrestrial Ecology Section, Department of Biology, University of Copenhagen, Universitetsparken 15, DK-2100 Copenhagen Ø, Denmark"*

To

*"Center for Volatile Interactions (VOLT), Department of Biology, University of Copenhagen, Universitetsparken 15, DK-2100 Copenhagen Ø, Denmark"*

In the acknowledgements, the following additional sentence was added:

*"The post-processing of ATD-GC-MS files by J. K. Chan was supported by the Danish National Research Foundation (VOLT, DNRF168)."*

In Figure 7 –

Capitalized first word of the caption's first sentence → "mean" to "Mean".

[Figure]

Figure RC2-1: full relative to-scale illustration of the Norunda BVOC infrastructure setup with all sonic anemometer heights of the tower sonic profile.

**Author Reply References**

Guenther, A. B., Zimmerman, P. R., Harley, P. C., Monson, R. K., and Fall, R.: Isoprene and monoterpene emission rate variability: model evaluations and sensitivity analyses, J. Geophys. Res. Atmos., 98, 12609-12617, 1993.

Jarvis, P.: Coniferous forest, Vegetation and the Atmosphere, 2, 171-240, 1976.

Karl, T., Potosnak, M., Guenther, A., Clark, D., Walker, J., Herrick, J. D., and Geron, C.: Exchange processes of volatile organic compounds above a tropical rain forest: Implications for modeling tropospheric chemistry above dense vegetation, J. Geophys. Res. Atmos., 109, 2004.

Mölder, M., Grelle, A., Lindroth, A., and Halldin, S.: Flux-profile relationships over a boreal forest—roughness sublayer corrections, Agr. Forest Met., 98, 645-658, 1999.

Pio, C. A., Silva, P. A., Cerqueira, M. A., and Nunes, T. V.: Diurnal and seasonal emissions of volatile organic compounds from cork oak (Quercus suber) trees, Atmos. Environ., 39, 1817-1827, https://doi.org/10.1016/j.atmosenv.2004.11.018, 2005.

Rannik, Ü.: On the surface layer similarity at a complex forest site, J. Geophys. Res. Atmos., 103, 8685-8697, 1998.

Rannik, Ü., Keronen, P., Hari, P., and Vesala, T.: Estimation of forest–atmosphere $CO_2$ exchange by eddy covariance and profile techniques, Agr. Forest Met., 126, 141-155, 2004.

Rantala, P., Taipale, R., Aalto, J., Kajos, M. K., Patokoski, J., Ruuskanen, T. M., and Rinne, J.: Continuous flux measurements of VOCs using PTR-MS—reliability and feasibility of disjunct-eddy-covariance, surface-layer-gradient, and surface-layer-profile methods, Boreal Environ. Res., 19, 87-107, 2014.

Rinne, J., Hakola, H., Laurila, T., and Rannik, Ü.: Canopy scale monoterpene emissions of Pinus sylvestris dominated forests, Atmos. Environ., 34, 1099-1107, 2000a.

Rinne, J., Tuovinen, J.-P., Laurila, T., Hakola, H., Aurela, M., and Hypén, H.: Measurements of hydrocarbon fluxes by a gradient method above a northern boreal forest, Agr. Forest Met., 102, 25-37, 2000b.

Striednig, M., Graus, M., Märk, T. D., and Karl, T. G.: InnFLUX–an open-source code for conventional and disjunct eddy covariance analysis of trace gas measurements: an urban test case, Atmospheric Measurement Techniques, 13, 1447-1465, 2020.

Wilczak, J. M., Oncley, S. P., and Stage, S. A.: Sonic anemometer tilt correction algorithms, Boundary Layer Meteorol., 99, 127-150, 2001.

**Author Reply References**

Atkinson, R.: Gas-phase tropospheric chemistry of volatile organic compounds: 1. Alkanes and alkenes, Journal of Physical and Chemical Reference Data, 26, 215-290, 1997.

Ciccioli, P., Brancaleoni, E., Frattoni, M., Di Palo, V., Valentini, R., Tirone, G., Seufert, G., Bertin, N., Hansen, U., and Csiky, O.: Emission of reactive terpene compounds from orange orchards and their removal by within-canopy processes, J. Geophys. Res. Atmos., 104, 8077-8094, 1999.

Foken, T.: 50 years of the Monin–Obukhov similarity theory, Boundary Layer Meteorol., 119, 431-447, 2006.

[revised manuscript text omitted]